# Competing cognitive pressures on human exploration in the absence of trade-off with exploitation

Clémence Alméras [1,2] ✉, Valerian Chambon[2,3] & Valentin Wyart [1,2] ✉

Exploring novel environments through sequential sampling is essential for efficient decision-making under uncertainty. In the laboratory, human exploration has been studied in situations where it is traded against reward maximisation. By design, these 'explore-exploit' dilemmas confound the behavioural characteristics of exploration with those of the trade-off itself. Here, we propose a sequential sampling task where exploration can be compared in the presence and absence of trade-off with exploitation. Detailed model-based analyses of choices reveal specific exploration patterns arising when information seeking is not traded against reward seeking or influenced by prospective value. Human choices are directed toward the most uncertain option available, but only after an initial sampling phase consisting of repeated choices from each novel option. These findings outline competing cognitive pressures on information seeking: the repeated sampling of the current option (local uncertainty minimisation), and the directed sampling of the most uncertain option available (global uncertainty minimisation).

Making decisions often requires choosing between exploiting a familiar option whose rewarding value is known, and exploring an unfamiliar option whose value is not known[1–3]. Exploration is essential to discovering the structure of a novel environment (e.g. mapping possible itineraries from home to work in a new city), optimising behaviour (finding the fastest or most pleasant itinerary) or adapting to changes in the environment (looking for an alternative itinerary when the familiar one is unavailable). Exploration is thus adaptive to acquire information about available options, but also risky: choosing an unfamiliar option means foregoing a known reward for an uncertain one. When the known reward can be obtained without delay (e.g. choosing between two snacks at a vending machine), humans arbitrate between the costs and benefits of exploration[4].

However, not all choices bring immediate rewards. For example, when browsing restaurant reviews online, exploring the reviews of different restaurants before ordering yields information at virtually no cost. In such situations, exploration is not traded against immediate reward maximisation, but used to acquire information about available options[5]. In some cases, like browsing restaurant reviews, the information acquired concerns the future rewarding values of available options, and can be readily used when choosing which restaurant to order from. But in other cases, like orienting oneself in a new city, the acquired information does not even concern value—whose definition depends on one's particular goal at a later point in time (e.g. going to work on a weekday, or finding a new friend's place on a Sunday).

In the laboratory, exploration is usually studied using bandit tasks in which agents are presented with two (or more) options to sample from[6–8]. Each option is associated with a specific reward distribution, and the agents are incentivized to maximise rewards through sequential sampling. Across successive choices, the agents learn to choose the most rewarding option—and thus trade exploration against immediate reward maximisation as we do when choosing snacks at the vending machine.

Variants of these tasks have dissociated information acquisition from reward acquisition in time, either by controlling the amount of information received (e.g. through forced choices[9,10]), or by delaying

[1]Laboratoire de Neurosciences Cognitives et Computationnelles, Institut National de la Santé et de la Recherche Médicale (Inserm), Paris, France. [2]Département d'Études Cognitives, École Normale Supérieure, Université PSL, Paris, France. [3]Institut Jean Nicod, Centre National de la Recherche Scientifique (CNRS), Paris, France. ✉e-mail: clemence.almeras@gmail.com; valentin.wyart@inserm.fr

**Fig. 1 | Experimental paradigm. a** *Task conditions.* Left: sequence instructions. In MATCH sequences, participants were asked to draw outcomes matching a target colour (e.g. blue). In GUESS sequences, the same participants were asked to identify option-colour associations. Centre: sequential sampling phase, identical across conditions. Each sequence consisted of 8, 12, 16 or 20 choices, varying pseudo-randomly across sequences. Right: final question. In GUESS sequences, participants were asked to report the colour associated with one of the two options. **b** *Colour shades of the shapes outcomes.* Outcome colours were drawn from a continuum ranging from blue to orange. The blue-dominant bags were associated to 66% of blue outcomes and 33% of orange outcomes, and vice versa for orange-dominant bags. The distribution of probabilities over the continuum of colour shades was chosen such that the shades were linearly related to the ratio of the log odds of the categories (blue dominant or orange dominant distribution), thus, summing outcomes provided the accumulated log odds ratio of the sequence of outcomes. **c** *Option-colour pairings.* In both conditions, participants were informed that it was possible for both options to be associated to the same colour, or to different colours.

the acquisition of rewards (e.g. in the 'sampling paradigm' from Decisions From Experience (DFE) research[5,11–13], where participants first sample the environment for information before they have to make a rewarded decision). However, even in these tasks, the information acquired during the sampling phase immediately concerns the ultimate rewarding values of options. Outcomes do not immediately contribute to participants' final earnings, but since they are expressed in points, they immediately reveal which option will ultimately be rewarding, potentially biasing choices—this is similar to browsing restaurant reviews. This means that the exploration patterns displayed in such delayed-reward tasks are still constrained by the prospective rewarding values of available options.

Here, we sought to fully dissociate exploration from reward-guided decision making by designing a two-armed bandit task where the outcomes of the two options are shades of colour instead of points or monetary rewards. In themselves, outcomes thus provide information about the structure of the task (the option-colour associations), but carry no intrinsic value (Fig. 1a). We contrasted three conditions in a within-subject design. In one condition, we rewarded participants for drawing shapes matching a particular colour, which was their target colour. This instruction does confer a rewarding value to each outcome and requires participants to trade exploration against exploitation to choose the option associated with the instructed colour—it mimics regular bandit tasks. By contrast, in a second condition, the target colour was only revealed at the end of the sampling sequence. Participants freely sampled the options to learn the option-colour pairing until a final two-alternative forced choice asked them to indicate an option of a specific colour (this colour was only revealed at the end of the sequence, as the final two-alternative forced choice probe). In this condition, participants were not rewarded on a sample-by-sample basis, but only based on their ability to indicate correct option-outcome associations after the end of the sequence. This instruction thus fully dissociates exploration from any form of immediate or prospective reward maximisation. Finally, we added a third, hybrid condition where participants were asked to find an option of one particular colour, which they only had to indicate once, at the

end of the sequence of several free choices. Again, participants were rewarded based on their ability to indicate correct option-outcome associations after the end of the sequence, but in this condition, each outcome directly informed them about the options' future value—similar to a classic delayed reward task. Except for these differences in instructions, the three conditions were matched in every possible aspect so that sampling patterns could be directly compared (Figs. 1 and 5; see 'Methods').

We collected a first discovery dataset and derived task-based variables to compare the different conditions, including a computational model of human exploration patterns. We observed differences in human exploration and we validated the existence of these effects in two additional datasets, whose data were analysed using the exact same procedures (see 'Methods'—the statistics reproduced in the text are from the confirmatory datasets). Together, the findings reveal that when exploration is released from an immediate trade-off with exploitation, participants not only engage in directed exploration (focusing their exploration efforts on the most uncertain option), but they initially sample in chunks of the same option (focusing their exploration effort on a single option at a time). We show that this tendency is reduced as soon as outcomes convey information about the future rewarding value of options, even if the real reward itself is delayed. The two strategies expose competing forms of information seeking: a local form of uncertainty minimisation that aims at gathering evidence about the statistics of the current option though repetitive sampling, and a global form of uncertainty minimisation that aims at gathering evidence about the task structure through directed sampling of the most uncertain option available.

## Results

### Differences in human exploration patterns between conditions
We started by assessing participants' performance in each condition. In the condition where participants were asked to match a target colour (MATCH), we assessed performance by plotting the fraction of sequences where participants chose the option truly associated with

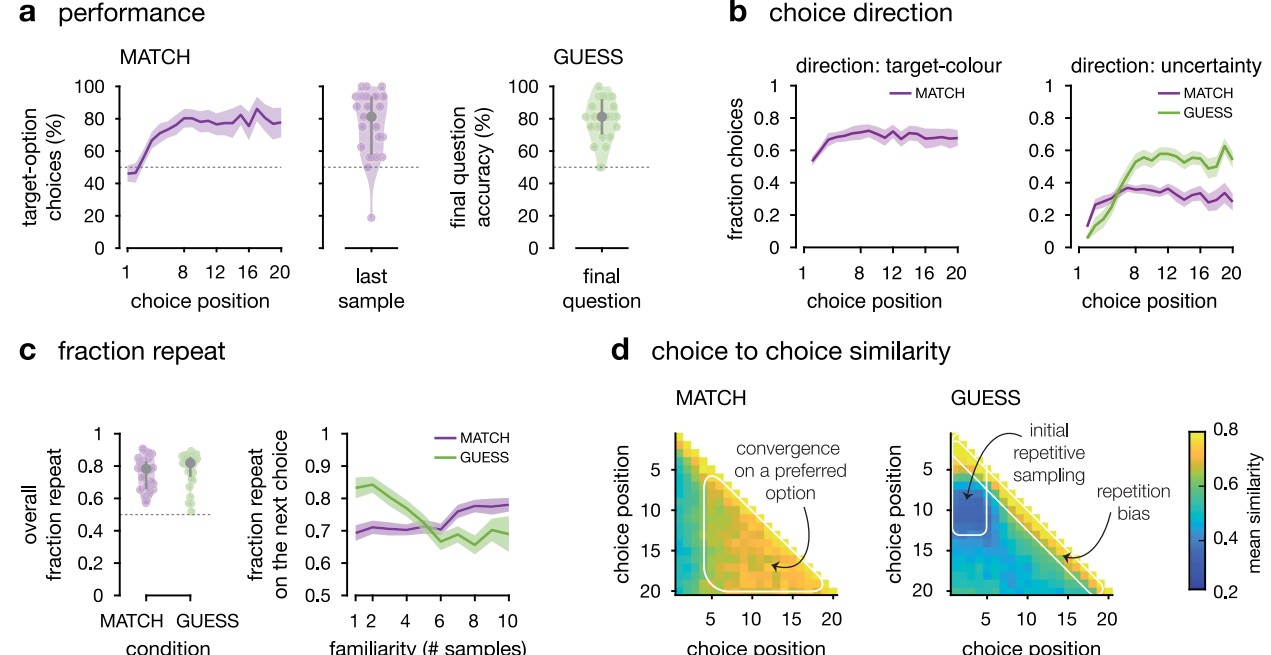

**Fig. 2 | Behavioural indices computed from the discovery dataset.** $N = 27$ participants – confirmatory dataset on Supplementary Fig. 1. MATCH condition in purple, GUESS condition in green. **a** *Performance*. Left: Fraction of choices towards the option truly associated to the target colour, plotted against choice position for single-target sequences (group means ± 95% CI, $N = 27$ participants). Center: Fraction of choices towards the option truly associated to the target colour at the last sampling decision of the sequence (8th, 12th, 16th or 20th; medians ± inter-quartile ranges, $N = 27$ participants). NB: participants did not know this was the last choice. Right: Fraction of correct responses to the final question in GUESS sequences (medians ± inter-quartile ranges, $N = 27$ participants). **b** *Choice direction*. Left: in the MATCH condition, fraction of target-directed choices at each choice in the sequence (group means ± 95% CI, $N = 27$ participants). For each choice, we defined the target option as the option with highest accumulated value in direction of the target colour category. Right: in both conditions, fraction of uncertainty-directed choices at each choice in the sequence (group means ± 95% CI, $N = 27$ participants). For each choice, we defined the uncertain option as the option with lowest absolute accumulated value. **c** *Fraction repeat*. Left: mean fraction of repeat decisions in the sequences (medians ± inter-quartile ranges, $N = 27$). Right: fraction of repeat decisions as a function of how many times they had already sampled this option earlier in the sequence (group means ± 95% within-subject CI, $N = 27$ participants). **d** *Choice to choice similarity*. For each condition, average similarity of each choice to the other choices in the same sequence. Each choice of a sequence was compared to each of the other choices in the sequence (choice #1 was compared to choice #2 through the last choice, etc.), resulting in a similarity score for each pair of choice positions within each sequence. This was averaged across sequences, resulting in an average similarity score (between 0 and 1) for each position in the sequence. Scores were averaged within conditions.

the target colour—focusing on sequences where only one option was associated with the target colour (Fig. 2a and Supplementary Fig. 1a, and Supplementary Table 1 for the descriptive statistics in the discovery sample). As expected, participants ($N = 27$ after exclusion in the discovery and confirmatory datasets; see 'Methods') chose the option associated with the target colour more often than the other option on the last choice of each sequence (confirmatory two-sided signed-rank test against chance, $z = 3.9$, $p < 0.001$, $r = 0.75$, 95% CI = [0.51: 0.88]). When participants were asked to guess the colour associated with each option (condition GUESS), we assessed performance as the fraction of correct option-outcome associations provided at the two-alternative forced choice at the end of each sequence (Fig. 1a). As expected, participants guessed better than chance the correct option-outcome associations across all sequences (confirmatory two-sided signed-rank test against chance, $z = 4.4$, $p < 0.001$, $r = 0.86$, 95% CI = [0.71: 0.93]).

We then studied participants' choice patterns in the two conditions. First, we computed the fraction of repeated choices. The average fraction of repeat decisions was similar in the two conditions (Fig. 2c and Supplementary Fig. 1c): even though the GUESS condition incentivised participants to maximise knowledge about both options at any time, it did not lead them to switching more between options overall − see also Supplementary Table 2 for the descriptive statistics in the discovery sample. Then we computed the fraction of repeat decisions as a function of the number of samples previously acquired from this option—in other words, the fraction of repeat decisions, as a function of participants' familiarity with the option before the current choice

(Fig. 2c and Supplementary Fig. 1c). In the GUESS condition, participants had a higher tendency to re-sample the option for which few samples had been acquired. This pattern was absent from the MATCH condition, where participants appeared to progressively converge on a preferred option (confirmatory repeated-measures ANOVA, interaction between number of acquired samples from the current option and condition, $F_{9,234} = 11.10$, $p < 0.001$, $\eta^2_G = 0.048$, see Supplementary Table 3). In other words, only in the GUESS condition, where participants were incentivised to maximise information about both options at any time, did they seem to repeatedly sample from the same option when they first encountered it. This pattern was confirmed by 'choice-similarity' matrices—the average similarity of a choice to all later choices in the same sequence (Fig. 2d, Supplementary Fig. 1d, and Supplementary Table 4 for the descriptive statistics in the discovery sample). We confirmed that participants converged progressively on a preferred option in the MATCH condition. In the GUESS condition, on the other hand, choice similarity matrices showed that participants not only sampled repeatedly from the same option in the first trials (choice similarity among trials 1–4, confirmatory two-sided signed-rank test against random sampling, $z = 3.9$, $p < 0.001$, $r = 0.75$, 95% CI = [0.52: 0.88]), but also sampled preferentially from the other option in subsequent trials (choice similarity between trials 1–4 and trials 8–11, confirmatory two-sided signed-rank test against random sampling, $z = -2.2$, $p = 0.024$, $r = -0.43$, 95% CI = [−0.70, −0.07]). Both conditions also showed an elevated similarity between consecutive choices.

**a.** model schematic

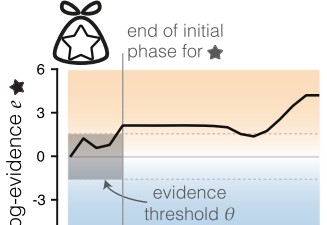

**b.** initial sampling parameters

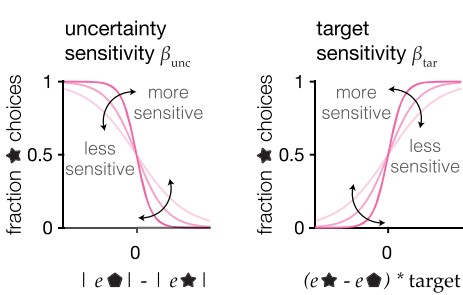

**c.** evidence accumulation

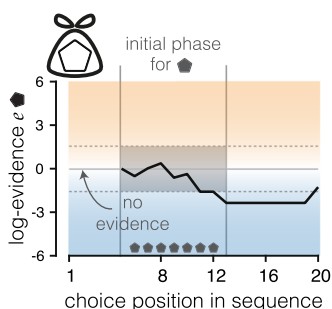

**d.** sensitivity parameters

**Fig. 3 | Model of sampling patterns. a** *Model schematic.* Left: learning was modelled as a perfect accumulation of outcomes observed at each choice. No update occurred for the unchosen option. Center, bottom: the decision variable was the sum of three components. First, the difference in absolute accumulated value between conditions (difference in uncertainty), weighted by parameter $\beta_{unc}$. Second, the difference in value accumulated towards the target-colour category (in the MATCH condition only), weighted by parameter $\beta_{tar}$. An optimal sampling agent would only feature these two parameters ($\beta_{unc} \to \infty$ in GUESS and $\beta_{tar} \to \infty$ in MATCH). Third, the decision variable of models fit to participants' choices featured the value of the previous choice, weighted by a repetition bias parameter ($b_{rep}$). This decision variable instantiated a softmax decision rule. In the ini- model, all choices were made by computing the decision variable. Center, top: the ini+ model additionally featured an initial sampling phase, where no decision variable was computed, but choices were simply repeated. When the model encountered an option for the first time, and while the evidence accumulated for this option was below a specific threshold $\theta$, the model repeated the same choice with probability $\varepsilon$. **b** *Initial sampling parameters.* The threshold on the level of evidence to reach before switching out of the initial repetitive sampling phase varied. Lower threshold hardness reintroduces some variability, similarly to an $\varepsilon$-greedy policy (probability that the agent will switch even when the evidence is below the $\theta$ threshold). **c** *Evidence accumulation.* The sum of observed colour values (equivalent to the log evidence in favour of either category) across an example sequence for both options – only the chosen option updates. An evidence threshold ($\theta$) is represented in dotted lines at around 1.5: the initial sampling phase for any option ends when its accumulated value exceeds the threshold. **d** *Sensitivity parameters.* Left: fraction of choices toward the star option as a function of the difference in absolute accumulated value between options for different values of sensitivity to uncertainty ($\beta_{unc}$, in shades of pink), on the left, and for different values of the sensitivity to the target ($\beta_{tar}$, in shades of pink), on the right.

Finally, we quantified how target-driven and/or how uncertainty-driven the choices of the participants were. In this task, the distributions of colours were designed such that summing the shades of the colour outcomes over consecutive trials was equivalent to accumulating information about the probability of the two distributions given the outcomes observed (Fig. 1b). This allowed us to identify, at each choice, based on what the participant had observed so far, the objectively most target-like option (the option with the highest accumulated value in direction of the target colour), and the objectively most uncertain option (the option with the lowest absolute accumulated value). At each trial, we could then categorise the choices of participants depending on whether they were directed toward the target option (only possible in condition MATCH), and whether they were directed toward the most uncertain option (in both conditions)–Fig. 2b and Supplementary Fig. 1b. As expected, a majority of participants' choices were target-directed in the MATCH condition (two-sided $t$-tests against 50%, confirmatory: peak

$t_{26} = 8.83$, cluster-level $p < 0.001$), and choices were uncertainty-directed in the GUESS condition more than in the MATCH condition (confirmatory two-sided paired $t$-tests between conditions: peak $t_{26} = 5.01$; cluster-level $p < 0.001$, trials 7–19). But this was not true in the first trials of each GUESS sequence, where participants' choices were more strongly biased toward the more certain option in the GUESS than in the MATCH condition (confirmatory two-sided paired $t$-test between conditions: peak $t_{26} = 3.95$; cluster-level $p = 0.010$, trials 2–5).

The specific choice pattern observed in the GUESS condition– where outcomes have no rewarding nor prospective value–differs sharply from the behaviour of an optimal sampling agent who chooses the option whose outcome is most uncertain on each trial of each sequence (see 'Methods'). Unlike participants, this optimal sampling agent does not sample repeatedly from one option, and then from the other in the first trials of each sequence (Supplementary Fig. 2).

## Modelling differences in human exploration patterns between conditions

Based on the specific features of human exploration patterns, we designed a process model of sampling choices derived from the optimal sampling agent (Fig. 3). In the MATCH condition, based on the perfectly accumulated values of the colour outcomes, the optimal sampling agent always selects the option with highest accumulated value in direction of the target colour (the most target-like option—Fig. 3d and Supplementary Fig. 2). Whereas in the GUESS condition, the optimal strategy is to always chose the option with lowest absolute accumulated value (the most uncertain option—Fig. 3d and Supplementary Fig. 2b). This allows the agent to maximise information about both shapes at any trial—thus maximising its probability of responding correctly to the final question, whose timing was not disclosed to participants. Like the optimal sampling agent, the choices of the model are sensitive to the difference in uncertainty between the two options (sensitivity parameter $\beta_{unc}$), and attracted toward the option most likely to be associated with the target colour in the MATCH condition (sensitivity parameter $\beta_{tar}$). In addition, and unlike the optimal sampling agent, the model features a repetition bias parameter $b_{rep}$, and an optional, initial sampling phase. In the beginning of the sequence, instead of relying on a decision variable computed from the three choice parameters ($\beta_{unc}$, $\beta_{tar}$ and $b_{rep}$), the model can acquire, in turn, a number of samples from each option. This initial sampling strategy is controlled by a threshold $\theta$, governing how much evidence the agent wants to initially acquire about the current option before switching to the other (see 'Methods' and Fig. 3). We fitted this model to human sampling patterns in the MATCH and GUESS conditions, and confirmed the recoverability of the model and its parameters (Supplementary Figs. 5 and 6).

Best-fitting parameter estimates obtained through maximum likelihood estimation (see Methods) confirmed behavioural observations (Fig. 4). First, participants' choices were attracted toward the option most likely to be associated with the target colour in the MATCH condition (Fig. 4a, b; sensitivity parameter $\beta_{tar}$, confirmatory two-sided $t$-test against zero: $t_{26} = 5.7$, $p < 0.001$, $d = 1.10$, 95% CI = [0.61: 1.57]). Second, participants' choices were sensitive to the difference in information acquired from the two options, but only in the GUESS condition (Fig. 4a, b, confirmatory two-sided $t$-test against 0: $t_{26} = 2.7$, $p = 0.013$, $d = 0.51$, 95% CI = [0.11: 0.91]). This means that participants' choices were attracted toward the option for which the associated colour is most uncertain only in the GUESS condition—where participants do not aim at maximising the immediate reward of each choice. There was no significant evidence for such directed exploration in the MATCH, reward maximising condition (two-sided paired $t$-test for $\beta_{unc}$ between conditions in the confirmatory sample: $t_{26} = 3.6$, $p = 0.001$, $d = 0.69$, CI = [0.26: 1.10] two-sided $t$-test for $\beta_{unc}$ in the MATCH condition against chance $t_{26} = -0.2$; $p = 0.875$; $d = -0.03$, 95% CI = [−0.41: 0.35]). Participants displayed significant repetition biases $b_{rep}$ in both conditions (Fig. 4a, b, confirmatory two-sided $t$-tests against chance: MATCH $t_{26} = 6.0$, $p < 0.001$, $d = 1.16$, 95% CI = [0.66: 1.64]; GUESS $t_{26} = 6.9$, $p < 0.001$, $d = 1.32$, 95% CI = [0.79: 1.83]), with no significant difference between conditions. The descriptive statistics for the discovery sample are reproduced in Supplementary Tables 5 and 6.

Finally, we compared variants of the model with vs. without the initial sampling phase using Bayesian model selection on cross-validated measures of model likelihood (Fig. 4a, b; see 'Methods'). In the MATCH condition, the model with the initial sampling phase was much less prevalent than the model without the initial sampling phase (Fig. 4a, b, confirmatory protected exceedance probability = 0.421). By contrast, the inclusion of an initial sampling phase improved model fits for a majority of participants in the GUESS condition (confirmatory protected exceedance probability = 0.992, probability that the model with initial phase was more prevalent in the GUESS than in the DRAW condition = 0.990). Additionally, in the GUESS condition, the

threshold level on the amount of evidence to initially accumulate for an option before switching was equivalent to a 0.75 (median, IQR = [0.54: 0.95]) probability of the underlying category. This level was significantly less in the MATCH condition (confirmatory two-sided signed-ranks test between conditions: $z = 3.2$, $p = 0.002$, $r = 0.61$, 95% CI = [0.30: 0.80]). The strength of this threshold ($\omega$) was comparably high in both conditions and both datasets.

To validate the notion that the initial sampling phase was required to reproduce the specific features of human exploration patterns in the GUESS condition, we performed a 'knock-out' simulation procedure in which we tested the effects of each aspect of the model on behavioural measures in the two conditions (Supplementary Figs. 3 and 4; see 'Methods'). The inclusion of an initial sampling phase was necessary to reproduce—both qualitatively and quantitatively—the sampling patterns observed in the GUESS condition. A repetition bias was necessary to explain participants' overall tendency to sample repeatedly from the same choice option in both conditions.

To summarise, participants' choices deviated from those of the optimal agent in two ways: quantitatively participants had lower sensitivity to condition relevant variables ($\beta_{unc}$ and $\beta_{tar} < \infty$, reflecting some randomness in choices), and qualitatively they featured repetition biases in both conditions (positive repetition bias), and a distinctive repetitive sampling strategy at the beginning of GUESS sequences only.

## Dissociating the effects of delayed exploitation and outcome valuation on exploration

The GUESS condition, in which participants sample freely the two options to learn option-outcome associations, differs from the MATCH condition, in which the same participants seek to sample a rewarded outcome, in two ways. First, in the MATCH condition, as in traditional bandits, participants are rewarded for the outcomes of each individual choice in the sequence, whereas in the GUESS condition rewards are delayed to a final, single two-alternative forced choice at the end of each sequence. Second, in the MATCH condition, outcomes are associated with a positive or negative value depending on the target colour (which option is good or bad), whereas in the GUESS condition, they only convey information about the structure of the task, but cannot be immediately related to the future rewarding value of options (which option is blue or orange). While it was the goal of our manipulation to completely release exploration from the influence of rewards, it follows that the two dimensions—delayed exploitation and outcome valuation—were confounded, and it was impossible to discern their relative contributions to the specific exploration patterns observed in the GUESS condition.

To address this issue, we designed a third, intermediate condition (FIND). Participants were asked to identify an option associated with a target colour at the end of the sequence (delayed rewards), but they were informed of the target colour from the beginning of the sequence (prospective outcome valuation—Fig. 5a). In this third condition, as in the GUESS condition, participants were not rewarded based on the outcomes they were sampling during the sequence, but rather based on their answer to a single two-alternative forced choice, at the end of the sequence. However, as in the MATCH condition, each outcome was associated with a positive or negative value depending on the target colour. This situation was akin to browsing restaurant reviews: choices yielded no immediate reward, but outcomes were still valued with respect to an end goal. It was also closer to existing tasks, like the Horizon task[9] or the 'sampling paradigm' from Decisions From Experience studies[12] Both paradigms disentangle information search from reward seeking—either by separating them in time, as in the 'sampling paradigm', or by controlling for the amount of information available to participants, as in the Horizon task. However, in both cases the immediate goal of each sampling decision is to identify which option will be most rewarding ultimately, and the outcomes convey

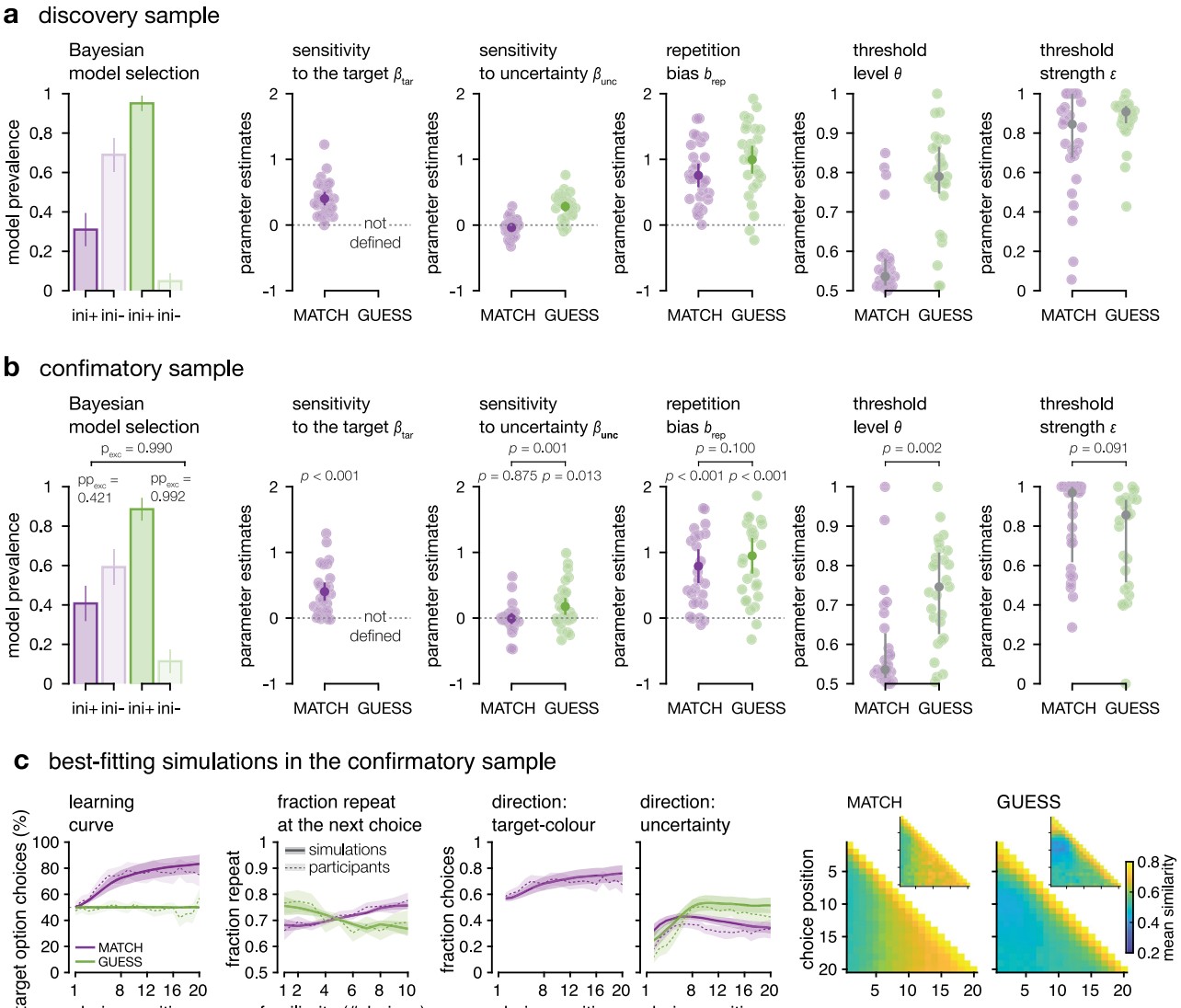

**Fig. 4 | Model fits.** MATCH condition in purple, GUESS condition in green.
**a** *Discovery sample* ($N = 27$ participants). Left: prevalence of the model with an initial sampling phase in each condition (mean ± standard deviation). The following subpanels show the distribution of participants estimates for each parameter (in colours: group means ± 95% CI for the sampling parameters ($\beta_{tar}$; $\beta_{unc}$; $b_{rep}$); in grey: group medians ± inter-quartile ranges for the bounded initial phase parameters $\theta$ and $\varepsilon$). **b** *Confirmatory sample* ($N = 27$ participants). Left: prevalence of the model with an initial sampling phase in each condition (mean ± standard deviation). Protected exceedance probability ($p_{exc}$) of the ini+ model against the ini- model in each condition, and probability that the ini+ model is more prevalent in the GUESS than in the MATCH condition. Following subpanels show the distribution of participants estimates for each parameter. For sampling parameters ($\beta_{tar}$; $\beta_{unc}$; $b_{rep}$), in colours: group means ± 95% CI, two-sided $t$-tests against 0 and two-sided paired $t$-tests between conditions. For bounded, initial phase parameters ($\theta$, $\varepsilon$), in grey: group medians ± inter-quartile ranges, two-sided signed-rank tests between conditions. **c** *Best-fitting simulations in the confirmatory sample*. Simulations ($n = 1000$) of the ini+ model featuring the initial sampling threshold, using the participants' best-fitting parameters ($N = 27$). Simulations in solid lines, participant behaviour in dotted lines. Learning curve and choice direction: lines plot the group means and ribbons plot the 95% CI, for the fraction repeat by familiarity, group means and within-participant 95% CI, participants' choice similarity matrices are inlaid.

information as a numerical value in the same unit as the reward. In other words, it is expected that a prospective reward value already taints the outcomes and biases choice patterns—i.e. participants will seek options associated to higher rewards in the future. While the original GUESS condition removed this concern, the addition of the FIND condition eliminates a gap in the paradigm. In a nutshell, the MATCH condition features immediate rewards, the FIND condition delays rewards but allows for prospective valuation, and the GUESS condition delays rewards and prevents prospective outcome valuation. The contrast between the FIND and MATCH conditions reflects the difference between delayed and immediate exploitation, whereas the contrast between the FIND and GUESS conditions reflects the

difference between the sampling of outcomes with and without rewarding value.

In this third dataset, participants ($N = 31$ after exclusion; see 'Methods') replicated the behavioural patterns previously described in the MATCH and GUESS conditions. They reached similar performance in both of the original conditions (Fig. 6a), and in the GUESS, but not in the MATCH condition, they initially sampled in chunks of repeated decisions for each option in turn. This was visible both in behavioural indices (Fig. 6) and in the distributions of parameter estimates (Fig. 7a). The statistics for these conditions are reproduced in Supplementary Tables 1– 6. We focus the remainder of this section on the third, FIND condition.

**a**  task conditions

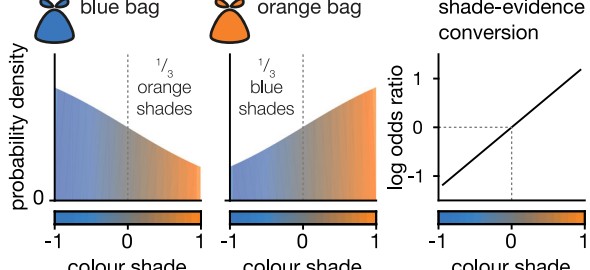

**b**  colour shades of the shapes outcomes

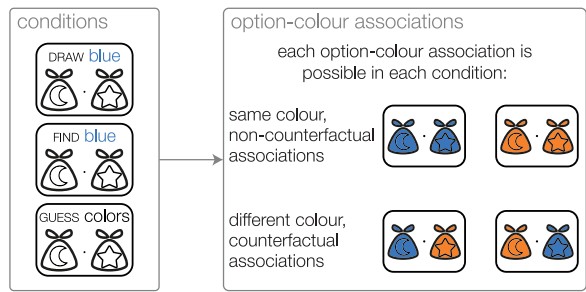

**c**  option-colour pairings

**Fig. 5 | Task design for the third dataset. a** *Task conditions*. Left: sequence instructions. In MATCH sequences, participants were asked to draw outcomes matching a target colour (e.g. blue). We informed them of the target at the beginning of the sequence and they knew they were rewarded based on the colour hue of each outcome. In FIND sequences, the same participants were asked to indicate a blue (or orange bag) at the final question. They were also given the target in the beginning of the sequences but they knew they were only rewarded based on their accuracy of their response to the final question – participants had a target, but exploitation was delayed. In the GUESS sequences, participants did not know in advance which colour they would have to indicate, but knew that they were rewarded only for being accurate on their final response - they had no target during the sampling sequence, and exploitation was delayed. Center: the sequential sampling phase was identical in all conditions, each sequence consisted of 8, 12, 16 or 20 choices, varying pseudo-randomly across sequences. Right: a final question was asked only in the GUESS and FIND conditions. **b** *Colour shades of the shapes outcomes*. Colour distributions were identical to previous datasets: blue-dominant bags produced 66% of blue outcomes and 33% of orange outcomes, and vice versa for orange-dominant bags. **c** *Option-colour pairings*. Participants were informed all colour-option associations were possible in all conditions.

In the FIND condition, participants responded accurately to the final question (Fig. 6a, median = 93.8%, IQR = [87.5% 100%], two-sided signed-rank test against 50% $z = 4.9$, $p < 0.001$, $r = 0.88$, 95% CI = [0.76: 0.94]). They also displayed a preference for sampling the target-coloured option during the sequence even though it was explicitly stated that this did not contribute to performance (Fig. 6a, median at the last sample = 62.5%, IQR = [50.0% 75.0%]). However, this tendency was less pronounced than in the MATCH condition (two-sided signed-ranks test between conditions: $z = 4.2$, $p < 0.001$, $r = 0.75$, 95% CI = [0.54: 0.87]), and the fact that participants selected the option associated with the target colour much more often on the final choice (93.8%) than on the last sampling choice of the sequence (62.5%) shows that they effectively delayed exploitation until the final choice.

Comparably to the original datasets, there was no significant difference between conditions in the average fraction of repeat decisions, (Fig. 6b) but only in the fraction of repeat decisions as a function of the number of previous samples acquired from this option (Fig. 6b; interaction between number of acquired samples from the current option and condition, $F_{18,540} = 6.2$, $p < 0.001$, $\eta_G^2 = 0.027$). As in the GUESS condition, participants tended to re-sample more from options for which few samples had been acquired in the FIND condition than in the MATCH condition, whereas the converse was true for options for which several samples had been acquired. On the other hand, choice similarity matrices in the FIND condition only partially resembled those of the GUESS condition. Participants did sample repeatedly from the same option in the early trials—similar to the GUESS condition (Fig. 6d, choice similarity among trials 1–4, two-sided signed-rank test against random sampling, $z = 3.9$, $p < 0.001$, $r = 0.71$, 95% CI = [0.47: 0.85]). But unlike in the GUESS condition, there was no evidence in

choice similarity matrices that they sampled preferentially from the other option in subsequent trials (similarity between trials 1–4 and trials 8–11, two-sided signed-rank test against random sampling in FIND, $z = -0.4$, $p = 0.681$, $r = -0.07$, 95% CI = [−0.42, 0.29]; two-sided signed-rank test between conditions GUESS and FIND: $z = -3.1$, $p = 0.002$, $r = -0.56$, 95% CI = [−0.76, −0.25]). Choice direction in the FIND condition also displayed features from both original conditions. As in the GUESS condition, participants made more uncertainty-directed choices in the FIND condition than in the MATCH condition (Fig. 6c, two-sided paired $t$-tests between the FIND vs. MATCH condition peak $t_{30} = 5.53$, cluster-level $p = 0.023$; no significant difference between the FIND and GUESS conditions. And, as in the MATCH condition, participants' choices were biased towards the more target-coloured option. This, however, was less pronounced than in the original MATCH condition (two-sided $t$-test against chance: peak $t_{30} = 6.43$, cluster-level $p < 0.001$; two-sided paired $t$-tests between FIND and MATCH conditions: peak $t_{30} = 10.9$, cluster-level $p < 0.001$).

Model fits corroborated these observations. First, the sensitivity of participants' choices to target outcomes, $\beta_{tar}$, was lower in the FIND than in the MATCH condition (Fig. 7a; one-sided $t$-test against zero: $t_{30} = 5.8$, $p < 0.001$, $d = 1.03$, 95% CI = [0.59, 1.47]; one-sided paired $t$-test between conditions: $t_{30} = 5.6$, $p < 0.001$, $d = 1.00$, 95% CI = [0.56, 1.43]). Participants' choices were also directed toward the more uncertain option in the FIND condition, as in the GUESS condition (Fig. 7a.; sensitivity parameter $\beta_{unc}$, one-sided $t$-test against zero in FIND: $t_{30} = 2.6$, $p = 0.008$, $d = 0.46$, 95% CI = [0.08: 0.83]; no significant difference between the FIND and GUESS conditions), but unlike the MATCH condition (one-sided paired $t$-test between FIND and MATCH: $t_{30} = -1.9$,

**a** performance

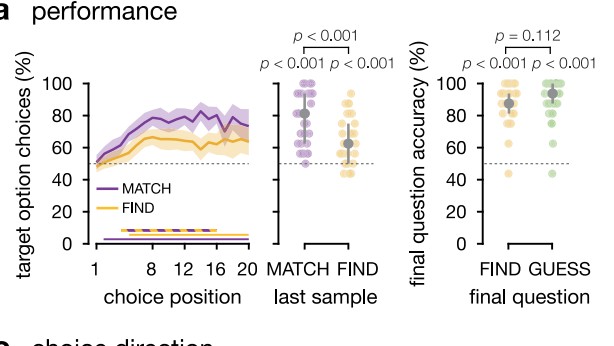

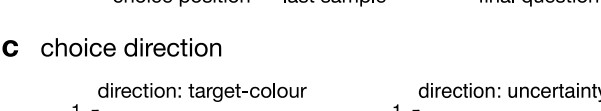

**b** fraction repeat

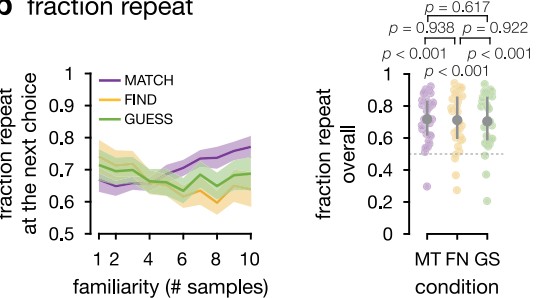

**c** choice direction

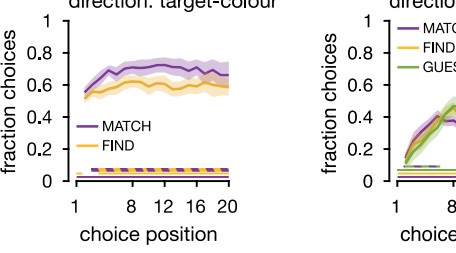

**d** choice to choice similarity

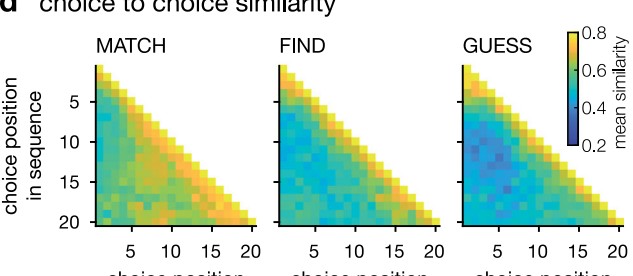

**Fig. 6 | Behavioural indices computed from the third dataset.** $N = 31$ participants, MATCH condition in purple, GUESS condition in green, FIND condition in yellow. Tests are two-sided signed-rank tests against chance or two-sided, paired signed-rank tests between conditions. **a** *Performance*. Left: The fraction of choices towards the option truly associated to the target colour is plotted against choice position for single-target sequences in the MATCH condition (group means ± 95% CI). The variable is also plotted for the FIND condition but should not be interpreted directly as performance for participants were explicitly instructed that their sampling choices did not count towards performance. Center: Fraction of choices towards the option truly associated to the target colour at the last sampling decision of the sequence (8th, 12th, 16th or 20th trial)−medians ± inter-quartile ranges. NB: participants did not know this was the last choice. Right: Fraction of correct responses to the final question in GUESS and FIND (medians ± inter-quartile ranges). **b** *Fraction repeat*. Left: fraction of repeat decisions in the discovery dataset, as a function of how many times they had already sampled this option earlier in the sequence (group means ± 95% within-subject CI). Right: mean fraction of repeat decisions in the sequences (medians ± inter-quartile ranges). **c** *Choice direction*. Left: towards the most target-like option. At each choice in the sequence, fraction of the times when the participant chose the option with the highest current accumulated level of evidence, based on participants' previous choices (group mean ± 95% CI). Right: towards the currently most uncertain option. At each choice in the sequence, fraction of the times where the participant chose the option with the current lowest absolute accumulated level of evidence, based on the participant's previous choices (mean ± 95% CI). Thick horizontal lines indicate clusters of significant differences between conditions. **d** *Choice to choice similarity*. Average similarity of each choice to each other choice in the same sequence - averaged over sequences and participants within condition.

---

$p = 0.033$, $d = -0.34$, 95% CI = [−0.70, 0.02]). As in previous datasets, participants' choices showed similar repetition biases, $b_{rep}$, across all three conditions (Fig. 7a; one-sided $t$-tests against zero: $p < 0.001$, $d > 0.69$ in all conditions, no significant difference between conditions).

In the FIND condition, the presence of an initial sampling phase only improved the fits for about a third of tested participants (Fig. 7a, mean = 35%, s.d. = 8.2%, protected exceedance probability = 0.332)−more than in the MATCH condition (mean = 12%, s.d. = 5.5%, protected exceedance probability = 0.027), but less than in the GUESS condition (mean = 81%, s.d. = 6.7%, protected exceedance probability = 0.953−see Fig. 7a for comparisons.). The level of evidence which participants initially wanted to secure about the current option before switching ($\theta$) was lower across participants in the FIND than in the GUESS condition (one-sided signed-rank test between the FIND and GUESS conditions: $z = -2.5$, $p = 0.006$, $r = -0.45$, 95% CI = [−0.70, −0.12]). The strength of the threshold was similarly high across all conditions. Finally, the inclusion of an initial threshold was necessary to reproduce the early sampling patterns observed in the GUESS and FIND conditions only. By contrast, as in previous datasets, a repetition bias was necessary to explain participants' overall tendency to sample repeatedly from the same option in all three conditions (Supplementary Fig. 7).

In summary, this additional dataset confirmed that it was only when rewards were delayed to the end of the sequence that participants engaged in uncertainty driven exploration and in initial repetitive sampling (in the FIND and GUESS conditions, but not in the MATCH condition). However, the prevalence of initial repetitive sampling was diminished when outcomes conveyed direct information about the future rewarding value of options (in the FIND condition: which option is the target, similar to the usual delayed rewards tasks[9,12]), compared to when they were only informative about the structure of the task (in the GUESS condition: which option is orange or blue).

## Positive covariations between exploration patterns across individuals

Human exploration patterns in the GUESS condition show two specific features absent from the MATCH condition. First, participants preferred sampling the option whose associated outcome is currently more uncertain (feature 1). But they also displayed initial sampling phases where sampling proceeds in chunks of repeated choices from the same option until a certain level of information is reached (feature 2). The first feature reflects a more global form of information seeking and is statistically optimal in this task: an optimal sampling agent would always choose the more uncertain option to maximise global knowledge and response accuracy during the final choice (Fig. 3, Supplementary Fig. 2 and 'Methods'). The second feature, on the other hand is a more local form of information gathering (reducing uncertainty about a single option at a time), and is statistically suboptimal (Supplementary Fig. 2b). From a rational standpoint, we would thus expect initial repetitive sampling to be detrimental to performance, and to co-occur with less choices directed towards the most uncertain option (more statistically suboptimal behaviour). But it is also possible that these two features of exploration are not traded-off, and rather

**a**  model fits

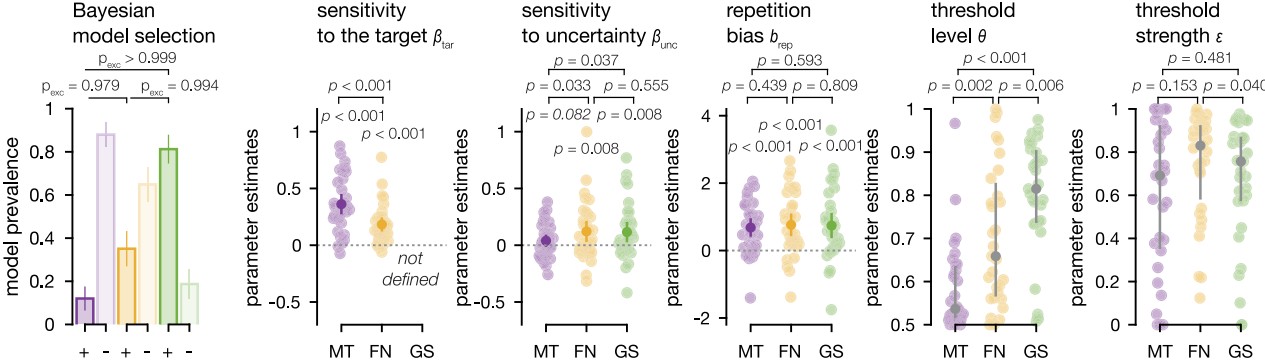

**b**  best-fitting simulations

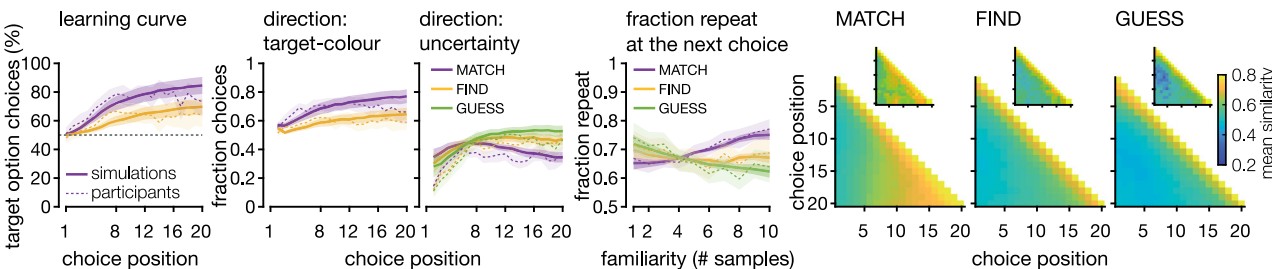

**Fig. 7 | Model results in the third dataset.** $N = 31$ participants, MATCH condition in purple, GUESS condition in green, FIND condition in yellow. **a** *Model fits*. Left: prevalence of the model with an initial sampling phase in each condition (mean ± standard deviation). Protected exceedance probability ($p_{exc}$) of the ini+ model against the ini- model in each condition, and probability that the ini+ model is more prevalent in one vs. in the other condition. The following subpanels show the distribution of participants estimates for each parameter ($N = 31$ participants). For sampling parameters ($\beta_{tar}; \beta_{unc}; b_{rep}$), in colours: group means ± 95% CI. $\beta_{tar}$ and $\beta_{unc}$: one-sided signed-rank test against chance and one-sided, paired signed-rank test between conditions. $b_{rep}$: one-sided signed-rank test against chance and two-sided, paired signed-rank test between conditions. For bounded, initial phase parameters ($\theta, \varepsilon$), in grey: group medians ± inter-quartile ranges. $\theta$: one-sided, paired signed-rank test between conditions. $\varepsilon$: two-sided, paired signed-rank test between conditions. **b** *Best-fitting simulations*. Simulations ($n = 1000$) of the ini+ model using the best fitting parameter values obtained from fitting the model to participants ($N = 31$ participants). Choices were obtained from simulating the model and analysed using the same pipeline as the choices of the participants. Learning curve (group means ± 95% CI, $N = 31$ participants); choice direction (group means ± 95% CI, $N = 31$ participants); fraction of repeat decisions (group means ± 95% within-subject CI, $N = 31$ participants); choice-to-choice similarity (averaged over participants and sequences).

reflect two different exploration strategies which can be co-expressed in the same individuals.

To address this issue, we studied covariations in exploration patterns in a between subject manner by pooling the three datasets together (total $N = 85$). We split participants based on which of the model with (*ini+* group, $N = 67$) or the model without an initial evidence threshold (*ini-* group, $N = 18$) best fit their behaviour, but to ensure a fair comparison between groups, we then compared parameter estimates obtained in the most specified version of the model (the version with initial threshold, ini+). As expected from this splitting criterion, in the GUESS condition, participants in the first group (*ini+*) showed a higher tendency to initially resample options (Fig. 8a, b, d and Supplementary Fig. 9b).

From this post-hoc analysis, the two exploration strategies (features 1 and 2) appeared to co-occur. Participants in the *ini+* group, who displayed a stronger tendency to initially repeat choices also showed a higher tendency to later prefer the most uncertain of options. This was visible on the choice direction curves (Fig. 8b) and in higher estimates for the sensitivity to uncertainty (Fig. 8e; $\beta_{unc}$ mixed-effects ANOVA, main effect of group (*ini+* or *ini-*): $F_{1,83} = 6.82$, $p = 0.011$, $\eta_G^2 = 0.055$; interaction between group and condition: $F_{1,83} = 8.49$, $p = 0.005$, $\eta_G^2 = 0.030$). In other words, the suboptimal, initially local information-seeking strategy was expressed along with the more optimal feature of global information-seeking strategies later on in the sequences.

Additionally, we observed that this combination of optimal and suboptimal exploratory features was associated to repetition bias and reward drive. Participants in the ini+ group also displayed stronger repetition biases, which is another statistically suboptimal feature. Their choice similarity matrices displayed thicker diagonals (Fig. 8d) and model fits showed higher $b_{rep}$ estimates (Fig. 8e; mixed-effects ANOVA; main effect of group: $F_{1,83} = 4.01$, $p < 0.05$, $\eta_G^2 = 0.041$, no significant interaction), which were strongly correlated across conditions ($N = 85$, linear correlation: $r^2 = 0.76$, 95% confidence interval = [0.63:0.84]). However, ini+ participants also had a more pronounced tendency to prefer the target-coloured options in the MATCH condition, which is optimal in this condition (choice direction curves on Fig. 8b; higher $\beta_{tar}$ sensitivity parameter on Fig. 8e; two-sided two-sample *t*-test between groups of participants: $t_{83} = 2.20$, $p = 0.031$, $d = 0.58$, bootstrapped 95% CI = [0.03: 1.10]). This strategic mixture of normatively optimal and suboptimal features was also associated to higher performance at the final question for ini+ participants (Fig. 8c; two-sided, two-sample rank-sum test between groups of participants in GUESS: $z = 2.1$, $p < 0.004$, probability of superiority = 0.66, bootstrapped 95% CI = [0.49, 0.82]), suggesting that the presence of suboptimal tendencies to repeat choices, both during the initial sampling phase and later throughout the sequence, was not due to mere task disengagement or inattention.

Taken together, these analyses suggest positive covariations between exploration patterns across tested participants and

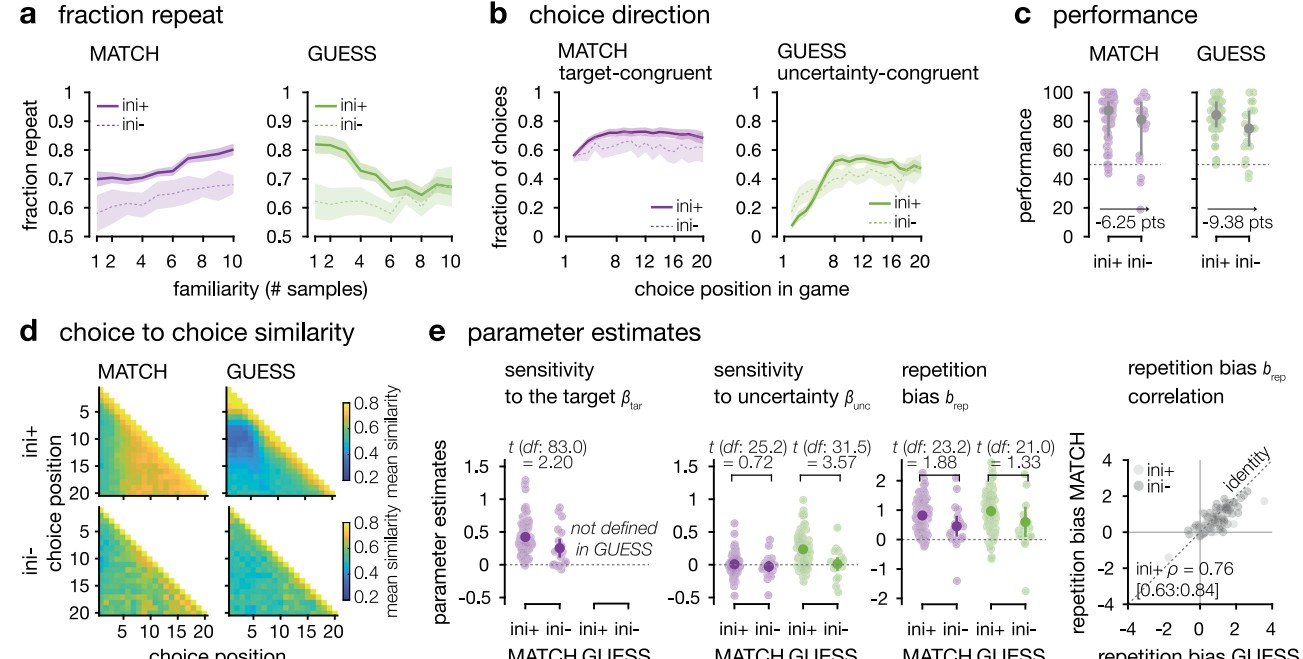

**Fig. 8 | Inter-individual differences in sampling patterns.** The three datasets were pooled N and participants (N = 85) were split in two groups based on whether a model with (ini+, N = 67 participants) or without (ini-, N = 18 participants) an initial sampling phase better accounted for their exploration patterns in GUESS sequences. MATCH condition in purple, GUESS condition in green. **a** *Fraction repeat*. Left: fraction of repeat decisions in the MATCH condition, as a function of the number of times the same option had already been sampled in the sequence, for participants whose behaviour was best fit by the ini+ (solid lines) or the ini- (dashed lines) model (means ± 95% CI). Right: fraction of repeat decisions in the GUESS condition, as a function of the number of times the same option had already been sampled in the sequence, for participants whose behaviour was best fit by the ini+ (solid lines) or the ini- (dashed lines) model (means ± 95% CI). **b** *Choice direction*. Left: fraction of target-directed choices in the MATCH condition for participants whose behaviour was best fit by the ini+ (solid lines) or the ini- (dashed lines) model (means ± 95% CI). Right: fraction of uncertainty-directed choices in the GUESS condition for participants whose behaviour was best fit by the ini+ (solid lines) or the ini- (dashed lines) model (means ± 95% CI). **c** *Performance*. Left: Fraction of choices made towards the option truly associated to the target distribution at the last sampling choice (8th, 12th, 16th or 20th) of single-target MATCH sequences (median ± inter-quartile range). Right: Fraction of correct responses at the final question of single-target GUESS sequences (median ± inter-quartile range). **d** *Choice to choice similarity*. Similarity of each choice to each of the other choices in the same sequence, averaged across sequences and participants. **e** *Parameter estimates*. Parameter estimates for ini+ vs. ini- participants: sensitivity to the target, to uncertainty, and repetition bias (group means ± 95% CI). Right: correlation of the repetition bias estimated in the MATCH *vs.* in the GUESS condition, for ini+ and ini- participants.

conditions. Stronger initial repetitive sampling could be associated to stronger sustained repetition bias (two statistically suboptimal features), but also to higher sensitivity to condition-relevant variables ($\beta_{tar}$ and $\beta_{unc}$, two statistically optimal features), and consequently higher performance.

## Discussion

Exploration is central to efficient decision-making: discovering and monitoring the current structure of one's environment allows agents to choose goal-relevant actions and to adapt to changes in their state. In the laboratory, however, exploration is generally studied in terms of its trade-off with immediate reward maximisation, using tasks that involve explore-exploit dilemmas (e.g., multi-armed bandits). Even in delayed-reward tasks, the outcomes sampled are typically immediately informative of the ultimate rewarding value of options, potentially also distorting exploratory sampling patterns[9,11,13]. Here, we sought to study human exploration patterns in situations where choices were completely unbiased by reward maximisation—as is often the case outside the laboratory. For this purpose, we compared sequences of choices where exploration conflicted with the exploitation of a rewarding option (similar to typical explore-exploit tasks), sequences where rewards were delayed but outcomes conveyed information about the prospective value of options (similar to existing delayed-rewards tasks), and sequences where choice options were only associated with reward after the end of the sampling phase—and exploration therefore did not conflict with reward maximisation. Across three datasets, we

obtained replicable evidence of specific exploration patterns arising in situations where exploration is released from reward maximisation. These patterns outline competing cognitive pressures on information gathering – between the continued sampling of the current option to refine the accuracy of its estimated statistics (a 'local' form of uncertainty minimisation), and the directed sampling of the most uncertain option to map the task structure (a 'global' form of uncertainty minimisation).

First, these results replicate previous findings, showing that participants always display random exploration, but only engage in directed sampling of uncertain options when rewards are delayed or horizons lengthened. Specifically, participants' choices were significantly directed toward the most uncertain option in the GUESS and FIND conditions, where participants were rewarded based on a single two-alternative forced choice at the end of each sequence. This was not the case in the MATCH condition—where they received rewards at each choice—despite matched objective levels of uncertainty across all three conditions. Targeted sampling of uncertain options has been documented in information gathering tasks[14–17], but when rewards are involved it critically depends on lengthening the temporal 'horizon' of the task[9,10]. Our results offer converging evidence. Moreover, because the paradigm allowed to contrast different delayed rewards situations, we observed that directed exploration appeared to only be sensitive to delaying rewards, but we found no evidence that it depended on whether outcomes were prospectively informative about the ultimate rewarding value of options (i.e., it reached comparable levels in the

GUESS and FIND conditions). In addition to directed exploration, which is optimal in the task, participants also displayed significant randomness in all conditions. We captured this randomness using a 'softmax' selection policy as in earlier work[2,3,6], but recent work has shown that some of this randomness can be triggered by random variance (*a.k.a.* noise) in the learning of option values and not only by stochasticity in the selection policy[18]. Future work should dissociate the relative contributions of learning noise and choice stochasticity to the variability of participants' choices.

Nevertheless, the exploration patterns we observed in delayed reward conditions (GUESS and FIND) diverged significantly from directed exploration in ways that cannot be explained by random exploration, learning noise, blind perseveration, or other stereotyped strategies (see also Supplementary Discussion). Specifically, participants repeated their previous choices much more than predicted by normative accounts: both throughout the sequences, but also specifically at the beginning of new sequences. Sustained repetition biases throughout the sequences in the delayed rewards conditions do not lend themselves to the traditional interpretation. In sequential sampling tasks (including bandits), they are typically attributed to a form of confirmation bias—either due to an asymmetric learning process favouring choice-supportive evidence[19–21], or because of a biased choice process assigning value to previous choices[22,23]. This account alone however, cannot explain our findings, as they describe confirmation biases as affecting the option/action values. Yet, in the GUESS condition, options are not associated to a rewarding value, thus repeating a previous choice does not—in itself—validate it. We suggest that in this condition, participants may instead be biased toward repeating their previous choice to estimate the statistics of the current option more accurately—a local, option-specific, form of uncertainty minimisation. While directed exploration reduces global uncertainty about the task structure by targeting the most uncertain option, repeating choices reduces local uncertainty by improving knowledge about the option currently being sampled.

In this perspective, early repetitive sampling could also reflect a prioritisation of local over global uncertainty reduction. When they first encountered a novel option (at the beginning of each sequence), participants showed a tendency to re-sample the novel option several times in a row, as if aiming to reach a specific, initial level of certainty about the option before switching. We modelled these early repetitions through a threshold on the amount of accumulated information to reach before switching. In the GUESS condition, because participants could be asked about any option at any time, performance relied on maximising knowledge about both options' statistics at each choice. In other words, only global uncertainty reduction through directed exploration of the most uncertain options was optimal—see Supplementary Fig. 2b. Yet, participants only engaged in directed exploration later in each sequence. Thus, it is precisely in conditions where they were incentivised to minimise global uncertainty that participants chose to prioritise local uncertainty minimisation. This likely reflects a trade-off between information search and cognitive costs: while directed sampling of uncertain options was most efficient in reducing global uncertainty, it was also cognitively costly to update beliefs about several options in parallel—e.g. switch costs, which have been associated with mental effort[24,25] and even subjective pain[26]. Initially prioritising local information by chunking samples into series of a single option likely alleviated these costs. This strategy, however, was only used by tested participants when rewards were delayed.

The trade-off between local and global uncertainty minimisation was specifically revealed by our paradigm. Indeed, this trade-off was visible whenever exploitation was delayed to the end of the choice sequence (GUESS and FIND), but attenuated when outcomes carried information about the ultimate rewarding value of options (FIND), and it was completely absent in the immediate reward condition (MATCH) despite the tight match of task variables across all conditions. Delaying

rewards is thus a necessary condition for this behaviour to arise, but it is also sensitive to the prospective value of outcomes—such selectivity explains why it has not been described in earlier work using tasks involving explore-exploit dilemmas[6,9,10]. However, while our findings highlight a balance between local and global uncertainty minimisation, they also relate to previous work showing evidence for strategy shifts over the course of a single sampling sequence. In a three-alternative version of the Horizon task, Dubois and colleagues showed that participants initially relied on a novelty heuristic in addition to more complex exploration computations[27]. Similarly, Spektor and Wulff[13] have shown that in the '*sampling paradigm*' of Decisions from Experience (DfE) experiments, participants prioritised observing all possible outcomes first, only switching to targeting the most uncertain options once all unique outcomes had been observed. Our results also document how participants flexibly adjust exploration strategies within a single sampling sequence.

A specific feature of our laboratory-based, controlled study is the use of environments with only two choice options. This feature facilitates the comparison of our findings to the existing literature[5,6,9,10], but the ecological validity of such a choice problem is indisputably limited[28,29]. However, our findings suggest that participants structure their sampling patterns in repetitive chunks as a way to minimise uncertainty in a cost-minimising fashion. The costs which we believe drive participants sampling patterns should only increase for choice problems with more than two options, such that initial repetitive chunks should also be visible in larger option spaces—in the absence of any explore-exploit trade-off. Furthermore, comparable sampling strategies have previously been described as an efficient adaptation to large hypothesis spaces[30]. The fact that we observe such patterned sampling in a two-option choice setting provides some indication that it stems from cognitive limitations (such as the cost of updating multiple hypotheses in parallel) rather than from adaptive optimisation to multi-alternative environments. It would be interesting for future research to test our findings in more naturalistic environments. Our study also focused on a specific feature of sampling patterns (their sensitivity to immediate or prospective rewards). Future research should investigate how this feature interacts with dimensions of choice behaviour—for example participants' prior expectations about option statistics, the types of uncertainties displayed[13], or the underlying structure of the option space[30]. Another constraint of the laboratory-based setting of the study is the sample size available (three datasets of about 30 participants each). This sample size was sufficient to replicate the effects we had observed in the discovery dataset, which we replicated twice, but it would be interesting for future research to replicate our findings again, in a more representative dataset.

We observed significant individual differences in our study: most, but not all participants displayed patterned sampling at the beginning of delayed exploitation sequences. And even when such patterned sampling was present, the specific threshold of evidence to reach before switching varied between participants. These individual differences allowed us to relate the early patterned sampling to other aspects of participants' behaviour in a between-participant approach. Participants who displayed initial repetitive sampling also had higher sustained repetition biases. But they were also more accurate at identifying the overall structure of each sequence (i.e. the option outcome associations), and were more sensitive to task-relevant decision variables even in conditions where they did not display the patterned sampling—i.e. in the MATCH condition, where sequences required to maximise immediate reward. These results are reminiscent of work from Hills and colleagues[5], using the "*sampling paradigm*" in Decisions From Experience tasks (where participants who sampled for longer also produced longer chunks for each option), and of work from Abir and colleagues[31], using a two-step task (in which participants who obtained higher performance also had higher repetition bias and more strategic sensitivity to uncertainty). The positive relationship we

document between initial repetitive sampling—a suboptimal strategy in normative terms—and task performance is important because it suggests that patterned sampling reflects an active cognitive strategy, rather than mere task disengagement or inattention. It suggests that statistically suboptimal strategies could be more cognitively efficient for human participants, operating under specific biological constraints. Even though they stem from cognitive limitations in the first place, strategies which produce objectively degraded evidence could be closer to cognitively optimal, under the biological constraints of the human brain (see[18,32,33] for related findings).

The interpretation of this positive relation between initial repeated sampling and task performance raises two questions. First, the question of the mechanisms through which repetitive sampling leads to higher performance despite degraded evidence. Repetitive sampling reduces switch costs[24,25] and could thus alleviate the cognitive effort required—e.g. by improving the processing of the evidence sampled (i.e. an attentional effect), by reducing the number of retrievals of the accumulated evidence associated with an option (i.e. a memory effect), or through another mechanism. A second question is whether the participants who deploy this patterned sampling strategy have different psychological traits or cognitive abilities than those who do not. Future research should investigate whether and which psychological traits affect participants' tendency to display this sampling strategy. Differences in impulsivity, orientation toward future events or attitude towards uncertainty could potentially explain individual differences in the patterned sampling of choice options observed in this study[5,29,34–36].

Taken together, our findings delineate specific features of human exploration patterns when freed from the trade-off with reward maximisation. Stripping sampled options from their rewarding value revealed competing cognitive pressures on human information seeking: 1. a local form of information seeking that aims at reducing uncertainty about the currently sampled option, and 2. a global form of information seeking that aims at reducing the overall uncertainty about the task structure. Such tension is neither described nor present in conditions eliciting explore-exploit dilemmas. We speculate that this paradoxical prioritisation of local uncertainty minimisation stems from a trade-off between information seeking and cognitive limitations. These results call for further research into the neural mechanisms of these exploration patterns outside of any explore-exploit dilemma, and into their ecological and psychiatric validity. A general conclusion of our study is that humans display specific exploration patterns when they can freely acquire information that is not associated with immediate or prospective reward. Our findings thus suggest that restricting the study of exploration to explore-exploit dilemmas may have taught us more about how humans arbitrate this trade-off than about exploration itself[37].

## Methods

### Participants

For the first (discovery) dataset, we recruited $N = 30$ adult participants (15 females, mean age = 25 years, s.d. = 4.3 years). For the discovery dataset, we had no specific expectation regarding effect sizes: no statistical method was used to pre-determine sample size, and it was determined to match or exceed earlier studies of human exploration patterns in the literature. Participants were recruited through a local, open mailing list advertising psychology and neuroscience experiments in the Paris area. Participant compensation was 22 euros in cash for 90 min of testing. Recruitment criteria included age between 18 and 35 years, right-handedness, normal or correct-to-normal vision, normal colour perception, and no history of psychiatric or neurological disorder.

For the second (replication) dataset, we also recruited $N = 30$ adult participants (15 females, mean age = 25 years, s.d. = 3.7 years). Participants were recruited through the local mailing list used for the

first dataset, and through a national website (L'Etudiant) advertising short-term jobs for adult students. The other recruitment criteria were identical to those used for the first dataset. Participants recruited through the local mailing list are typically students (in the age range we targeted) and typically accustomed to psychology experiments. Participants recruited through L'Etudiant were also a majority of students, but they had never taken part in psychology studies. Participant compensation was 22 euros in cash for 90 min of testing.

For the third (three-condition) dataset, we balanced the lower number of sequences per condition by aiming to reach $N = 30$ adult participants after exclusion. We recruited a total of $N = 38$ participants, exclusively on the website l'Etudiant (21 females, mean age = 24 years, s.d. = 4.4 years). The other recruitment criteria were identical to those used for the first two datasets. Participant compensation was 20 euros in cash for 75 min of testing.

All participants gave written informed consent before taking part in the study. Ethics approval was obtained from the relevant authorities (Comité de Protection des Personnes Ile-de-France VI, ID RCB: 2007-A01125-48, 2017-A01778-45), which covers all three datasets. Informed consent was obtained for all three datasets.

We recruited participants to achieve a roughly equal number of self-reported males and females in each sample. However, we had no theoretically motivated reason to perform disaggregated sex or gender-based analyses. We did not collect consent from participants to share their individual self-reported sex data.

We excluded participants whose sampling strategy was stereotyped or did not depend on the presented outcomes. We computed the number of stereotyped sequences for each participant: sequences where the participant systematically alternated between left and right responses, or between the two options, and sequences where the participant sampled a single option throughout the whole sequence. We excluded participants who produced more than 50% of such stereotyped sequences in any condition ($N = 3$ participants in the discovery dataset, $N = 3$ participants in the replication dataset, and $N = 6$ participants in the third dataset). In the third experiment, one additional participant was sent home before taking part in the study due to COVID-19 symptoms. After exclusion, the final datasets comprised: 27 participants in the discovery dataset (12 females, mean age = 25, age standard deviation = 4), 27 participants in the replication dataset (13 females, mean age = 25, age standard deviation = 4), 31 participants in the third dataset (16 females, mean age = 24, age standard deviation = 4).

The experiment was within-participants: all participants in the discovery and replication datasets saw the two conditions, and all the participants of the third dataset saw all three conditions. There was no randomisation and no blinding involved.

### Experiment

**Experimental design.** Participants made sequences of choices between two options depicted by shapes, and each choice yielded an outcome. Outcomes were colour mixtures, drawn from a continuum between blue and orange (through grey in the middle of the range). The outcomes of each option were drawn from one of two probability distributions: an 'orange' distribution with 67% of dominantly orange outcomes (and 33% of dominantly blue outcomes), and a 'blue' distribution with 67% of dominantly blue outcomes (and 33% of dominantly orange outcomes)—Fig. 1b. The dominant colour associated with an option was independent of the dominant colour associated with the other option, such that participants needed to sample each individual option to learn its associated colour (Fig. 1c). Each sequence consisted of a series of choices (trials) whose number varied pseudo-randomly and uniformly across conditions from 8, 12, 16, or 20 choices (Fig. 1a.).

Each sequence was preceded by condition-specific instructions. In the MATCH condition, participants were asked to draw a maximum of

outcomes matching a target colour (orange or blue, counterbalanced across sequences). Participants were informed that the colour intensity of each outcome was translated into points: the more the outcome was of the target colour, the higher the number of points. This task feature rendered each outcome intrinsically rewarding. In this first condition, the sequence ended with the last choice: No final question was asked and participants immediately moved on to the instructions screen for the next sequence. In the GUESS condition, participants were asked to guess the dominant colour associated with each of the two options. Each sequence ended with a final question, where participants were asked about the dominant colour associated with one of the two options. In this second condition, participants were rewarded solely based on the accuracy of their responses to this final question. Each outcome was not intrinsically rewarding, but it was informative regarding the dominant colour associated with the sampled option. In the FIND condition, participants were asked to find an option associated with a target colour (orange or blue, counterbalanced across sequences). As in the GUESS condition, participants were rewarded solely based on the accuracy of their responses to a final question, which in this case asked them to select an option associated with the target colour at the end of the sequence. To further enhance the symmetry between the FIND and GUESS conditions in the third experiment, we changed the final question of the GUESS sequences to match that of the FIND sequences. Instead of asking the colour of a given option, we asked to identify an option of a given colour (orange or blue, counterbalanced across sequences). The single difference between the two conditions is that the question (i.e., the target colour to identify at the end of the sequence) was known in advance in the FIND condition, but unknown in the GUESS condition.

Eight geometric shapes were used to instantiate options, arranged into twenty-eight possible pairs. Shape-colour associations changed pseudo-randomly between sequences in order to minimise shape repetition. The colour intensities of outcomes were pre-defined in advance. Colour intensity indexes the amount of information provided by the outcome about its associated probability distribution: a more intense colour is more informative than a less intense one. This advanced pre-definition of colour intensities allowed us to make sure that, for each choice, the amount of information that could be acquired from the two options was exactly the same. Because the colour associated with an option could not be identified from the colour associated with the other option, different sequences used options associated either with the same or with different colours. This means that in the MATCH condition, some sequences had one single target option, whereas other sequences had either zero, or two target option(s). We included only sequences with a single target option when measuring performance. Each unique sequence was duplicated to be used once in each condition, so that each condition was tightly matched with the other(s) – except for the instructions provided to the participants. In particular, the uncertainty associated with each condition was therefore matched across conditions.

**Experimental procedures.** Participants were instructed on how to perform the task by a step-by-step, self-paced tutorial, followed by a practice session of eight sequences. Participants were then asked to describe the instructions to the experimenter, and it was emphasised again for all participants that: 1. the colour intensity of each outcome was relevant, 2. each option would not always draw the exact same shade, so the task was to identify its dominant colour, 3. there were only two possible fixed distributions (blue and orange), 4. the two options were not always associated with different colours (they could be both associated with orange, or blue), 5. each sequence would end after a variable, unpredictable number of choices, and 6. performance was measured by the colours of all sampled outcomes in the MATCH condition, and by the final question in the GUESS and FIND conditions.

For the discovery dataset, the experiment (excluding practice sequences) consisted of 48 sequences of each condition (96 in total, divided in 8 blocks of 12 sequences). For the replication dataset, the experiment consisted of 64 sequences of each condition (128 in total, divided in 8 blocks of 16 sequences). For the third dataset (with the additional FIND condition), the experiment consisted of 32 sequences of each condition (96 in total, divided in 4 blocks of 24 sequences).

Conditions were interleaved pseudo-randomly within blocks, with self-paced breaks between blocks. Conditions were pseudo-randomised to balance the number of sequences of each condition within each block, avoid too many (>3) consecutive repetitions of the same condition, and avoid immediate repetitions of the same shapes or the same response sides.

The task was run using the Psychtoolbox-3 toolbox (version 3.0.14[38]) for MATLAB 2017b (The Mathworks). The testing conditions were slightly different across datasets. For the first (discovery) dataset, eye position was recorded during the task using an EyeLink-1000 Plus eye-tracker system (SR Research), and participants' head movements were constrained by a chin rest. For the second (replication) and third datasets, eye position was not recorded and no chin rest was used. Participants were seated with their eyes at approximately 70 cm from a 24-in. LCD screen with a resolution of 1920 × 1080 pixels and a refresh rate of 60 Hz.

The trial structure is depicted in Fig. 1a. After reading the instructions and initiating a sequence by pressing the spacebar, each sampling decision began with the two shapes appearing on the left and right sides of a fixation circle - sides were pseudo-randomly assigned. This choice screen was displayed until participants selected an option by pressing the corresponding [A] (left shape) or [P] (right shape) key on an azerty keyboard, after which the outcome was displayed for 500 ms. The outcome corresponds to the filling of the selected shape with the sampled colour mixture. A fixation circle was then displayed alone for 500 ms before the trial began. In the MATCH condition, each sequence ended as soon as all the sampling choices were made (8, 12, 16, or 20) and their outcomes seen. The instructions for the following sequence then appeared on screen. In the GUESS (and FIND) conditions, each sequence ended with a final question screen. In the first two datasets, one shape (pre-selected at random) was displayed and participants were asked to report its associated (dominant) colour. In the third dataset, one colour (pseudo-randomly selected) was displayed on screen and participants were asked to identify an option associated to the demanded colour. Participants selected a colour or option with a key press, and a selection box was displayed around the selected colour or option for 400 ms. Feedback on the accuracy of this final response was provided by filling the selected response with its associated colour during 500 ms. The instructions for the following sequence then appeared on screen.

## Computational model

**Description of the model.** The computational model was updated during revisions: again, we first developed the model on the discovery dataset before testing it in the confirmatory samples. The computational model of sampling behaviour featured the same learning module across all three conditions. In practice, the colour distributions were designed such that the log-evidence for the categories ($x$) was proportional to the colour shades $c$ (ranging from −1 pure blue to +1 pure orange – see Fig. 1).

$$x_i = \log\left(\frac{p(c_i|\text{blue})}{p(c_i|\text{orange})}\right) \propto c_i, \text{ where } c_i = [-1:1] \qquad (1)$$

At each trial, the model could then accumulate evidence for each option ($\hat{x}$) by summing the colour outcomes $c$ observed from each

option up to the current trial:

$$\hat{x}_{option,\, t} = \sum_{1:t}^{i} c_i = \log\left(\frac{p(\text{blue}|c_1 \ldots c_t)}{p(\text{orange}|c_1 \ldots c_t)}\right) \quad (2)$$

The sum of evidence observed for this option $\hat{x}_{option,\, t}$, contained information about which colour distribution was more likely to have produced the observed samples ($sign(\hat{x}_{option,\, t})$), but also about how much information had been accumulated ($|\hat{x}_{option,t}|$). Based on these quantities, the decision variable $DV_t$ was computed at each trial $t$, as the sum of three components:

$$DV_t = \beta_{\text{tar}}*(\hat{x}_{1,\, t} - \hat{x}_{2,\, t})*c_{\text{tar}} + \beta_{\text{unc}}*(|\hat{x}_{2,\, t}| - |\hat{x}_{1,\, t}|) + b_{\text{rep}}*r_{t-1} \quad (3)$$

The first component was the difference in accumulated values between the two options, signed by the target colour: $(\hat{x}_{1,\, t} - \hat{x}_{2,\, t})*c_{\text{tar}}$, where $c_{\text{tar}}$ was participants' target colour (+1 for orange, –1 for blue, and 0 when no colour was announced at the beginning of the sequence). In practice, this first component was positive when the first option was closer to the target colour, and negative when the second option was closer to the target colour. This component was weighted by a sensitivity parameter $\beta_{\text{tar}}$, controlling the sensitivity of the model choices to the target colour. For sequences when participants played the guessing game, no target colour was given, so $c_{\text{tar}}$ was set to zero, and this first component was null. The second component was the difference in the amount of information accumulated for each option: $|\hat{x}_{2,\, t}| - |\hat{x}_{1,\, t}|$. This was also scaled by a sensitivity parameter $\beta_{\text{unc}}$, controlling the sensitivity of the model choices to the amount of information already acquired for each option. This component was defined in all conditions. The third component was also defined in all conditions for all choices except the first choice of each sequence, and corresponds to an additive bias in the direction of the previous choice, controlled by a repetition bias parameter $b_{\text{rep}}$. Its value was +1 when the previous choice ($r_{t-1}$) was option 1, and –1 when the previous choice was option 2, effectively biasing the decision variable towards choosing option 1. The choice process itself could take two different forms. By default, it was modelled by a standard *softmax* process on the decision variable $DV_t$. But the *ini+* model also included an initial sampling phase, where, as long as the level of evidence acquired for an option was below a given threshold ($\theta$), choices were not controlled by the decision variable described above, but the model repeated the previous decision with probability $\varepsilon$. This initial sampling phase of each option ($initialPhase_o$) was therefore controlled by the threshold $\theta$ on the amount of information already accumulated for the option, and by the threshold hardness $\varepsilon$, governing how much the model will repeat the previous decision while under the $\theta$ threshold. If the threshold hardness $\varepsilon$ is at 1, the model systematically repeats the previous choices as long as it is below the $\theta$ threshold, whereas it the hardness is at 0 the model does not implement the initial repeated sampling and always relies on the decision variable $DV_t$. Similarly, if the level of the threshold $\theta$ is at 0, the model also does not implement the initial sampling phase:

$$\text{if} \qquad initialPhase_o = \text{true} \,\&\, |\hat{x}_{o,t}| < \theta \qquad p(r_t = r_{t-1}) = \varepsilon \quad (4)$$

$$\text{else} \qquad\qquad\qquad\qquad p(r_t = r_{t-1}) = \frac{1}{1 + e^{-DV_t}} \quad (5)$$

$$\text{if} \qquad r_t \neq r_{t-1} \qquad\qquad initialPhase_o = \text{false} \quad (6)$$

To avoid numerical discontinuities in the likelihood function of the model due to the presence of a hard bound $\theta$, we assume that $\theta$ follows a normal distribution with a small standard deviation of 0.1 in logLR units (which is considered as small given that a single colour sample can provide up to 1.4 in logLR units).

For plotting and statistical testing, $\theta$ was transformed from logLR units into probabilities (and therefore treated as a bounded parameter). It can be understood as a certain probability (of the current shape being associated to either the blue or the orange distribution) that the agent wants to reach before terminating the initial repetitive sampling phase.

Note that the computational model described above nests the optimal sampling agent for the MATCH and GUESS conditions. The optimal agent does not include an initial sampling phase or a repetition bias in either condition. In the MATCH condition, the optimal sampler chooses based solely on the first component: the difference between the mean colour mixture estimates signed by the target colour, with $\beta_{\text{tar}} \to \infty$ (i.e. an argmax choice process). In the GUESS condition, the optimal sampler chooses based solely on the difference between the information already acquired from each option, with $\beta_{\text{unc}} \to \infty$. The optimal sampling agent was simulated for each participant and each condition to compare its behavioural measures to those of tested participants (Supplementary Fig. 2).

**Fitting procedures.** Best-fitting parameter estimates for the model, which can either include (ini+) or not include an initial sampling threshold (ini-), were obtained through maximum likelihood estimation (MLE) in two ways. First, we used Matlab's fmincon function to cover a wider range of possible parameter values and avoid local minima (the decision variable parameters were initialised at random points in the range [–2:2] for $\beta_{\text{tar}}$ and $\beta_{\text{unc}}$ and [–4:4] for $b_{\text{rep}}$, while the threshold value was set at a hundred different values in the interval [0.05:5]). Second, we used the best-fitting estimates obtained from this first fmincon pass as starting points for the Bayesian Adaptive Direct Search algorithm[39] (version 1.0.8). This second pass was two-fold: we used leave-one-out cross-validation on the sequences used to obtain model likelihood estimates. These were used in Bayesian model selection to compare variants of the model with vs. without the initial sampling phase. BMS was conducted using a random-effects approach, which assumes that different participants may rely on different models. It consists of estimating the distribution over models that participants are drawn from. We used the Dirichlet parameterisation of the random-effects approach implemented in SPM12 (http://www.fil.ion.ucl.ac.uk/spm - version 6906), with default (uniform) priors. We also used the BADS algorithm on all the sequences at once to obtain final parameter estimates[39].

**Knock-out simulations.** To test the relative effects of the two suboptimal aspects of the model (the initial sampling phase and the repetition bias) on behavioural measures in the different conditions, we used a 'knock-out' simulation procedure: starting from the 'full' model, separate model simulations were conducted after inactivating the initial sampling phase and the repetition bias independently of each other (by setting $\theta = 0$, or $b_{\text{rep}} = 0$, none, or both).

**Model recovery.** To assess whether our fitting procedure could distinguish in principle between the behaviour of an agent with and without initial sampling phase, we performed model recovery (Supplementary Fig. 5). In each condition, we simulated the behaviour of agents with or without an initial sampling phase, and used Bayesian model selection (BMS[40] (http://www.fil.ion.ucl.ac.uk/spm)) to measure which of the model with, or without initial sampling phase best fit the simulated data. During simulations, the general choice sensitivity parameters were set to the average values of the observed best-fitting parameters in each condition. For the initial sampling phase parameters, in the GUESS condition, they were either set to the average of best-fitting values (ini+ agent) or to zero (ini− agent). In the MATCH condition, they were set to either the average of best-fitting values of the GUESS condition (ini+ agent), or to zero (ini− agent). This is because best-fitting parameters in the MATCH condition do not reflect

the presence of an initial sampling phase. We then used the spm_BMS toolbox[40] (http://www.fil.ion.ucl.ac.uk/spm) to compute (a) the average frequency of both fit models for each simulated agent (confusion matrix), and (b) the probability of the simulated agents given the fit of each model (inversion matrix).

**Parameter recovery.** To evaluate our fitting procedure, we simulated the behaviour of artificial agents on each participant's true stimulus sequences, and fit those artificial choices using our initial fitting procedure (see Fitting Procedure, and Supplementary Fig. 5). We first permuted the values of each best-fitting parameter to remove correlations between parameters (<0.07 absolute Spearman's *r* between parameters) - this preserved parameter distributions while selectively removing between-parameters correlations. We then simulated choices for agents with initial sampling phases, under these decorrelated parameters (*simulated parameters*) and fit the model to the simulated choices using the same fitting procedure we had used for participants' choices (yielding the *recovered parameters*). The fitting procedure was evaluated within-parameter, based on the correlation between simulated and recovered values (Spearman correlations, weighted by the goodness of the original fit for each parameter value), and we used correlations between recovered parameters to assess whether artificial, between-parameter correlations were introduced during fitting (Spearman correlations, weighted by the goodness of the original fit for each parameter value). To confirm that the presence of an initial sampling phase in the model did not substantially bias the estimation of the later phase parameters (sensitivity to the target, sensitivity to uncertainty, and repetition bias), we also performed recoveries. We simulated choice behaviour for agents either with or without an initial sampling phase and recovered the parameters using a model *with* initial phase parameters (see Supplementary Fig. 6). We then assessed the bias of the model (the difference between recovered and simulated parameter values) for agents with or without an initial sampling phase.

**Principal component analysis.** To check that our model captured not only the mean behaviour of tested participants but also individual differences in the data, we ran a principal component analysis (PCA−Supplementary Fig. 11). We used choice similarity matrices as a summary snapshot of behaviour in the task, and we contrasted the behaviour of original participants with the behaviour of agents simulated under the best-fitting parameters of the participants ($n = 1000$ simulations). We derived the principal components (PCs) explaining the behaviour of the original participants on the one hand, and the simulated behaviour on the other hand.

**Stability of estimated parameters throughout the task.** To test for a change in exploration strategies throughout the task we compared the parameter estimates obtained from the first vs. the second half of the task, in the pooled datasets (Supplementary Fig. 12). The fitting procedure was the same as for the individual datasets (cf. above section 'Fitting Procedures').

**Statistical analyses**
Unless noted otherwise, statistical analyses of differences between unbounded metrics relied on two-sided parametric tests (single sample and paired *t*-tests), for bounded metrics we used two-sided non-parametric tests (Wilcoxon signed-rank test). Given our sample sizes, these statistical tests were applied outside the small-sample regime. Data were not explicitly tested for normality. *p* values were not corrected for multiple comparisons.

**Performance.** In all conditions, performance is reported in single-target sequences only, as computing performance in target-absent or both-target sequences was not interpretable. Because performance is typically not distributed normally across participants, we performed Wilcoxon signed-rank tests to compare participants' performance to chance level (50%), or between conditions. Effect sizes are reported as $r = \frac{z}{\sqrt{n}}$ with 95% confidence intervals. We used a cluster-level permutation-based analysis to identify time periods where the fraction of target-directed choices was significantly different from chance performance. We first identified clusters of contiguous trials where choice direction was significantly different from chance using a two-sided *p*-value of 0.05 at the trial-wise level (*t*-tests against 50%). We extracted the statistics of each identified cluster as its sum of *t*-values. We then computed 'null' distributions for the cluster statistic by extracting the highest absolute cluster statistic for simulated random choices ($N = 1000$ simulations). We then computed the empirical cluster-level *p*-value as the fraction of 'null' cluster statistics exceeding the observed cluster statistic.

**Fraction of re-sampling the same option.** We performed repeated-measures ANOVAs[41] on the fraction of re-sampling from the same option as a function of the number of previous samples acquired from this option. We tested the interaction between the condition (MATCH and GUESS in the discovery and replication datasets; MATCH, GUESS and FIND in the third dataset) and the number of previous samples acquired from the current option (from 1 to 10). Effect sizes are reported as $\eta_G^2 = \frac{SS_{effect}}{SS_{effect} + \sum SS_{residuals}}$, where $SS_{effect}$ corresponds to the sum of squares of the effect (interaction) of interest, and $SS_{residuals}$ is the aggregate sum of squares of the residual errors[42]. To identify for which numbers of acquired samples the fraction of resampling choices was significantly different between conditions, we used cluster-level permutation-based tests. First, we identified clusters of contiguous numbers of previous trials where the fraction resample was significantly different between conditions (*p*-value obtained from a paired-sample *t*-test <0.05). We extracted the statistics of each identified cluster as its sum of *t*-values. Then we computed the 'null' distribution for the cluster sum of *t*-value by shuffling the condition labels of individual sequences within participants. For each participant, we re-assigned the sequence choices to each condition randomly and recomputed the cluster statistic in the same way ($N = 1000$ permutations). We then computed the empirical cluster-level *p*-value as the fraction of 'null' cluster statistics exceeding the observed cluster statistic.

**Choice similarity.** At the choice level, choice similarity is defined as 1 if the participant chose the same option, and 0 if they chose a different option. Each choice in each sequence was compared to all the other choices in the same sequence (choice 1 was compared to choices 2:end, choice 2 to choices 3:end, etc), resulting in a similarity score for each pair of choice positions within each sequence. This was averaged across sequences, resulting in an average similarity score (between 0 and 1) for each position in the sequence. Because choice similarity is typically not distributed normally across participants, we performed Wilcoxon signed-rank tests to compare participants' choice similarity to chance level (50%).

**Choice direction.** We used a cluster-level permutation-based analysis to identify time periods where choices were significantly target- or uncertainty-directed. We first identified clusters of contiguous trials where choice direction was significantly different from chance using a two-sided *p*-value of 0.05 at the trial-wise level (*t*-tests against 50%). We extracted the statistic of each identified cluster as its sum of *t*-values. We then computed 'null' distributions for the cluster statistic by extracting the highest absolute cluster statistic for simulated random choices ($N = 1000$ simulations). We then computed the empirical cluster-level p-value as the fraction of 'null' cluster statistics exceeding the observed cluster statistic. We used a similar procedure to compare the fractions of uncertainty-directed choices between conditions. In

this case, we computed 'null' distributions by permuting randomly the condition labels of individual sequences and extracting cluster statistics for these permuted conditions.

**Model parameter estimates.** For the initial statistics in the confirmatory dataset, all tests were two-sided tests: $t$-tests against zero and paired $t$-tests between conditions for unbounded, sampling parameters $\beta_{tar}$ and $\beta_{unc}$, and for the repetition bias $b_{rep}$; and signed-rank tests between conditions for the bounded, initial sampling phase parameters $\theta$ and $\varepsilon$. Effect sizes are reported as Cohen's $d_z$ for $t$-tests, with 95% confidence intervals, and as $r = \frac{z}{\sqrt{n}}$ for Wilcoxon's tests, with 95% confidence intervals. For the third dataset, we did one-sided tests where we had a clear prediction of the direction of the effect. We tested unbounded sensitivity parameters $\beta_{tar}$ and $\beta_{unc}$ with one-sided $t$-tests against zero and paired between conditions. The presence of a repetition bias was assessed through one-sided $t$-tests against zero, but two-sided paired $t$-tests to compare conditions. Similarly, we used one-sided signed-rank tests to compare $\theta$ values between conditions but two-sided signed-rank tests to compare $\varepsilon$ values, as we had no directional prediction. The exact test used is reported either in the Results or on the corresponding Figure legend and in Supplementary Tables. Effect sizes are reported as Cohen's $d_z$ with 95% confidence intervals for $t$-tests, and as $r = \frac{z}{\sqrt{n}}$ with 95% confidence intervals for Wilcoxon's tests.

**Inter-individual analyses.** For this analysis, we split participants into groups of different sizes based on whether a model with or without the initial sampling phase best fit their behaviour in the GUESS condition. The groups were of unequal sizes. In this analysis, all tests were post-hoc and exploratory. We formally tested for equal variance between groups using the Levene test from Matlab. Differences in performance were tested with two-sided sum-rank tests (groups of unequal size and unequal variance), and effect sizes are reported as the probability of superiority along with bootstrapped 95% CIs. For differences in sensitivity to the target colour ($\beta_{tar}$) between the two groups (ini+ and ini− participants), no evidence was detected for a difference in variance between groups, and we compared the conditions with two-sided, two sample $t$-tests, effect size was computed as Cohen's $d$ along with bootstrapped 95% CI. To test for differences in sensitivity to uncertainty ($\beta_{unc}$) between conditions and between groups of participants (ini+ vs. ini−) we ran a mixed effects anova as implemented by Laurent Caplette[41] for Matlab. Effect sizes are reported as $\eta_G^2 = \frac{SS_{effect}}{SS_{effect} + \sum SS_{residuals}}$, where $SS_{effect}$ corresponds to the sum of squares of the effect of interest, and $SS_{residuals}$ is the aggregate sum of squares of the residual errors[42]. For post-hoc tests, the Levene test detected a difference in variance between groups and between conditions. We ran two-sided, two-sample $t$-tests for unequal variance with MATLAB's ttest2 function (which uses Welch's $t$-test and Satterthwaite approximation for degrees of freedom). Effect sizes were computed as Cohen's $d$ using the average of the variance of the two groups/conditions, along with bootstrapped 95% confidence intervals. We used the same approach for differences in repetition biases, except the Levene test did not detect evidence of a difference in variance, so we used the regular implementation of Matlab's ttest2 function for two-sided, two-sample $t$-tests. Effect sizes were computed as Cohen's $d$ using the pooled variance, along with bootstrapped 95% confidence intervals.

### Reporting summary
Further information on research design is available in the Nature Portfolio Reporting Summary linked to this article.

## Data availability
The data used in this study (raw anonymised behavioural data and summary tables) are available at https://gitlab.com/cle-a/colpub[43].

## Code availability
The code used to run the experiments described in the study, and the main analysis pipeline is available at https://gitlab.com/cle-a/colpub[43]. The main analysis pipeline has been uploaded prior to the collection of the replication dataset on a dedicated repository (https://gitlab.com/cle-a/colrep). The model fitting code has been updated during the analysis of the third dataset and during review, and the final version is available on the main repository.

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

## Acknowledgements

This work was supported by a starting grant from the European Research Council (ERC-StG- 759341) awarded to V.W. V.C. was supported by the Agence Nationale de la Recherche (ANR–16- CE37-0012-01 and ANR-19-CE37-0014-01). All authors benefited from an institutional grant from the Agence Nationale de la Recherche (ANR–17-EURE-0017).

## Author contributions

Valentin Wyart (V.W.) acquired funding and supervised the study. Clémence Almeras (C.A.), Valerian Chambon (V.C.) and V.W. designed the experiments. C.A. acquired the data. C.A. and V.W. designed the model. C.A. and V.W. analysed the data. C.A., V.C. and V.W. discussed the results. C.A. and V.W. drafted the manuscript. C.A., V.C. and V.W. reviewed and edited the manuscript.

## Competing interests

The authors declare no competing interests.
