## [Transparent Peer Review file · Nature Communications]

Competing cognitive pressures on human exploration in the absence of trade-off with exploitation

Corresponding Author: Dr Clemence Almeras

Version 0:

Reviewer comments:

Reviewer #1

(Remarks to the Author)

In this paper, Almeras and colleagues study human exploration strategies under different experimental conditions. When people are asked to solve a prediction problem (guessing the outcome or finding the option that produces a particular option), they repeatedly sample each option until they have sufficiently reduced their uncertainty ('local' information seeking). In contrast, when people are asked to solve a reward maximization problem (picking the rewarding option), they selectively sample the most uncertain option ('global' information seeking). These patterns are not captured by a 'rational' model. The authors develop a process model that gives an excellent fit to the data.

I have somewhat mixed feelings about this paper, though on the whole I like it. On the positive side, both the experimental and modeling parts are rigorous, the figures are beautiful, the topic is interesting and timely. I also commend the authors for replicating their results, which yields strong confidence in their results. On the negative side, I felt that the paper is overburdened by lots of analyses that provide partially redundant information (e.g., multiple metrics that illustrate the same choice pattern). The paper could be made more readable by condensing these results and perhaps moving some of them to the supplement. A more substantive issue is that I'm not sure how big a conceptual leap the results offer. Some of these findings (e.g., directed exploration of uncertain options) have been shown before. Repeated sampling of an option to reduce its uncertainty is, I believe, novel, though there are possibly other explanations for this kind of "streaky" behavior (see below).

Specific comments:

Information selection based on uncertainty reduction has been widely studied in cognitive psychology. So I was surprised that references to this literature were missing. See Nelson (2005) for a useful review. More recently, there has been relevant work by Bramley et al. (2015), Coenen et al. (2015), Meder et al. (2019), to name a few.

It is stated that the tendency to sample in streaks is suboptimal (p. 13). But as the authors acknowledge in the Discussion (p. 17), one determinant of streaks might be the avoidance of switch costs. If we take this into consideration, perhaps streaks aren't suboptimal after all? That is, people sample in streaks because switching is in some way costly.

Another argument for streaks might come from consideration of policy complexity, whereby perseveration arises from a limit on the number of bits available to encode a policy (Gershman, 2020).

Yet another perspective comes from studies of concept learning, which have documented a kind of "stickiness" in hypothesis testing behavior. Restle (1962) proposed that subjects sample one hypothesis at a time, only switching when they receive disconfirming evidence (see also Bower & Trabasso, 1963). A more recent stochastic version of this idea was introduced by Bonawitz et al. (2014).

Minor comments:

Line 160: I think this should be a reference to Supplementary Figure 2?

There appears to be a mismatch between the panel labels and the caption for Figure 6.

Were subjects told about the relationship between color intensity and informativeness?

Reviewer #2

(Remarks to the Author)

Review of Competing cognitive pressures on human exploration in the absence of trade-off with exploitation

Summary

This paper presents three studies on human exploration in a two-option task. Participants are presented repeatedly with two options, and choosing an option provides an outcome on a continuous colour scale from blue to orange. Different conditions set different goals: In the DRAW condition, participants were instructed to obtain outcomes closest to one of the colours (this is like a standard two-armed bandit task). In the GUESS condition, they could sample freely from the options, and then were asked to determine the dominant colour for each of the options (Exp 1 and 2) or to pick an option with a specified dominant colour (Exp 3). The FIND condition was like the GUESS condition in Exp 3, but participants were informed beforehand of the colour their final choice should correspond to. Results show that participants allocate their choices differently to the options in the different conditions. In the DRAW condition, participants gradually choose more often the option which delivers more of the rewarding colour. In the GUESS condition, choice patterns show “streaks” where participants choose one option for some time (regardless of outcome) and then the other. The FIND condition appears to be a mix of these two. These patterns are eluded in a variety of analyses, including a computational analysis.

Evaluation

Whilst I think there are interesting aspects to the studies, and the findings show potentially interesting patterns, I don't believe the paper in its present form is ready for publication. The key finding about initial streaks in choices in the GUESS/FIND conditions may be of interest to select readers, but needs a more convincing explanation than provided. It also does not warrant a paper of this length. I will detail my major concerns are detailed below.

Major issues

1. The framing of the paper seems a little off to me. The main aim is to study exploration with and without an exploitation trade-off. The FIND condition seems a more appropriate comparison to the DRAW condition for this than the GUESS condition, as the latter changes the goal of information seeking. In the DRAW and FIND conditions, the aim is to find the option which dominates in producing more outcomes of a given colour. In the GUESS condition, the aim is to learn the dominant colour of each options. That choice patterns differ when given a different goal is not that striking. One goal involves a direct comparison between options (which option offers more of something), whilst the other option does not (need to learn characteristics of each option). Differences between the DRAW and FIND condition are more informative about the influence of the explore-exploit tradeoff on exploration. So, to me, the key conditions are DRAW and FIND, and the GUESS condition is more of an add-on. But the paper focuses mostly on the DRAW and GUESS conditions, and then introduces the FIND condition as an additional check. I suggest focusing on the third experiment, and then perhaps showing that the results of two conditions replicate in two other experiments. This could offer a more focused and shorter paper (see also below).
2. The main finding is that in the GUESS/FIND conditions, participants show initial “streaks” in choices, repeatedly sampling the same option in a row. The explanation for this is vague, and also not clearly addressed in their model, which just includes this as a component without being able to base this on e.g. aspects of experience or the goal. The explanation provided (“a ‘local’ form of

information seeking that aims at generating and testing a hypothesis about the currently sampled option” which contrasts to a “global” form of information seeking through directed sampling of the most uncertain option) may be true, but is entirely speculative. Even the model provided does not include any notion of hypothesis testing,

3. The “process model”, as I see it, is not really a process model, but more of a data-analytic model. No reason is provided for turning the the “initial sampling” component “on” or “off”. A data-analytic model is fine, and it succeeds in highlighting the differences in search patterns. But then there are claims that certain parameter settings for this model correspond to rational models, yet it is entirely unclear what criterion of rationality is alluded to. If I understand the task correctly, the purported “rational models” would fail at the task completely! The model includes mean colour estimates for each option, as well as what I presume are the absolute values of these estimates. For the “rational” DRAW model, choices are based on the difference in mean colour estimates between the options, i.e. $P(\text{choose } 1) = F(\beta_{\text{(tar)}} \times (\hat{x}_{\text{(t, 1)}} - \hat{x}_{\text{(t, 2)}}))$, with $F(x) = 1/(1 + \exp(-x))$. This is a simple model for a two-armed bandit, but by no means optimal. Given the short horizon, an optimal model might be possible through defining the task as a POMDP (accounting also for the variability in horizons). So the model is not optimal for the task, which is usually what is meant by a rational model. Furthermore, the authors claim that the rational model would set $\beta_{\text{(tar)}} \rightarrow \infty$. This implies a pure exploit strategy, without exploration. This model always chooses option 1 if $\hat{x}_{\text{(t, 1)}} - \hat{x}_{\text{(t, 2)}} > 0$ and option 2 if $\hat{x}_{\text{(t, 1)}} - \hat{x}_{\text{(t, 2)}} < 0$. If both options have a positive mean, but option 2 a higher mean, then it could be the case that after sampling option 1, option 2 is never sampled, so the model would not find the best option. In the present experiment, there are only two possible means for each option, so this is not a practical concern, but were participants aware of this design constraint? For the GUESS condition, the “rational” model bases its choices (if I understand this correctly, the information provided is unclear) on $P(\text{choose } 1) = F(\beta_{\text{(inf)}} \times (|\hat{x}_{\text{(t, 2)}}| - |\hat{x}_{\text{(t, 1)}}|))$. I don’t follow how the absolute mean value estimates encode “information already acquired”. It would make more sense to me to relate this to the number of times an option has been sampled, or a value encoding the uncertainty associated with the value estimate (which will also depend on the number of samples). Again, the “rational model” sets $\beta_{\text{(inf)}} \rightarrow \infty$, which, if $|\hat{x}_{\text{(t, 2)}}| - |\hat{x}_{\text{(t, 1)}}|$ encodes information obtained, might make sense here. But to what extent is this rational? Does following these choice strategy maximise the probability of getting the dominant colour for both options right? The model should be specified in much more detail (I had to guess certain parts) and it should be considered how this model provides optimal (or at least very good) performance on the tasks.

4. Whilst the highly detailed analyses have merits, I struggled to “see the forest for the trees”. A much more focused and shorter paper with key analyses presented in the main paper and supporting ones in supplementary material, would work better, I believe. For choice patterns, we get a range of analyses (sampling imbalance, option re-sampling, choice-choice similarity) which all indicate the same thing (early streaks vs gradual increase to preferred option). I suggest focusing on one of these and including other analyses as supplementary material. I also suggest to mainly focus on experiment 3, and report the other two as replicating effects.

5. There is a large literature on contrasting “Decisions from description” to “Decisions from experience”. The latter is either operationalised as a standard multi-armed bandit, or via a “sampling paradigm” where participants can freely sample from the options, until making a final (rewarded) choice. These two operationalisations have also been contrasted (see e.g. Wulf et al 2017, “A Meta-Analytic Review of Two Modes of Learning and the Description–Experience Gap”). This literature is highly relevant but has been (almost) completely ignored in this paper.

6. The paper is unclear/does not provide crucial information in various places. Key aspects (such as that this was a within-subjects

design), should be reported in the introduction or results (this was only evident later when it turned out to be possible to study covariation over conditions). Crucial parts of the design (e.g., is the order of conditions randomised or counterbalanced, and if so how?) are not detailed. The model is described in vague terms, rather than a clear formula of its workings and how parameters are initialized. It also seems that the GUESS condition differs between Exp3 and Exp 1&2 (i the last experiment, participants are asked which of two options is blue or orange, whilst in the first two they are asked about the dominant colour of one of the options), but this is not clearly stated. How is choice similarity defined? In Fig 7. In the top plots, it is unclear that subplots compare DRAW to GUESS. Etc.

Version 1:

Reviewer comments:

Reviewer #1

(Remarks to the Author)

I'm satisfied with the revision. One small typo in the abstract: "patterns arising in when" -> "patterns arising when"

(Remarks on code availability)

Reviewer #2

(Remarks to the Author)

The authors did a good job responding to the issues raised in the reviews of the previous paper. The presentation is more focused, and the modelling has improved. That said, I still have doubts about the theoretical significance of the findings (initial streaks in the GUESS condition), due to the different goals between the conditions. The focus should be on comparing the two options in the DRAW and FIND conditions (i.e. which option has more of the target colour), but in the GUESS condition, the focus needs to be on identifying the nature of each option, which can be done in isolation and does not require contrasting the options. To determine the nature of a single option, "streaky" exploration may be subject to less memory demands than more evenly distributed exploration. A heuristic for the other conditions is to pitch the options against each other, for instance sampling one and then the other and just determining which one had the most of the target colour in each consecutive pairwise comparison. This seems quite reasonable to me and would account for the results without many of the processes relied upon in the discussion. That said, the finding of a difference in exploration behaviour is supported well by the analyses and the "streaky" exploration is suboptimal when the length of a sequence is unknown and a trial can end at any time. But I do think the discussion can be shortened and streamlined further.

One major issue I have with the current version relates to the "Positive covariations between exploration patterns across individuals" section, where participants are split into two groups based on whether the model with initial "streaky" sampling fits better than a model without. Part of the analysis here focuses on the remaining parameters of the model. If I understand the model correctly, then – because the initial choices are governed strongly by the initial sampling policy in the INI+ group, the estimates of the remaining parameters will be mostly determined by later choices (after the initial sampling phase is over). In the INI- group, the remaining parameters likely cover the whole sequence (or at least most of the sequence). Results such as that the sensitivity to the reward maximising option is higher in the INI+ group than the INI- group are then not overly surprising, as one would expect that during the later trials participants would have determined the best option. So, these comparisons are biased due to the parameters in the INI+ group being optimized for the later parts of the sequences, whilst this is less the case for the INI- group. In my view, then, comparisons between the INI+ and INI- group, in terms of model parameters, are not overly meaningful. If such a comparison is of interest, I suggest to only focus on the behavioural results (e.g. proportion of maximising choices, repeat choices, etc.)

A more minor issue is that I think the Horizon task is misrepresented on page 9: "This new condition was akin to browsing restaurant reviews: choices yielded no immediate reward, but outcomes were still valued with respect to an end goal. It was also closer to existing delayed-reward tasks, like the Horizon task9 or in the "sampling paradigm" from Decisions From Experience studies12. In these paradigms, rewards are only granted after a sequence of so-called "free choices" (as in the GUESS condition), but (in contrast to the GUESS condition), choice outcomes are numerical values, and therefore directly inform the participants of the option's ultimate rewarding value." This seems to imply delayed rewards for free choices in the Horizon task, but - as far as I'm aware - rewards are not delayed in the Horizon task, beyond an initial sequence of forced choices which manipulate the amount of information gathered. So I think this is a misrepresentation of the Horizon task.

(Remarks on code availability)

I do not have access to Matlab, on which the code relies. I did have a look and the repository exists. But the contents are from 2 years ago and as the modelling has changed since the last manuscript, I think it needs updating.

Version 2:

Reviewer comments:

Reviewer #3

(Remarks to the Author)

The current manuscript investigates information search patterns in three conditions, divorcing information search from choice outcomes to different extents. Across three experiments, evidence is presented that suggests different information search patterns, with conditions with delayed or fully suspended rewards producing behavior directed more strongly at reducing uncertainty. To account for this, the authors suggest a computational model. Overall, the presented results are interesting and suitable for Nature Communications. I am inclined to recommend publication, but I believe the authors should first address the following comments, which may help clarify the exposition and interpretation of their findings. This can be achieved without conducting additional analyses or studies.

The simplicity of the behavioral paradigm is intriguing but also limiting. It does not afford the study of the full complexity of information search situations, which includes prior expectations and types of uncertainty. In our recent work forthcoming in PNAS (Spektor & Wulff, 2024), we attempted to reveal additional details of the full complexity of human information search by means of analyzing a large database of the sampling paradigm. This revealed that people enter the task with ecological (structural) knowledge, which they update across the task and use to steer their information search behavior. It further revealed that individuals reduce multiple types of uncertainties and differ not only in the details of a common computational strategy but likely recruit entirely different strategies. Finally, it also showed evidence of strategy shifts during the task, which may be related to the local-global strategy shift described here. Not all aspects can be studied with the current paradigm, which is fine. However, I think it is important to emphasize more strongly and discuss in greater length the limitations of both the paradigm and the computational account with respect to capturing the full complexity of human information search behavior.

I believe the distinction between local and global search is interesting. However, I am not sure it makes sense to map it strictly to the reliance on uncertainty minimization. Local search also results in the acquisition of information and thus can be used to reduce one or the other type of uncertainty. Relatedly, it is important to distinguish between exploration and information search. Moreover, I am unsure on what basis the authors infer that participants engage in hypothesis testing during local search. I recommend that the authors clarify the relationship between local and global search and uncertainty minimization, as well as potential evidence of hypothesis testing.

I believe the authors should discuss the relatively small sample sizes of their studies and potential limitations in terms of representativeness.

Concerning the responses to past reviewers, which I was asked to evaluate, my impression is that the authors place too much trust in their computational model. It is unclear whether it represents a sufficiently accurate and unbiased account of the empirical data to draw such strong inferences. Even if recovery is demonstrated using simulations, it is always wise not to place too much trust in any computational model as the data-generating process could be entirely different. In fact, given the research highlighted in my review, this is to be expected. To address this, I recommend that the authors present a separate descriptive analysis of switching behavior and individual differences to investigate the hypothesis expressed in the first comment of reviewer two and demonstrate the existence of individual differences questioned in the second comment of reviewer two.

Finally, I would appreciate it if the authors could clarify aspects of their model. If I understand correctly, the authors model whether a sampling decision is repeated as a function of the decision variable DV, independent of which option was sampled last. Consequently, I would expect that the probability of a repeat should be based on the absolute differences in the amount and direction of evidence for the two options rather than their signed differences (i.e., $\hat{x}_{1,t} - \hat{x}_{2,t}$ and $|\hat{x}_{1,t}| - |\hat{x}_{2,t}|$), but the definition of DV suggests the opposite. The result is that only differences in one direction (e.g., $\hat{x}_{1,t} > \hat{x}_{2,t}$) but not $\hat{x}_{1,t} < \hat{x}_{2,t}$) can trigger a switch to the other option. I don't understand how this makes sense.

(Remarks on code availability)

Version 3:

Reviewer comments:

Reviewer #4

(Remarks to the Author)

Having reviewed R3's comments and the authors' responses to them, I believe that the authors have done a great job addressing all the comments thoroughly and convincingly, substantially improving the conceptual clarity of the paper and adding helpful model-independent analyses.

(Remarks on code availability)

We thank both reviewers for their helpful comments. We have done our best to address the different suggestions of the reviewers, we hope they agree that this has improved the manuscript. Overall, we have condensed the presentation of the results, which we hope has increased the clarity of the paper. Thanks to the suggestions reviewer #2, we have changed the process model we use: evidence accumulation now directly reflects the optimal strategy, and the initial sampling phase no longer depends on a *number* of samples but rather on the *level of evidence* acquired about the current option. We address the specific comments of the reviewers below; our responses are in green and we highlighted quotes from the revised manuscript in blue.

Reviewer #1 (Remarks to the Author):

In this paper, Almeras and colleagues study human exploration strategies under different experimental conditions. When people are asked to solve a prediction problem (guessing the outcome or finding the option that produces a particular option), they repeatedly sample each option until they have sufficiently reduced their uncertainty ('local' information seeking). In contrast, when people are asked to solve a reward maximization problem (picking the rewarding option), they selectively sample the most uncertain option ('global' information seeking). These patterns are not captured by a 'rational' model. The authors develop a process model that gives an excellent fit to the data.

I have somewhat mixed feelings about this paper, though on the whole I like it. On the positive side, both the experimental and modeling parts are rigorous, the figures are beautiful, the topic is interesting and timely. I also commend the authors for replicating their results, which yields strong confidence in their results. On the negative side, I felt that the paper is overburdened by lots of analyses that provide partially redundant information (e.g., multiple metrics that illustrate the same choice pattern). The paper could be made more readable by condensing these results and perhaps moving some of them to the supplement. A more substantive issue is that I'm not sure how big a conceptual leap the results offer. Some of these findings (e.g., directed exploration of uncertain options) have been shown before. Repeated sampling of an option to reduce its uncertainty is, I believe, novel, though there are possibly other explanations for this kind of "streaky" behavior (see below).

We thank the reviewer for their careful assessment of our manuscript, their helpful comments, and literature suggestions. We have trimmed the presentation of the results and we hope this helps highlight the specific contribution of the paper. We answered specific comments below.

Specific comments:

Information selection based on uncertainty reduction has been widely studied in cognitive psychology. So I was surprised that references to this literature were missing. See Nelson (2005) for a useful review. More recently, there has been relevant work by Bramley et al. (2015), Coenen et al. (2015), Meder et al. (2019), to name a few.

We thank the reviewer for pointing out these relevant references: we agree that sampling patterns aiming to reduce uncertainty have been documented in other tasks. We believe that the contribution of our manuscript is to propose very comparable conditions, where sampling patterns can be directly contrasted under different constraints/goals. This contrast allows us to show how the immediate goal of accumulating rewards (in the canonical studies of exploration in bandit tasks) distorts sampling patterns. We have now included these references to the discussion of directed exploration:

Discussion page 15: *"Targeted sampling based on uncertainty reduction has been documented in various information gathering tasks¹³⁻¹⁶. However, when rewards are involved, previous work has shown that such*

targeted selection of uncertain options critically depends on lengthening the temporal ‘horizon’ of the task (i.e. delaying rewards): such targeted sampling is often described as ‘directed’ exploration in contrast to ‘random’ exploration^{9,10}. Consistently, in our dataset, directed exploration was evident whenever rewards were delayed: when participants were rewarded for their response at the final question. In addition, because the paradigm allowed to contrast different delayed rewards situations, we observed that directed exploration appeared to only be sensitive to delaying rewards, and not to whether outcomes were immediately informative about the ultimate rewarding value of options (i.e., it reached comparable levels in the GUESS and FIND conditions).”

It is stated that the tendency to sample in streaks is suboptimal (p. 13). But as the authors acknowledge in the Discussion (p. 17), one determinant of streaks might be the avoidance of switch costs. If we take this into consideration, perhaps streaks aren't suboptimal after all? That is, people sample in streaks because switching is in some way costly.

Thank you for helping us realise this wasn't clear. Repetitive sampling is *statistically* suboptimal (i.e. it is not the optimal strategy): in the GUESS condition, the best strategy is to always sample from the option for which the least information has been accumulated, as this will maximise information gain. We have made this clearer on Figure 3 and Supplementary Figure 2, as well as in the description of the model page 6: *“Whereas in the GUESS condition, the optimal strategy is to always chose the option with lowest absolute accumulated value (the most uncertain option – Figure 3.d., Supplementary Figure 2.b.). This allows the agent to maximise information about both shapes at any trial – thus maximising its probability of responding correctly to the final question, whose timing was not disclosed to participants.”*

The strategy of the participants (repeatedly sampling the same option) is thus suboptimal, and will produce *objectively* degraded evidence. We propose that this strategy results from cognitive limitations: because switching is costly, participants preferred to sample in streak. Yet, we observed that the participants who adopted such repeated sampling strategies obtained higher performance. This suggests that repetitive sampling could be cognitively more efficient: that observing evidence in streaks makes accumulation by human brains easier. We speculate that reducing switch costs, for instance, may have allowed participants to make better use of the evidence they sampled, however statistically degraded. In other words, even though sampling streaks are *statistically suboptimal*, they may be *cognitively optimal*: they overcome cognitive limitations (the switch costs) to produce more efficient evidence and thus better performance. It is unclear at this point whether this works by improving the (e.g. sensory) quality of the evidence sampled (akin to an attentional effect), or by reducing the probability of forgetting all or part of the information about the option during a switch (a memory retrieval effect), or through any other mechanism.

We thank the reviewer for helping us clarify this point, we have amended the discussion to highlight the difference between *statistical* and *cognitive* suboptimality.

Discussion page 17: *“[...] Participants who displayed patterned sampling had higher repetition biases, but were also more accurate at identifying the overall structure of each sequence (i.e. the option outcome associations), and were more sensitive to task-relevant decision variables, even in conditions where they did not display the patterned sampling (in the DRAW condition, where sequences required to maximise immediate reward). These results are reminiscent of work from Hills and colleagues⁵, using the sampling paradigm in Decisions From Experience tasks (where participants who sampled for longer also produced longer streaks for each option), and of work from Abir and colleagues³⁸, using a two-step task (in which participants who obtained higher performance also had higher repetition bias and more strategic sensitivity to uncertainty). The positive relationship we document between initial repetitive sampling (a suboptimal strategy in normative terms) and task performance is important, because it suggests that patterned sampling reflects an active cognitive strategy, rather than mere task disengagement or inattention. **It suggests that statistically suboptimal strategies could be more efficient for human subjects, which operate under specific biological constraints. Even though they stem from cognitive limitations in the first place, strategies which produce objectively degraded evidence could be more cognitively optimal, under the specific biological constraints of the human brain (see ^{18,39,40} for comparable***

findings).

The interpretation of this positive relation between initial repeated sampling and task performance raises two questions. First, the question of the mechanisms through which repetitive sampling leads to higher performance despite degraded evidence. Repetitive sampling reduces switch costs^{24,25} and could thus alleviate the cognitive effort required; but it is unclear at this point whether this improves the (e.g. sensory) quality of the evidence sampled (akin to an attentional effect), or reduces the probability of forgetting all or part of the information about the option during a switch (a memory retrieval effect), or facilitates processing through any other mechanism.”

Another argument for streaks might come from consideration of policy complexity, whereby perseveration arises from a limit on the number of bits available to encode a policy (Gershman, 2020).

We thank the reviewer for pointing out this relevant reference which shows that policy perseveration can arise from memory limitations leading to policy simplification/compression. In our case, participants were only ever choosing between two options, which limits the complexity of their potential policies. For this reason, we don't expect complexity reduction to be the prime pressure shaping participants' choices. In addition, in our specific case of *initial* repetitions, and considering the revised version of the model wherein repeating initial decisions depends on whether the level of accumulated evidence for the current option has surpassed a given information threshold, we propose that streaking is not mere perseveration, and actually makes the policy *more* complex. Repetition here is not merely motor, as the options' symbols could switch sides at any time – repeating therefore required active executive control. But repetition of a symbol is also not blind: it requires the agent to compare the current level of evidence to the threshold (parameter θ in the model), and to decide whether to repeat or not, based on this comparison. In these terms, we believe that initial streaking is a more complex policy than simply always choosing the most uncertain option (the optimal strategy). While streaking seems behaviourally simple, and while it possibly stems from cognitive limitations (the costs of switching), it does not necessarily manifest in simpler policies. While we argue that streaking might lead to more efficient, perhaps easier, evidence accumulation, it is indeed a more complex sampling strategy.

We have added a paragraph to the discussion to address this comment pages 16-17: *“Similarly, the evidence-dependent repetitive sampling in the beginning of sequences is not reducible to mere perseveration. While previous studies have shown that policy perseveration could arise from memory limitations leading to policy simplification^{33,34}, the initial repetitions we describe are hardly a simplification of the policy. Repetitions here are not blind: participants only repeat initial choices as long as the level of evidence acquired on the preceding choices is below a certain threshold. In other words, this conditional policy requires to compare the current level of evidence for the option to the threshold level (parameter θ in the model), and based on this comparison, to decide whether to repeat or to compute the decision variable. As such, initial repetitions are a more complex policy than simply always computing the decision variable. Incidentally, in this task, participants are only choosing between two options: This restrains the complexity of potential policies, and makes it unlikely that policy compression be the prime driver of the smart repetition strategies we observed.”*

Yet another perspective comes from studies of concept learning, which have documented a kind of "stickiness" in hypothesis testing behavior. Restle (1962) proposed that subjects sample one hypothesis at a time, only switching when they receive disconfirming evidence (see also Bower & Trabasso, 1963). A more recent stochastic version of this idea was introduced by Bonawitz et al. (2014).

Again, we thank the reviewer for suggesting interesting references. In the tasks mentioned by the reviewer, participants are presented with stimuli, and the task is to correctly categorise them. In these tasks, adults and children are shown to stick to their current hypothesis (about category boundaries compatible with their

observations), as long as they don't receive critically disconfirmatory information. In our task as well, participants had to categorise each bag as either orange or blue. But, in contrast to the tasks mentioned above, our participants also had to *actively select* which option to sample from: the repetitive sampling patterns we describe don't refer to participants' sampling into the *hypothesis space*, but between the two options. It is thus very well possible that participants stuck to a hypothesis ("this bag is blue / orange") unless critically disconfirmed (like the participants of Restle, Bower and Trabasso, and Bonawitz and colleagues), but this alone doesn't explain why they should want to re-sample from the same *option* - especially since they are incentivised to maximise information about both options. We could speculate that they repeated their choice until they received sufficiently *disconfirmatory* information. But the updated version of the model we propose suggests, on the contrary, that participants actually initially switched options only when their category hypothesis about the current option had been *sufficiently confirmed* (i.e. when they reached a specific level of evidence in favour of the current hypothesis). Note that the two possibilities are not mutually exclusive: it is possible that participants only switched *hypotheses* about an option when sufficiently disconfirmed, but our results suggest that they also wanted to resample the same *option* until they were x% confident in their hypothesis.

We have added a discussion of this point to the discussion on page 16: *"It is important to distinguish between the sampling of options and the sampling of the hypotheses. Previous studies of categorisation or concept learning have shown that when participants were gradually presented with evidence, they didn't tend to resample the best-fitting hypothesis with each piece of new incoming evidence, but rather stuck to the same hypothesis until it was critically disconfirmed (e.g. ²⁹⁻³¹). Our results pertain to a different level of description: stickiness about the hypotheses doesn't explain why participants would want to resample the same option. It is very well possible that our participants were also conservative about hypotheses and only changed their minds in the face of sufficiently contradictory evidence, but these results also suggest that they wanted to resample the same option until they were sufficiently convinced of their current hypothesis – i.e. until they reached a certain level of evidence in favour of said hypothesis."*

Minor comments:

Line 160: I think this should be a reference to Supplementary Figure 2?

There appears to be a mismatch between the panel labels and the caption for Figure 6.

Thank you for spotting this, we have fixed these imprecisions.

Were subjects told about the relationship between color intensity and informativeness?

Yes, indeed. We apologise for the ambiguity; we have amended the Methods to make this clearer, thank you. Page 20: *"Participants were instructed on how to perform the task by a step-by-step, self-paced tutorial, followed by a practice session of eight sequences. Participants were then asked to describe the instructions to the experimenter, and it was emphasised again for all participants that: 1. **the colour intensity of each outcome was relevant**, 2. each option would not always draw the exact same shade, so the task was to identify its dominant colour, 3. there were only two possible fixed distributions (blue and orange), 4. the two options were not always associated with different colours (they could be both associated with orange, or blue), 5. each sequence would end after a variable, unpredictable number of choices, and 6. performance was measured by the colours of all sampled outcomes in the DRAW condition, and by the final question in the GUESS and FIND conditions."*

We are grateful to reviewer #1 for their valuable challenges and suggestions. We hope that our responses can convince the reviewer that this set of experiments documents genuinely novel contrasts in sampling patterns, which could not be evidenced in existing tasks and cannot be directly explained by existing data.

Reviewer #2 (Remarks to the Author):

Review of Competing cognitive pressures on human exploration in the absence of trade-off with exploitation

Summary

This paper presents three studies on human exploration in a two-option task. Participants are presented repeatedly with two options, and choosing an option provides an outcome on a continuous colour scale from blue to orange. Different conditions set different goals: In the DRAW condition, participants were instructed to obtain outcomes closest to one of the colours (this is like a standard two-armed bandit task). In the GUESS condition, they could sample freely from the options, and then were asked to determine the dominant colour for each of the options (Exp 1 and 2) or to pick an option with a specified dominant colour (Exp 3). The FIND condition was like the GUESS condition in Exp 3, but participants were informed beforehand of the colour their final choice should correspond to. Results show that participants allocate their choices differently to the options in the different conditions. In the DRAW condition, participants gradually choose more often the option which delivers more of the rewarding colour. In the GUESS condition, choice patterns show “streaks” where participants choose one option for some time (regardless of outcome) and then the other. The FIND condition appears to be a mix of these two. These patterns are eluded in a variety of analyses, including a computational analysis.

Evaluation

Whilst I think there are interesting aspects to the studies, and the findings show potentially interesting patterns, I don't believe the paper in its present form is ready for publication. The key finding about initial streaks in choices in the GUESS/FIND conditions may be of interest to select readers, but needs a more convincing explanation than provided. It also does not warrant a paper of this length. I will detail my major concerns are detailed below.

We thank the reviewer for taking the time to evaluate our paper, and for raising several points which allowed us to improve the computational analyses and presentation of the results. We hope this revised paper will convince the reviewer of the strength of our findings and their relevance in light of the existing literature on human exploration patterns.

Major issues

1. The framing of the paper seems a little off to me. The main aim is to study exploration with and without an exploitation trade-off. The FIND condition seems a more appropriate comparison to the DRAW condition for this than the GUESS condition, as the latter changes the goal of information seeking. In the DRAW and FIND conditions, the aim is to find the option which dominates in producing more outcomes of a given colour. In the GUESS condition, the aim is to learn the dominant colour of each options. That choice patterns differ when given a different goal is not that striking. One goal involves a direct comparison between options (which option offers more of something), whilst the other option does not (need to learn characteristics of each option). Differences between the DRAW and FIND condition are more informative about the influence of the explore-exploit trade-off on exploration. So, to me, the key conditions are DRAW and FIND, and the GUESS condition is more of an add-on. But the paper focuses mostly on the DRAW and GUESS

conditions, and then introduces the FIND condition as an additional check. I suggest focusing on the third experiment, and then perhaps showing that the results of two conditions replicate in two other experiments. This could offer a more focused and shorter paper (see also below).

We would like to thank the reviewer for giving us the opportunity to clarify the framing of the paper. We agree that the FIND condition makes our results more directly comparable to the existing literature, and that this condition was necessary to disentangle the effects of delaying rewards from those of the absence of a target colour in the GUESS condition. The FIND condition relates more directly to existing delayed-rewards task: notably, the Horizon task (e.g. Wilson2014) and the Sampling Paradigm of Decisions From Experience tasks (e.g. Hills and Hertwig 2010, Wulff et al. 2015). In both these cases, participants are allowed to sample freely during a certain period before they can make an incentivised choice (either at a forced time in the Horizon task or self-paced in the DFE tasks). These tasks have shown that delaying rewards allows directed exploration (horizon task), and that participants who explore longer also alternate less, effectively exploring in streaks (Hills & Hertwig 2010).

We have included these references in the introduction of the third dataset page 9: *“This new condition was akin to browsing restaurant reviews: choices yielded no immediate reward, but outcomes were still valued with respect to an end goal. It was also closer to existing delayed-reward tasks, like the Horizon task⁹ or in the “sampling paradigm” from Decisions From Experience studies¹². In these paradigms, rewards are only granted after a sequence of so-called “free choices” (as in the GUESS condition), but (in contrast to the GUESS condition), choice outcomes are numerical values, and therefore directly inform the participants of the option’s future rewarding value. In these tasks, it is thus possible that a future reward value already taints the outcomes and biases choice patterns. While the original GUESS condition was necessary to eliminates this concern, the addition of the FIND condition eliminate a gap in the paradigm. Here, the contrast between the FIND and DRAW conditions reflects the difference between delayed and immediate exploitation, whereas the contrast between the FIND and GUESS conditions reflects the difference between the sampling of outcomes with and without rewarding value.”*

We have also related Hills and Hertwig’s key observations to our results in the discussion page 17: *“These results are reminiscent of work from Hills and colleagues⁵, using the sampling paradigm in Decisions From Experience tasks (where participants who sampled for longer also produced longer streaks for each option), and of work from Abir and colleagues³⁸, using a two-step task (in which participants who obtained higher performance also had higher repetition bias and more strategic sensitivity to uncertainty).”*

Nevertheless, in all these tasks, even though outcomes don’t directly contribute to participants’ earnings, they are expressed in a format which is directly informative about future rewards. It is thus possible that outcomes already acquire a *prospective reward value*, potentially skewing the sampling patterns. In the Horizon task or the sampling paradigm, this can happen because outcomes are directly expressed in points, and in the FIND condition, this can happen because participants already know the target colour (on which they will be rewarded) before the start of the sequence. In this condition, participants did delay proper exploitation to the end of the sequences: the lower tendency to seek the target colour in the FIND than in the DRAW condition, and the difference in target-oriented choices between the last sampling decision and the final question in the FIND condition both attest that the delayed-reward manipulation did work. However, even though they were explicitly not rewarded at each decision, participants did have a significant tendency to sample the target coloured option (visible on the choice direction curves and on the estimates of their sensitivity to the target β_{tar}). In other words, sampling patterns during the sequence were affected by participants’ knowledge of the target colour, which tainted outcomes with prospective reward value, even though actual rewards were delayed.

As an alternative, it is possible that target sensitivity in the FIND condition reflected the fact that knowing the target colour in advance channelled the attention of participants onto a single option at a time. Because we did not ask participants to report “the bluest option”, but simply a blue option, it was sufficient for performance to quickly identify the bluest option in the first couple of trials (possibly through streaks of repeated decisions), and then to only sample the bluest option unless sufficiently disconfirmed. In other

words, knowing the target colour in advance narrowed the focus of the task to a single symbol at a time: it was sufficient to attend to a single shape. Such a strategy would also manifest as a preference for the target colour, and our analysis wasn't designed to disentangle these two interpretations. Yet, the correlation of target sensitivities in the DRAW and FIND conditions (Supp. Fig. 7) suggests that target sensitivity in the FIND condition could reflect outcome valuation— i.e. that outcome valuation would at least partially drive target preference in the FIND condition. Fully disentangling the two interpretations would require additional work, including new task conditions. Regardless of the interpretation, behaviour shows that exploration in conditions of delayed rewards, like FIND, is still not completely free from the attraction of exploitation.

For these reasons, in order to study exploration when it is completely separated from exploitation, we needed a task in which outcomes were not immediately translated into a prospective reward value. Using colour shades gave us this flexibility. Through different instructions, we could manipulate the information and value they carried. In the GUESS condition, we only revealed the rewarding colour at the end of the sequence: during sampling, outcomes were thus completely free of any intrinsic reward value – i.e. it was not any better to see orange or blue outcomes. This condition was thus crucial to study sampling patterns outside of the influence of rewards. And indeed, while directed exploration (the fact that choices were sensitive to the uncertainty of options) was present whenever rewards were delayed (FIND and GUESS conditions), repetitive sampling was most prevalent when rewards were delayed *and* outcomes conveyed no immediate information about the ultimate reward value of options (GUESS). It was sensibly decreased in the FIND condition (where the colour outcomes were immediately translatable into a prospective reward value because participants were given the target colour at the beginning of the sequence). Completely removing the immediate influence of rewards (the GUESS condition) was thus necessary for the novel sampling patterns to manifest. We hope that this helps convince reviewer #2 that the DRAW vs. GUESS contrast is relevant as the main focus of this paper.

We have tried to clarify this in the general introduction p 2-3: *Variants of these tasks have dissociated exploration from reward maximization by delaying the acquisition of rewards. In such tasks, participants either receive information (e.g. ^{9,10} through forced choices) or sample the environment for information (e.g. in the sampling paradigm from Decisions From Experience research^{5,11,12}) before they have to make a rewarding decision. However, in these tasks, the information acquired during the sampling phase immediately concerns the ultimate rewarding values of options. Outcomes do not immediately contribute to participants' final earnings, but since they are expressed in points, they immediately reveal which option will ultimately be rewarding, which potentially biases choices – this is similar to browsing restaurant reviews. **This observation means that the exploration patterns observed in such delayed-rewards tasks are still constrained by the prospective rewarding values of available options. Here we sought to fully dissociate exploration from reward-guided decision-making by designing a novel two-armed bandit task where the outcomes of the two options are shades of colour instead of points or monetary rewards. In themselves, outcomes thus provide information about the structure of the task (the option-colour associations), but carry no intrinsic value (Figure 1.a.).** In one condition, we rewarded participants for drawing one particular colour. This instruction does confer a rewarding value to each outcome, and requires participants to trade exploration against exploitation to choose the option associated with the instructed colour – as in regular bandit tasks. By contrast, in a second condition, the same participants (a within subject design) were told that they were sampling to find an option of one particular colour, which they only had to indicate once, at the end of the sequence of several free choices. Participants were rewarded based on their ability to indicate correct option-outcome associations after the end of the sequence, but each outcome directly informed them about the options' future value – similar to a classic delayed reward task. **In a third, novel condition, the target colour was only revealed at the end of the sampling sequence. Participants sampled the options to learn the option-colour pairing and a final two-alternative forced choice asked them to indicate an option of a specific colour (only revealed then). Because participants could not foresee the ultimate rewarding value of options, this instruction fully dissociates exploration from any form of immediate or***

prospective reward maximisation. Except for this difference, the three conditions were matched in every possible aspect so that sampling patterns could be directly compared between them (Figure 1 & 5; see Methods).”

We have also highlighted this difference when introducing the third dataset (condition FIND), on page 9: “The GUESS condition (in which participants sample freely the two options to learn option-outcome associations), differs from the DRAW condition (in which the same participants seek to sample a rewarded outcome) in two important ways. First, in the DRAW condition, as in traditional bandits, participants are rewarded for the outcomes of each individual choice in the sequence, whereas in the GUESS condition rewards are delayed to a final, single two-alternative forced choice at the end of each sequence. Second, in the DRAW condition, outcomes are associated with a positive or negative value depending on the target colour (they are directly associated to points), whereas in the GUESS condition, they only convey information about the structure of the task (which option is blue or orange), but cannot immediately be related to the future rewarding value of option. While it was the goal of our manipulation to completely release exploration from the influence of rewards, it means that the two factors (delayed exploitation and outcome valuation) were confounded and it was impossible to know the extent to which each of them contributed to the specific exploration patterns observed in the GUESS condition.

To address this issue, we designed a third condition (FIND) where participants were asked to identify, at the end of the sequence, an option associated with a target colour instructed at the beginning of the sequence (Figure 5.a.). In this new condition, as in the DRAW condition, each outcome was associated with a positive or negative value depending on the target colour. However, as in the GUESS condition, participants were not rewarded based on the outcomes they were sampling, but based on their answers to a final two-alternative forced choice at the end of each sequence (where they had to select an option associated with the target colour). This new condition was akin to browsing restaurant reviews: choices yielded no immediate reward, but outcomes were still valued with respect to an end goal. It was also closer to existing delayed-reward tasks, like the Horizon task⁹ or in the “sampling paradigm” from Decisions From Experience studies¹². In these paradigms, rewards are only granted after a sequence of so-called “free choices” (as in the GUESS condition), but (in contrast to the GUESS condition), choice outcomes are numerical values, and therefore directly inform the participants of the option’s ultimate rewarding value. In these tasks, it is thus possible that a future reward value already taints the outcomes and biases choice patterns. While the original GUESS condition was necessary to eliminate this concern, the addition of the FIND condition eliminates a gap in the paradigm. Here, the contrast between the FIND and DRAW conditions reflects the difference between delayed and immediate exploitation, whereas the contrast between the FIND and GUESS conditions reflects the difference between the sampling of outcomes with and without rewarding value.”

And finally tried to highlight the specific contribution of the GUESS condition at the end of the discussion, on page 18: “Taken together, our findings delineate specific features of human exploration patterns in the absence of trade-off with immediate reward maximisation. While random exploration and repetition biases were present in all cases, we only found evidence for directed exploration in cases where rewards were delayed to the end of the sequence. **Additionally, we showed that when outcomes only provide information about the structure of the world (as is often the case in the real world), rather than directly about the option’s ultimate value, participants preferred to initially explore each option in turn before they engaged in directed exploration. Removing the prospective-value bias reveals competing cognitive pressures on information seeking that are neither described nor present in conditions eliciting explore-exploit dilemmas:** 1. a ‘local’ form of information seeking that aims at generating and testing a hypothesis about the currently sampled option, and 2. a ‘global’ form of information seeking that aims at reducing the overall uncertainty about the task structure. These results call for further research into the neural mechanisms of these exploration patterns outside of any explore-exploit dilemma, and into their ecological/psychiatric validity. A general conclusion of our study is that humans display specific exploration patterns when they can freely acquire information that is not associated with

reward. Our findings thus suggest that restricting the study of exploration to explore-exploit dilemmas may have taught us more about how humans arbitrate this trade-off – and its neural correlates, than about exploration itself³².”

2. The main finding is that in the GUESS/FIND conditions, participants show initial “streaks” in choices, repeatedly sampling the same option in a row. The explanation for this is vague, and also not clearly addressed in their model, which just includes this as a component without being able to base this on e.g. aspects of experience or the goal. The explanation provided (“a ‘local’ form of information seeking that aims at generating and testing a hypothesis about the currently sampled option” which contrasts to a “global” form of information seeking through directed sampling of the most uncertain option) may be true, but is entirely speculative. Even the model provided does not include any notion of hypothesis testing,

We understand the reviewer’s point. We have updated the model to a more cognitively motivated version (see below, our response to point (3)), in a way that directly addresses the reviewer’s comment. The duration of the initial streak of repeated choices is now dependent on how much *information* the agent has gathered about the current option. More specifically, in this model, the agent will initially repeat its choice until it has gathered a specific, agent-dependent level of information about the option: It will repeat its choice until it has reached a certain level of confidence about its hypothesis. This revised model yields comparable results to the initial version, but provides a more motivated account of participants’ behaviour. We speculate that participants initially need to form a sufficiently convincing representation of the current option before they switch: in the updated model, how clear this representation needs to be is governed by the θ threshold on accumulated information. Participants first explored locally, by repeatedly sampling the same option in order to form a consistent idea of its associated colour, before they started switching more between options. In the second phase of GUESS sequences, their choices were driven by their sensitivity to the level of information acquired for each option (along with random exploration and a repetition bias), leading them to explore more globally, by targeting the most uncertain option.

3. The “process model”, as I see it, is not really a process model, but more of a data-analytic model. No reason is provided for turning the the “initial sampling” component “on” or “off”. A data-analytic model is fine, and it succeeds in highlighting the differences in search patterns. But then there are claims that certain parameter settings for this model correspond to rational models, yet it is entirely unclear what criterion of rationality is alluded to. If I understand the task correctly, the purported “rational models” would fail at the task completely! The model includes mean colour estimates for each option, as well as what I presume are the absolute values of these estimates. For the “rational” DRAW model, choices are based on the difference in mean colour estimates between the options, i.e. $P(\text{choose } 1) = F(\beta_{\text{(tar)}} \times (\hat{x}_{\text{(t, 1)}} - \hat{x}_{\text{(t, 2)}}))$, with $F(x) = 1/(1 + \exp(-x))$. This is a simple model for a two-armed bandit, but by no means optimal. Given the short horizon, an optimal model might be possible through defining the task as a POMDP (accounting also for the variability in horizons). So the model is not optimal for the task, which is usually what is meant by a rational model. Furthermore, the authors claim that the rational model would set $\beta_{\text{(tar)}} \rightarrow \infty$. This implies a pure exploit strategy, without exploration. This model always chooses option 1 if $\hat{x}_{\text{(t, 1)}} - \hat{x}_{\text{(t, 2)}} > 0$ and option 2 if $\hat{x}_{\text{(t, 1)}} - \hat{x}_{\text{(t, 2)}} < 0$. If both options have a positive mean, but option 2 a higher mean, then it could be the case that after sampling option 1, option 2 is never sampled, so the model would not find the best option. In the present experiment, there are only two possible means for each option, so this is not a practical concern, but were participants aware of this design constraint? For the GUESS condition, the “rational” model bases its choices (if I understand this correctly, the information provided is unclear) on $P(\text{choose } 1) = F(\beta_{\text{(inf)}} \times (|\hat{x}_{\text{(t, 2)}}| - |\hat{x}_{\text{(t, 1)}}|))$. I don’t follow how the absolute mean value estimates encode “information already acquired”. It would make more sense to me to relate this to the number of times an option has been sampled, or a

value encoding the uncertainty associated with the value estimate (which will also depend on the number of samples). Again, the “rational model” sets $\beta_{\text{(inf)}} \rightarrow \infty$, which, if $|\hat{x}_{\text{(t, 2)}} - \hat{x}_{\text{(t, 1)}}|$ encodes information obtained, might make sense here. But to what extent is this rational? Does following these choice strategy maximise the probability of getting the dominant colour for both options right? The model should be specified in much more detail (I had to guess certain parts) and it should be considered how this model provides optimal (or at least very good) performance on the tasks.

This is an important point, and we thank the reviewer for raising it. Based on the remarks of the reviewers, we revised several aspects of the computational analyses: we updated two features of the model (the “learning” component and the threshold on the initial sampling phase), and consequently we changed our fitting procedure. We will detail some changes when addressing the specific points of the reviewer but we have also added more details about the model’s specifics in the methods.

We did use the word “rational” in the sense of “optimal” in the task, and the reviewer was right that the initial version of the model was not rational in this sense. This is because the initial version of the model was based on *averaging* the values of successive outcomes instead of summing them. We have changed this in the revised version: the model now “learns” by perfectly summing the outcomes observed (no leak). In this sense, the revised version of the learning module is Bayes-optimal. Because the colour distributions were designed such that the colour shades were proportional to the log odds ratio of the colour distributions (i.e., each shade carried a specific amount of information about its generative distribution), summing shades observed over successive trials was equivalent to accumulating evidence about the option’s underlying category (see figure a. below).

$$\sum_{1:t} \text{outcome}_i = \log \left(\frac{p(\text{blue} \mid \text{outcome}_1 \dots \text{outcome}_t)}{p(\text{orange} \mid \text{outcome}_1 \dots \text{outcome}_t)} \right)$$

Based on this quantity, at each trial, we could simply identify the option with most evidence in favour of the target colour distribution (target option), and the option with the lowest evidence in favour of any category (the most uncertain option).

The reviewer first raises the question of whether it is optimal in the task to then set $\beta_{\text{tar}} \rightarrow \infty$ in the DRAW condition. As they point out, our task is a specific version of bandits, with only two possible fixed means. Participants know this, and they are familiarised with the distributions first hand during training. Under these circumstances, the agent doesn’t risk fixating on a “less blue” option, and setting $\beta_{\text{tar}} \rightarrow \infty$ (pure exploitation) does maximise the expected “target-ness” of the next outcome (i.e. is optimal). We thank the reviewer for letting us realise this was unclear, we have made this explicit in the *Experimental Procedures* section of the Methods, page 20: “[...] *it was emphasised again for all participants that: 1. the colour intensity of each outcome was relevant, 2. each option would not always draw the exact same shade, so the task was to identify its dominant colour, 3. there were only two possible fixed distributions (blue and orange), 4. the two options were not always associated with different colours (they could be both associated with orange, or blue), 5. each sequence would end after a variable, unpredictable number of choices, and 6. performance was measured by the colours of all sampled outcomes in the DRAW condition, and by the final question in the GUESS and FIND conditions.*”

Second, regarding the optimality of the model in the GUESS condition. The reviewer is correct that absolute mean values did not index the information acquired about the option. As we explained earlier, in the revised version of the model, $|\hat{x}_{2,t} - \hat{x}_{1,t}|$ truly encodes the relative uncertainty about options. From there, the reviewer questions whether it is optimal to set $\beta_{\text{unc}} \rightarrow \infty$ in the GUESS condition. Because of the way we designed the distributions, the probability of correctly identifying the underlying category of any option increased as a function of the level of evidence acquired with the following relation: $\frac{1}{1+e^{-\text{evidence}}}$ (Supplementary Figure 2.a-b.). Because there are only two, symmetrical, distributions, the evidence increase that an agent could expect from sampling an outcome from any option was constant (see an arbitrary example on Supplementary Figure 2.b.). However, because of the convex relationship between evidence and the probability of correctly categorising the option, the same increment in evidence did not

always lead to same gain in performance. More specifically, for the same increment in evidence, the expected performance gain was higher when the initial level of evidence was lower – see figure b. below. For the agent whose goal is to maximise the probability of correctly answering the final question about any option at any time (and provided that the expected evidence boost is a priori equal between options), this means that they can expect a larger performance boost from sampling the option with lowest current level of evidence. In other words, the optimal strategy to maximise performance at the final question is to always sample the option whose current level of evidence is lower. This corresponds to setting $\beta_{unc} \rightarrow \infty$.

A final point raised by the reviewer is whether our model qualifies as a process or data analytic model. We believe that the revised version of the model is a more cognitively motivated account of participants' behaviour than the initially version. The revised model accumulates evidence in a Bayesian fashion and while the original model was simply fitting the number of choices in the initial phase, in the revised version initial repetitions are conditioned on a threshold over accumulated evidence. We believe the revisions make this more of a process model and we thank the reviewer for this comment.

As the reviewer requested, we have added more accurate details about the model specifics in the methods sections “computational model” and “fitting procedures” (p21-23), and we added figures illustrating the model and its mechanics in more details (Figure 3, Supplementary Figure 2.a-b.).

Methods, pages 21-22: **“Computational model:** *The computational model was updated during revisions: again, we first developed the model on the discovery dataset before testing it in the confirmatory samples. The computational model of sampling behaviour featured the same learning module across all three conditions. In practice, the colour distributions were designed such that the log-evidence for the categories (x) was proportional to the colour shades c (ranging from -1 pure blue to +1 pure orange – see Fig.1.).*

$$x_i = \log \left(\frac{p(c_i | blue)}{p(c_i | orange)} \right) \propto c_i, \text{ where } c_i = [-1 : 1]$$

At each trial, the model could then accumulate evidence for each option (\hat{x}) by summing the colour outcomes c observed from each option up to the current trial:

$$\hat{x}_{option,t} = \sum_{1:t} c_i = \log \left(\frac{p(blue | c_1 \dots c_t)}{p(orange | c_1 \dots c_t)} \right)$$

The sum of evidence observed for this option $\hat{x}_{option,t}$, contained information about which colour distribution was more likely to have produced the observed samples ($sign(\hat{x}_{option,t})$), but also about how much information had been accumulated ($|\hat{x}_{option,t}|$). Based on these quantities, the decision variable DV_t was computed at each trial t , as the sum of three components:

$$DV_t = \beta_{tar} * (\hat{x}_{1,t} - \hat{x}_{2,t}) * tc + \beta_{unc} * (|\hat{x}_{2,t}| - |\hat{x}_{1,t}|) + b_{rep} * pc$$

$$tc \in \{-1: blue,$$

$$+1: orange,$$

$$0: \text{in GUESS sequences} \}$$

$$pc \in \{+1: \text{if resp}_{t-1} = 1$$

$$-1: \text{if resp}_{t-1} = 2 \}$$

[revised manuscript text omitted]

4. Whilst the highly detailed analyses have merits, I struggled to “see the forest for the trees”. A much more focused and shorter paper with key analyses presented in the main paper and supporting ones in supplementary material, would work better, I believe. For choice patterns, we get a range of analyses (sampling imbalance, option re-sampling, choice-choice similarity) which all indicate the same thing (early streaks vs gradual increase to preferred option). I suggest focusing on one of these and including other analyses as supplementary material. I also suggest to mainly focus on experiment 3, and report the other two as replicating effects.

We thank the reviewer for helping us trim the presentation of the results: we have removed some behavioural indices and moved some of the replication data to the supplementary material. We have condensed the presentation of the replication results in the third dataset to a single paragraph to shift the focus to the novel (third, FIND) condition, on page 10: *“In this new dataset, participants (N = 31 after exclusion; see Methods) replicated the behavioural patterns previously described in the DRAW and GUESS conditions. They reached similar performance in both of the original conditions (Figure 6.a.), and in the GUESS, but not in the DRAW condition, they initially sampled in streaks of repeated decisions for each option in turn. This was visible both in behavioural indices (Figure 6), and in the distributions of parameters estimates (Figure 7.a.). The statistics for these conditions are reproduced in Supplementary tables 1 to 6. We focus the remainder of this section on the new, FIND condition.”*

We did keep two indices of choice patterns (choice similarity triangles and the fraction of repeat decisions), because they provide specific insights. Choice similarity triangles are a more global, and rawer visualisation of the effect: they allow to clearly display the behaviour of interest at the scale of full sequences, while still being a relatively less processed characterisation of the choices. The curves of the fraction of repeat decisions are more remote from the raw choice data but allow to directly contrast early sampling behaviour in a model-free approach – they reveal a cross-over interaction. We believe both indices are necessary to convey a good understanding of the effect we describe.

We hope that the reviewer accepts our response to point 1 as justification for keeping the DRAW vs. GUESS contrast (experiment 1 and 2) as the main focus of this paper.

5. There is a large literature on contrasting “Decisions from description” to “Decisions from experience”. The latter is either operationalised as a standard multi-armed bandit, or via a “sampling paradigm” where participants can freely sample from the options, until making a final (rewarded) choice. These two operationalisations have also been contrasted (see e.g. Wulff et al 2017, “A Meta-Analytic Review of Two Modes of Learning and the Description–Experience Gap”). This literature is highly relevant but has been (almost) completely ignored in this paper.

This is very relevant indeed and we are grateful to reviewer #2 for suggesting to dive further into this literature. Two papers using the “sampling paradigm” of DFEs (Hills & Hertwig, 2010 and Wulff et al. 2015) are particularly relevant to this study as they directly analysed the exploration patterns during the sampling phase (while, to our knowledge, other DFE papers were more interested in contrasting the final decisions made after experience – akin to our final question –, against those made after receiving descriptive information). We have added these papers to the general introduction page 2: *Variants of these tasks have dissociated exploration from reward maximization by delaying the acquisition of rewards. In such tasks, participants either receive information (e.g. ^{9,10} through forced choices) or sample the environment for information (e.g. in the sampling paradigm from Decisions From Experience research^{5,11,12}) before they have to make a rewarding decision. However, in these tasks, the information acquired during the sampling phase immediately concerns the ultimate rewarding values of options. Outcomes do not immediately contribute to participants’ final earnings, but since they are expressed in points, they*

*immediately reveal which option will ultimately be rewarding, which potentially biases choices – this is similar to browsing restaurant reviews. This observation means that the exploration patterns observed in such delayed-rewards tasks are still constrained by the prospective rewarding values of available options.” We also discuss them in the introduction of the third dataset, as they relate more specifically to the FIND condition we introduced in this dataset, page 9: “This new condition was akin to browsing restaurant reviews: choices yielded no immediate reward, but outcomes were still valued with respect to an end goal. It was also closer to existing delayed-reward tasks, like the Horizon task⁹ or **in the “sampling paradigm” from Decisions From Experience studies**¹². In these paradigms, rewards are only granted after a sequence of so-called “free choices” (as in the GUESS condition), but (in contrast to the GUESS condition), choice outcomes are numerical values, and therefore directly inform the participants of the option’s future rewarding value. In these tasks, it is thus possible that a future reward value already taints the outcomes and biases choice patterns. While the original GUESS condition was necessary to eliminate this concern, the addition of the FIND condition eliminates a gap in the paradigm. Here, the contrast between the FIND and DRAW conditions reflects the difference between delayed and immediate exploitation, whereas the contrast between the FIND and GUESS conditions reflects the difference between the sampling of outcomes with and without rewarding value.”*

However, as we mentioned in our response to the reviewer’s first point, the “sampling paradigm” is closer to the FIND condition: exploitation is delayed to the end of a free exploration phase, but the outcomes of options are numbers, and therefore directly informative about the option’s ultimate rewarding value. The results we observed in the FIND condition (namely that participants’ choices were driven towards the most target option even though this was not increasing their earnings), suggest that outcomes which can be directly related to rewards already orient choice patterns towards the ultimately rewarding option. In this paper, using coloured outcomes instead of numerical points allowed us to introduce the GUESS condition, completely removing the intrinsic value of options: we showed that this reveals specific sampling patterns.

6. The paper is unclear/does not provide crucial information in various places. Key aspects (such as that this was a within-subjects design), should be reported in the introduction or results (this was only evident later when it turned out to be possible to study covariation over conditions).

We have added this detail more explicitly in the general introduction, on pages 2-3: *“Here we sought to fully dissociate exploration from reward-guided decision-making by designing a novel two-armed bandit task where the outcomes of the two options are shades of colour instead of points or monetary rewards. In themselves, outcomes thus provide information about the structure of the task (the option-colour associations), but bear no intrinsic value (Figure 1.a.). In one condition, we rewarded participants for drawing one particular colour. This instruction does confer a rewarding value to each outcome, and requires participants to trade exploration against exploitation to choose the option associated with the instructed colour – as in regular bandit tasks. **By contrast, in a second condition, the same participants (a within subject design) were told that they were sampling to find an option of one particular colour, which they only had to indicate once, at the end of the sequence of several free choices.** Participants were rewarded based on their ability to indicate correct option-outcome associations after the end of the sequence, but each outcome directly informed them about the options’ future value – similar to a classic delayed reward task. In a third, novel condition, the target colour was only revealed at the end of the sampling sequence. Participants sampled the options to learn the option-colour pairing and a final two-alternative forced choice asked them to indicate an option of a specific colour (only revealed then). Because participants could not foresee the ultimate rewarding value of options, this instruction fully dissociates exploration from any form of immediate or prospective reward maximisation. Except for this difference, the three conditions were matched in every possible aspect so that sampling patterns could be directly compared between them (Figure 1 & 5; see Methods).”*

Crucial parts of the design (e.g., is the order of conditions randomised or counterbalanced, and if so how?) are not detailed. The model is described in vague terms, rather than a clear formula of its workings and how parameters are initialized.

We have added more details about the counterbalancing in the Methods section on page 20: *“Conditions were interleaved pseudo-randomly within blocks, with self-paced breaks between blocks. Conditions were pseudo-randomised to balance the number of sequences of each condition within each block, avoid too many (>3) consecutive repetitions of the same condition, and avoid immediate repetitions of the same shapes or the same response sides.”*

In response to the reviewer’s third point, we have updated the computational model and we provide more details about its workings in the Methods on pages 21-22 (this section is reproduced in response to the reviewer’s third point). We have also added figures illustrating the model and its mechanics (see Figure 3, Supplementary Figure 2.a-b.), and we clarified the presentation of the model in the results section. Results, page 6: *“Based on the specific features of human exploration patterns, we designed a process model of sampling choices derived from the optimal sampling agent (Figure 3). In the DRAW condition, based on the perfectly accumulated values of the colour outcomes, the optimal sampling agent always selects the option with highest accumulated value in direction of the target colour (the most target-like option - Figure 3.d., Supplementary Figure 2). Whereas in the GUESS condition, the optimal strategy is to always chose the option with lowest absolute accumulated value (the most uncertain option – Figure 3.d., Supplementary Figure 2.b.). This allows the agent to maximise information about both shapes at any trial – thus maximising its probability of responding correctly to the final question, whose timing was not disclosed to participants. Like the optimal sampling agent, the choices of the model are sensitive to the difference in uncertainty between the two options (sensitivity parameter β_{unc}), and attracted toward the option most likely to be associated with the target colour in the DRAW condition (sensitivity parameter β_{tar}). In addition, and unlike the optimal sampling agent, the model features a repetition bias parameter b_{rep} , and an optional initial sampling phase. In the beginning of the sequence, instead of relying on a decision variable computed from the three choice parameters (β_{unc} , β_{tar} and b_{rep}), the model can acquire, in turn, a number of samples from each option. This initial sampling strategy is controlled by a threshold θ , governing how much evidence the agent wants to initially acquire about the current option before switching to the other (see Methods, Figure 3). We fitted this model to human sampling patterns in the DRAW and GUESS conditions, and confirmed the recoverability of the model and its parameters (Supplementary Figure 5).”*

Regarding parameters initialisation, the fitting procedures for the revised model is now described in more details in the Methods section, on page 22: **“Fitting Procedures:** *Best-fitting parameter estimates for the model, which can either include (ini+) or not include an initial sampling threshold (ini-), were obtained through maximum likelihood estimation (MLE) in two ways. First we used Matlab’s fmincon function to cover a wider range of possible parameter values and avoid local minima (the decision variable parameters were initialised at random points in the range [-2:2] for β_{tar} and β_{unc} and [-4:4] for b_{rep} while the threshold value was set at a hundred different values in the interval [0.05:5]). Second, we used the best-fitting estimates obtained from this first fmincon pass as starting points for the Bayesian Adaptive Direct Search algorithm⁴⁵. This second pass was two-fold: we used leave-one-out cross validation on the sequences used to obtain model likelihood estimates. These were used in Bayesian model selection to compare variants of the model with vs. without the initial sampling phase. BMS was conducted using a random-effects approach, which assumes that different participants may rely on different models. It consists in estimating the distribution over models that participants are drawn from. We used the Dirichlet parameterization of the random-effects approach implemented in SPM12 (<http://www.fil.ion.ucl.ac.uk/spm/>), with default (uniform) priors. We also used the BADS algorithm on all the sequences at once to obtain final parameter estimates⁴⁵.”*

It also seems that the GUESS condition differs between Exp3 and Exp 1&2 (in the last experiment, participants are asked which of two options is blue or orange, whilst in the first two they are asked about the dominant colour of one of the options), but this is not clearly stated.

The wording of the final question was changed in the third dataset to enhance the symmetry between the GUESS and FIND condition. We have added more details about this in the experimental design and procedures section page 21, we thank the reviewer for raising this point. *“In the first two datasets, one shape (pre-selected at random) was displayed and participants were asked to report its associated (dominant) colour. In the third dataset, one colour (pseudo-randomly selected) was displayed on screen and participants were asked to identify an option associated to the demanded colour.”*

How is choice similarity defined?

At the choice level, choice similarity is defined as 1 if the participant chose the same option, and 0 if they chose a different option. Each choice of a sequence was compared to all the other choices in the sequence (so choice 1 was compared to choices 2:end, choice 2 to choices 3:end, etc...). This was averaged across sequences, resulting in an average similarity score (between 0 and 1) for each position in the sequence. We have clarified this definition in the Methods section (reproduced below) and have detailed the description in the figure captions (e.g. Figure 2).

Methods, page 24, section Statistical analyses, paragraph Choice similarity: *“Choice similarity At the choice level, choice similarity is defined as 1 if the participant chose the same option, and 0 if they chose a different option. Each choice in each sequence was compared to all the other choices in the same sequence (choice 1 was compared to choices 2:end, choice 2 to choices 3:end, etc...), resulting in a similarity score for each pair of choice positions within each sequence. This was averaged across sequences, resulting in an average similarity score (between 0 and 1) for each position in the sequence. Because choice similarity is typically not distributed normally across participants, we performed Wilcoxon signed-rank tests to compare participants’ choice similarity to chance level (50%).”*

In Fig 7. In the top plots, it is unclear that subplots compare DRAW to GUESS. Etc.

We have amended this figure to be more specific (it is now Figure 8).

We thank the second reviewer once again for helping us improve the accuracy and clarity of the analyses, and for challenging us to spell out more clearly the exact contributions of this set of experiments. We hope that this revised version of the paper will convince the reviewer.

We thank the reviewers for taking the time again to assess the revised version of our manuscript. We are very pleased to read that Reviewer #1 is satisfied with the revisions, and that Reviewer #2 found the revised manuscript more focused and the modelling improved. We address below the remaining comments of Reviewer #2 in green. Quotes from the new revisions to the manuscript are highlighted in blue.

Reviewer #1 (Remarks to the Author):

I'm satisfied with the revision. One small typo in the abstract: "patterns arising in when" -> "patterns arising when"

We thank the reviewer for spotting this typo. We have corrected it in the revised manuscript. We are happy that the first reviewer is satisfied with the revised manuscript and we thank them again for their insights.

Reviewer #2 (Remarks to the Author):

The authors did a good job responding to the issues raised in the reviews of the previous paper. The presentation is more focused, and the modelling has improved. That said, I still have doubts about the theoretical significance of the findings (initial streaks in the GUESS condition), due to the different goals between the conditions. The focus should be on comparing the two options in the DRAW and FIND conditions (i.e. which option has more of the target colour), but in the GUESS condition, the focus needs to be on identifying the nature of each option, which can be done in isolation and does not require contrasting the options. To determine the nature of a single option, "streaky" exploration may be subject to less memory demands than more evenly distributed exploration. A heuristic for the other conditions is to pitch the options against each other, for instance sampling one and then the other and just determining which one had the most of the target colour in each consecutive pairwise comparison. This seems quite reasonable to me and would account for the results without many of the processes relied upon in the discussion. That said, the finding of a difference in exploration behaviour is supported well by the analyses and the "streaky" exploration is suboptimal when the length of a sequence is unknown and a trial can end at any time. But I do think the discussion can be shortened and streamlined further.

We are very happy that the reviewer found our responses to their previous comments convincing, and the revised manuscript more focused with an improved modelling of human behaviour.

We thank the reviewer for allowing us to further clarify the theoretical significance of our findings. The reviewer proposes that participants sample options in streaks in the GUESS condition because they were free to identify the dominant colour associated with each option *in isolation*. The reviewer suggests that participants do not use this strategy in the DRAW condition because they were pushed to compare options by sampling one after the other to immediately compare the pairs of outcomes.

We believe that this interpretation cannot explain our findings. First, we do not observe basic alternated sampling in the DRAW and FIND conditions, but rather a focus on the

target-coloured option. The model results also suggest that participants were accumulating information over several samples, rather than comparing the pair sample by sample. In addition, repetition biases are positive (meaning that participants do not tend to alternate between options in any condition), comparable and correlated across all three conditions (Figure 7.a. and Supplementary Figure 7.b.), as well as the sensitivity to uncertainty across the GUESS and FIND conditions (Figure 7.a. and additional figure a. below). Given such consistency in parameter estimates across conditions (including repetition biases which would be negative if participants indeed relied on the strategy suggested by the reviewer), it is very unlikely that participants resorted to wildly different heuristics in the different conditions.

From a theoretical perspective, in the GUESS condition the incentive really was on identifying the colour of *both* options *as fast as possible* (since the game ended after a variable number of choices), yet this is the condition where participants did sample in streaks – i.e. they were taking the time to explore a single option at the expense of uncertainty regarding the other option. We fully agree with the reviewer that such “streaky sampling” is likely cognitively easier for participants – e.g. in terms of memory demands – and this is the focus of a dedicated paragraph in the Discussion section. Importantly, there was also no actual need to compare options in the DRAW and FIND conditions. Participants had learnt during the training (both by explicit instructions and by sampling during training games) that the dominant colour of each option was independent of the dominant colour of the other option. This means that comparing options would have been a bad strategy for half of the games where the dominant of the two options was the same. And indeed, as stressed above, we did not find any evidence that participants tended to alternate between options in the DRAW and FIND conditions.

Finally, if sampling in streaks is cognitively easier, and if there was no actual benefit to alternate / compare options in the DRAW (and FIND) condition(s), it is difficult to explain why participants did not produce streaks in this condition but only in the GUESS condition, where they were incentivised to sample from both options in a single game. This means that the presence of rewarding value of outcomes across conditions (present in the DRAW and FIND conditions, absent in the GUESS condition) constitutes a more parsimonious explanation for the different exploration patterns observed in the three conditions.

We have included a discussion of the reviewer’s point in the discussion on page 16: *“An alternative explanation is to propose that participants produced streaks in the GUESS condition because they were free to study each option in isolation (probably because repetitive sampling was cognitively easier than alternating), whereas in the DRAW and FIND conditions they were pushed to compare options by sampling one after the other to immediately compare the pairs of outcomes. However, we did not find any evidence that participants tended to alternate between options in the DRAW and FIND conditions. Rather, repetition biases were positive (meaning that participants did not tend to alternate between options in any condition), comparable and correlated across all three conditions (Figure 7.a. and Supplementary Figure 7.b.), and so was the sensitivity to uncertainty across the GUESS and FIND conditions (Figure 7.a.). Given such consistency in parameter estimates across conditions, it is very unlikely that participants resorted to wildly different heuristics in the different conditions. In addition, participants had learnt during training that the dominant colour of each option*

was independent of the dominant colour of the other option. Thus, comparing options would have been a bad strategy for half of the games where the dominant of the two options was the same. If sampling in series is cognitively easier, and if there was no actual benefit to alternate / compare options in the DRAW and FIND conditions, it is difficult to explain why participants did not produce streaks in this condition but only in the GUESS condition, where they were incentivised to sample from both options in a single game. The presence of (immediate or prospective) reward value appears to be the main determinant of participants' early sampling strategies.”

a sensitivity to uncertainty correlation

Additional figure a. Sensitivity to uncertainty correlation. Correlation of sensitivity to uncertainty estimates (β_{unc}) obtained in the GUESS and FIND conditions. Spearman's correlation.

To follow the reviewer's recommendations, we have edited short parts of the discussion for clarity, but we believe all sections are relevant to discussing the significance of the results. The discussion is longer with the previous suggestions from the reviewers, but we believe these were relevant, sometimes necessary, additions.

One major issue I have with the current version relates to the "Positive covariations between exploration patterns across individuals" section, where participants are split into two groups based on whether the model with initial "streaky" sampling fits better than a model without. Part of the analysis here focuses on the remaining parameters of the model. If I understand the model correctly, then – because the initial choices are governed strongly by the initial sampling policy in the INI+ group, the estimates of the remaining parameters will be mostly determined by later choices (after the initial sampling phase is over). In the INI- group, the remaining parameters likely cover the whole sequence (or at least most of the sequence). Results such as that the sensitivity to the reward maximising option is higher in the INI+ group than the INI- group are then not overly surprising, as one would expect that during the later trials participants would have determined the best option. So, these comparisons are biased due to the parameters in the INI+ group being optimized for the later parts of the sequences, whilst this is less the case for the INI- group. In my view, then, comparisons between the INI+ and INI- group, in terms of model parameters, are not overly meaningful. If such a comparison is of interest, I suggest to only focus on the behavioural results (e.g. proportion of maximising choices, repeat choices, etc.)

We agree with the reviewer that comparing parameters obtained in two different versions of the model would be biased *in principle*. This is why, for this analysis, we split the participants based on which of the models best fit their behaviour in the GUESS condition, but then compared parameters obtained in the most specified version of the model (the ini+ model). This ensures that the “later phase parameters”

(repetition bias and sensitivity parameters) were immediately comparable. The reviewer is right that this was not explicitly described in the text, and we have therefore added the following clarification on page 13 in the revised manuscript: “We split participants based on which of the model with (*ini+* group, $N = 67$) or the model without an initial evidence threshold (*ini-* group, $N = 18$) best fit their behaviour, *but to ensure a fair comparison between groups we then compared parameter estimates obtained in the most specified version of the model (the version with initial threshold, ini+).*”. We thank the reviewer for spotting this imprecision.

However, we would like to point out that the presence of an initial sampling phase of the model does not bias the estimation of “later phase parameters” (repetition bias and sensitivity parameters).

Regarding the sensitivity to the target specifically: in the DRAW condition (where sensitivity to the target is estimated), threshold values estimated from the *ini+* were very close to 0.5 for the large majority of participants, resulting in virtually *ini-* models (i.e. even when taking the *ini+* model, the estimated threshold values are so low that it results in a threshold free, *ini-* behaviour – see additional Figure b. below).

Additionally, we performed recovery analyses showing that all “later phase parameters” (repetition bias and sensitivity parameters) were actually equally well estimated by an *ini+* for agents with or without initial sampling phase. We simulated data from agents with or without the initial sampling phase, and then fitted this data with an *ini+* model (so a model with a free θ parameter, allowing for an initial sampling phase). As illustrated on a new Supplementary Figure 6, we observed that regardless of whether the agent’s behaviour truly displayed an initial sampling phase or an absence of initial sampling phase (simulated as *ini+* or *ini-*), “later phase parameters” were equally well recovered with an *ini+* model. In other words, the additional initial phase parameters featured in an *ini+* model did not introduce a bias in the estimation of later phase parameters (repetition bias and sensitivity to uncertainty).

We hope that this additional analysis helps convince the reviewer that the presence of an initial sampling phase in the model does not negatively impact the estimation of other parameters, and therefore, that splitting participants based on the presence/absence of an initial phase in the best-fitting model for their behaviour does not preclude further analysis of the remaining parameters.

Additional figure b. Threshold level. Threshold levels (θ) obtained in the *ini+* model in the DRAW condition, for participants assigned to *ini+* or *ini-* group based on the fits obtained in the GUESS condition.

As the reviewer suggests, we have emphasised the model-free differences between the two groups. The proportion of reward maximising and the proportion of repeat choices are provided, e.g. on page 13: “However, *ini+* participants also had a stronger

tendency to prefer the target-coloured options in the DRAW condition, which is optimal in this condition (choice direction curves on Figure 8.b.; higher β_{tar} sensitivity parameter on Figure 8.e.; two-sample t -test between groups of participants: $t_{83} = 2.20$, $p < 0.05$, $d = 0.58$)." and "As expected from this splitting criterion, in the GUESS condition, participants in the first group (*ini+*) showed a higher tendency to initially resample options (Figure 8.a.,b.,d.)."

Because we split participants based on which of the model with or without initial sampling phase best fit their behaviour, differences in the proportion of repeat decisions were expected and merely confirm the basis of the split (i.e. that participants whose behaviour is best represented by the *ini+* model display more repeated choices at the beginning of sequences). Thus, we focussed on differences in performance and sensitivity to the target and to uncertainty because these are independent from our splitting criterion. We agree with the reviewer that it is particularly striking that participants assigned *ini+* based on their behaviour in the GUESS condition also have higher performance in *both* conditions. These results suggest that suboptimal exploratory behaviours (initial repetitive sampling) co-occur with more optimal strategies (target-seeking and directed exploration in the DRAW and GUESS & FIND conditions respectively).

A more minor issue is that I think the Horizon task is misrepresented on page 9: "This new condition was akin to browsing restaurant reviews: choices yielded no immediate reward, but outcomes were still valued with respect to an end goal. It was also closer to existing delayed-reward tasks, like the Horizon task⁹ or in the "sampling paradigm" from Decisions From Experience studies¹². In these paradigms, rewards are only granted after a sequence of so-called "free choices" (as in the GUESS condition), but (in contrast to the GUESS condition), choice outcomes are numerical values, and therefore directly inform the participants of the option's ultimate rewarding value." This seems to imply delayed rewards for free choices in the Horizon task, but - as far as I'm aware - rewards are not delayed in the Horizon task, beyond an initial sequence of forced choices which manipulate the amount of information gathered. So I think this is a misrepresentation of the Horizon task.

The reviewer is right that the wording was confusing, we have stated the difference explicitly. p9: "In these paradigms, as in the GUESS condition, rewards are only granted after a sequence of "non-incentivised choices" (*after a predefined sequence of forced choices in the Horizon task, or after a self-terminated sequence of free but non-rewarded choices in the "sampling paradigm"*). However, in both cases choice outcomes are numerical values, and therefore directly inform the participants of the option's ultimate rewarding value, in contrast to the GUESS condition."

Reviewer #2 (Remarks on code availability):

I do not have access to Matlab, on which the code relies. I did have a look and the repository exists. But the contents are from 2 years ago and as the modelling has changed since the last manuscript, I think it needs updating.

The reviewer is correct that we had not yet made the revised code public. It is now updated and publically available at the same link: <https://gitlab.com/cle-a/colrep>

**Reviewer #3**

*The current manuscript investigates information search patterns in three conditions, divorcing*
*information search from choice outcomes to different extents. Across three experiments, evidence is*
*presented that suggests different information search patterns, with conditions with delayed or fully*
*suspended rewards producing behavior directed more strongly at reducing uncertainty. To account for*
*this, the authors suggest a computational model. Overall, the presented results are interesting and*
*suitable for Nature Communications. I am inclined to recommend publication, but I believe the authors*
*should first address the following comments, which may help clarify the exposition and interpretation of*
*their findings. This can be achieved without conducting additional analyses or studies.*

**We thank the reviewer for their evaluation of our work, and we are pleased to read that they are**
**inclined to recommend publication. We have amended the manuscript in line with the reviewer’s**
**recommendations and we hope they will be fully satisfied with the revised submission; we thank**
**them again for their help in improving the final paper.**

*The simplicity of the behavioral paradigm is intriguing but also limiting. It does not afford the study of*
*the full complexity of information search situations, which includes prior expectations and types of*
*uncertainty. In our recent work forthcoming in PNAS (Spektor & Wulff, 2024), we attempted to reveal*
*additional details of the full complexity of human information search by means of analyzing a large*
*database of the sampling paradigm. This revealed that people enter the task with ecological (structural)*
*knowledge, which they update across the task and use to steer their information search behavior. It*
*further revealed that individuals reduce multiple types of uncertainties and differ not only in the details*
*of a common computational strategy but likely recruit entirely different strategies. Finally, it also showed*
*evidence of strategy shifts during the task, which may be related to the local-global strategy shift*
*described here. Not all aspects can be studied with the current paradigm, which is fine. However, I think*
*it is important to emphasize more strongly and discuss in greater length the limitations of both the*
*paradigm and the computational account with respect to capturing the full complexity of human*
*information search behavior.*

**We thank the reviewer for pointing out their recent and relevant paper. The study from Spektor**
**and Wulff focusses on a meta-analysis of the sampling paradigm; it demonstrates that**
**participants’ information search strategies tackle different types of uncertainty (structural**
**uncertainty first then estimation uncertainty, while also minimising computational uncertainty),**
**and that such strategies are informed by prior expectations which are then updated throughout**
**the task, as well as by perceived difficulty. We appreciate the contribution of this large-scale**
**study to the understanding of information search outside of explore-exploit situations, and have**
**added its insights to the following paragraph of the Discussion (p. 16):**

***“However, while our findings highlight a novel balance between local and global***
***uncertainty minimisation, they also relate to recent work showing evidence for strategy***
***shifts over the course of a single sampling sequence. In a three-alternative version of the***
***Horizon task, Dubois and colleagues showed that participants initially relied on a novelty***
***heuristic in addition to more complex exploration computations²⁷. Similarly, Spektor and***
***Wulff⁴³ have shown that in the sampling paradigm of Decisions from Experience (DfE)***
***experiments, participants prioritised observing all possible outcomes first, only***

*switching to targeting the most uncertain options once all unique outcomes had been*
*observed. Our results also document how participants flexibly adjust exploration*
*strategies within a single sampling sequence.”*

Our study differs in its focus from this recent (and relevant) work. Indeed, the aim of our study
is to study exploration in situations where the sampled information is *not even prospectively or*
*conditionally rewarding*. In the sampling paradigm, numerical outcomes can be treated as
prospective rewards. Our control, FIND condition, which mimics the sampling paradigm,
demonstrates that they are. Forewarning participants about which colour will be associated with
rewards in the future (*after the end of the free sampling*) already shapes their sampling
decisions *during* the sampling phase. This is evident in the computational results (participants’
sensitivity to the target-value of options is positive – Figure 7.a), but also in the behaviour
(Figure 6.a). For this reason, the sampling paradigm provides a situation where information
search is released from immediate reward maximisation, but not from the prospective influence
of rewards. To observe exploration outside of the reward constraint, we also need for outcomes
to be as neutral as possible regarding the future rewarding value of options. We used colours as
outcomes because they are fully neutral in isolation, unlike numerical outcomes. In our study,
the GUESS condition is the only one to fully disentangle information search from reward
acquisition. To our knowledge, no other paradigm has afforded a contrast between different
conditions displaying the effect of immediate, prospective or delayed rewards on information
search.

We wholeheartedly agree with the reviewer that not all aspects of human exploration can be
investigated with any single paradigm, but we do think that our main conclusions are warranted
in light of the different points mentioned by the reviewer.

1/ Regarding participants’ expectations. Individual differences in expectations within each
condition were not our focus of our study, and we therefore focused our main analyses on mean
group-level differences between conditions of our paradigm. We would of course be interested
to study whether and how individual differences in expectations may influence exploration
patterns in the GUESS condition where information search is divorced from reward seeking, but
this is not the aim of the present study.

2/ Regarding variability in information search strategies. There is evidence in the data for
between-participant variation in information search strategies (visible in the spread of
behavioural measures and parameters estimates - Figures 2 & 4, 6 & 7). This is the topic of the
last Results section. But the reviewer also suggests that participants could recruit completely
different computational strategies to what we describe and implement in the model. The goal of
our computational account was not to exhaust the full range of possible exploration strategies,
and indeed, we agree with the reviewer that this is always a possibility. Still, our model is
derived from behavioural observations, and it provides a good fit and a plausible explanation for
behaviour in the task. To check that it captures not only the mean behavior of tested
participants but also individual differences in the data, we turned to a principal component
analysis (PCA). We used choice similarity matrices as a summary snapshot of behaviour in the
task, and we contrasted the behaviour of original participants with simulated behaviour using
the best-fitting parameters ($N = 1,000$ simulations). We derived the principal components (PCs)
explaining the behaviour of the original participants on the one hand and the simulated

behaviour on the other hand. Supplementary Figure 11 (reproduced on p. 4 of this point-by-
point response to comments) shows that our computational model accurately captures
individual differences in behaviour, and thus provides a relevant description of participants
behaviour.

The reviewer also mentions potential strategy shifts *during* the task. To check for this, we
assessed the similarity of parameter estimates between the first and the second half of the task
in a new analysis (Supplementary Figure 12, reproduced below on p. 5 of this response).
Overall, parameter estimates did not change substantially between the two parts. Sensitivity to
uncertainty was slightly increased in the second half of the experiment, but with a small effect
size (paired *t*-test in the pooled datasets, first vs. second half $t_{81} = 2.11$, $p = 0.04$, $d = 0.23$). This
may reflect that participants improved with experience in doing the task, but is not convincing
evidence for a full strategy shift *at the scale of the task*. We do, however, document a strategy
shift *at the scale of the single sequence*, and this is similar to what is described in Spektor &
Wulff 2024. This is the main result we report in our study: participants began with local
information seeking (initial repetitive sampling) and later turned to more global search (directed
towards the most uncertain option) – we return to this in response to the reviewer’s next point.
We have emphasized this similarity between the two studies in the Discussion section by citing
the Spektor & Wulff 2024 paper (Discussion, p. 16):

*“Similarly, Spektor and Wulff¹³ have shown that in the sampling paradigm of Decisions*
*from Experience (DfE) experiments, participants prioritised observing all possible*
*outcomes first, only switching to targeting the most uncertain options once all unique*
*outcomes had been observed. Our results also document how participants flexibly adjust*
*exploration strategies within a single sampling sequence.”*

a variance explained in participants' behaviour

by PCs obtained from participants' behaviour

by PCs obtained from simulated behaviour

b first component coefficients and scores

c median split on the scores of the first component in participants' behaviour

split in participants' behaviour

split in simulated behaviour

Supplementary Figure 11. Principal component analysis (PCA) of participants' behaviour vs. simulated behaviour.

The three datasets were pooled for this analysis ($N = 85$). **a. Variance explained in participants' behaviour.** Percentage of variance explained by the first five principal components (PCs) of the PCA on the participants' behaviour (left), and on the simulated behaviour (right), in the DRAW and the GUESS conditions. All three datasets were pooled for this analysis, using only the DRAW and GUESS conditions. The total variance explained by the first five principal components is annotated, plain line represents the cumulative explained variance, and horizontal black lines on the simulation plots (right) reproduce the variance explained in the participants' PCA. **b. First component coefficients and scores.** Comparison of the coefficients and scores of the first component of the PCA done on participants' behaviour v. on simulated behaviour. Bar plots show the cosine similarity of the first component coefficients in each PCA. Scatter plots show the correlation of the individual scores (weights) in the first component of the participants v. the simulations PCAs. Dotted lines are the identity line, solid lines are regression lines. Pearson's ρ correlation coefficients and bootstrapped 95% confidence intervals. **c. Median split on the scores of the first component in participants' behaviour.** Choice similarity matrices plotted for the participants' behaviour, for participants whose first PC was above and below the median (left). And for simulated behaviour based on the best-fitting parameter values of participants whose first PC scores were above or below the median (right).

a distribution of parameter estimates

Supplementary Figure 12. Parameter reliability in the first v. second half of the task. The three datasets were pooled for this analysis ($N = 85$). **a. Distribution of parameter estimates.** For the DRAW (top row) and the GUESS condition (bottom row): distribution of participant's parameter values estimated in the first v. the second half of the task – group means \pm 95% CI plotted in colours, and group medians \pm inter-quartile ranges for the bounded parameters plotted in grey. Data points outside of the first and third quartile by 1.5 inter-quartile ranges were considered outliers and excluded to avoid insensitive testing between first and second halves (Tuckey's method).

Nevertheless, the reviewer is right that we do not claim to offer a definitive explanation for the
full complexity of information search behaviour. We have thus emphasised the limitations of our
paradigm in the following Discussion paragraph (p. 17):

*“It would be interesting for future research to test our findings in more naturalistic*
*environments. Our study also focussed on a specific feature of sampling patterns (their*
*sensitivity to immediate or prospective rewards). Future research should investigate how*
*this feature interacts with dimensions of choice behaviour – for example participants’*
*prior expectations about option statistics, the types of uncertainties displayed¹³, or the*
*underlying structure of the option space³⁸. Another constraint of the laboratory-based*
*setting of the study is the sample size available (three datasets of about 30 participants*
*each). This sample size was sufficient to replicate the effects we had observed in the*
*discovery dataset, which we replicated twice, but it would be interesting for future*
*research to replicate our findings again, in a more representative dataset.”*

We thank the reviewer for her/his insights and comments, which we have used to improve the
discussion of our main findings.

*I believe the distinction between local and global search is interesting. However, I am not sure it makes*
*sense to map it strictly to the reliance on uncertainty minimization. Local search also results in the*
*acquisition of information and thus can be used to reduce one or the other type of uncertainty.*
*Relatedly, it is important to distinguish between exploration and information search. Moreover, I am*
*unsure on what basis the authors infer that participants engage in hypothesis testing during local*
*search. I recommend that the authors clarify the relationship between local and global search and*
*uncertainty minimization, as well as potential evidence of hypothesis testing.*

We thank the reviewer for prompting us to clarify the different concepts at play in our study. We
have updated the manuscript to make sure that we are consistent in the concepts we use, and
we explain our rationale below.

Following most of the literature, we take the standard definition of ‘exploration’ to be choices
that do not maximise immediate reward (Sutton and Barto, 1998), including stochastic (e.g.,
softmax, epsilon-greedy) policies. We use ‘information search’ to denote non-random
exploration: this encompasses directed exploration, but also possibly initial repetitive sampling
in the delayed reward conditions. We agree with the reviewer that both of these, directed
exploration (global search) and initial repetitive sampling (local search) result in uncertainty
minimisation. But they differ in: 1/ the type of uncertainty they target (global uncertainty in the
first case, local uncertainty in the second case), and 2/ their efficiency at increasing
performance in the delayed reward conditions.

In the GUESS condition, because participants could be asked about any option, at any time,
performance relied on maximising knowledge about both options’ statistics at each choice.
Global search decreases global uncertainty about the task structure much more than local
search (see Supplementary Figure 2.b). On the other hand, local search does minimise local
uncertainty, by improving the estimates for the sampled option only – at the expense of global
uncertainty. Thus, while both reduce uncertainty, global uncertainty minimisation offers higher
performance improvements in the GUESS condition, and it is the only strategy expected from an

optimally behaving agent. Therefore, the distinction between local and global search is not so
much about whether one type of search minimizes uncertainty more than the other: it is rather
about the type of uncertainty (local or global) that is being minimised. It is striking that
participants choose to prioritise local uncertainty minimisation at the beginning of each game,
in the condition where they are incentivised to minimise global uncertainty.

Regarding hypothesis testing, we suggest that participants might form a hypothesis on their
very first sample from the option, and go on to test it on the following samples until they were
sufficiently convinced – at which point they could switch options. We phrased this as a
suggestion, but we agree with the reviewer that this concept may confuse readers if it is used
throughout the paper. We have therefore revised the manuscript to restrict the mention of
hypothesis testing to a single paragraph of the Supplementary Discussion section. Importantly,
we now use only “local uncertainty minimisation” to qualify initial repetitive sampling. We
believe this reflects participants trying to form a sufficiently accurate estimation of the current
option before switching to the other option and engaging in global uncertainty minimisation –
what they deem sufficient being reflected in their threshold value. This results in a two-step
strategy, where participants prioritise different forms of uncertainty minimisation over time. This
may be related to the findings described in Spektor and Wulff (2024), where participants’
behaviour was initially best accounted for by a model seeking to observe all possible outcomes
from the current option, before relying on “estimation-driven search” (targeting the most
uncertain option, also known as “directed exploration”, leading to global uncertainty reduction).
We have related the findings of the study from Spektor and Wulff (2024) in the paragraph of the
Discussion cited above. We have also revised the manuscript, including the abstract, to be
consistent in the concepts we use to interpret our findings.

We therefore want to thank the reviewer for prompting us to clarify the concepts, and to
dissipate any confusion in future readers of the paper.

*I believe the authors should discuss the relatively small sample sizes of their studies and potential*
*limitations in terms of representativeness.*

We thank the reviewer for giving us the opportunity to discuss this point further. Indeed, the
sample size of our *discovery* dataset was set to 30 based on existing research performed in
similar (laboratory-based) conditions. While this is still an average sample size for laboratory-
based tasks, it is a relatively small N . On the other hand, the sample size for subsequent
samples was defined through a power calculation on the between-condition differences in
parameter values estimated in the discovery dataset. With an alpha set at 0.05 and a power of
90%, the power analysis called for a sample size of $N < 20$. Considering that replication effect
sizes are often smaller, we collected about 30 participants in subsequent samples (27 in the first
replication, 31 in the second replication with an additional condition) – both of which therefore
exceeded the theoretically needed sample size to replicate our finding with 90% power.

In addition, it should be noted that we acquired a relatively high number of trials per participant,
an important variable often not considered when thinking about sample sizes. Participants
played 96, 128 and 96 sequences in the different datasets (equivalent to what Spektor & Wulff
called “choice problems”), which comprised a total of 1,344 to 1,792 sampling decisions per
participant – this is higher than is typically the case in delayed rewards paradigms (e.g., 404

sampling decisions on average in Spektor & Wulff, 2024). Recent methodological research in
psychology has documented a trade-off between the number of participants and the number of
trials per participant: a higher ratio of trials by participant provides better intra-individual
estimation and lowering the estimates of the sample's variance (Baker et al., 2021, Smith &
Little, 2018). Finally, we would like to emphasize that we report two different replications of our
results in this manuscript. Therefore, while we took a somewhat smaller-*N* approach, we are
confident that the findings we describe are genuine. We have already replicated them in
additional samples for subsequent studies.

Nevertheless, we agree that representativeness is a relevant point, and have therefore reflected
the reviewer's comment on sample size and representativeness in the following section of the
Discussion on page 17:

*“Another constraint of the laboratory-based setting of the study is the sample size*
*available (three datasets of about 30 participants each). This sample size was sufficient*
*to replicate the effects we had observed in the discovery dataset, which we replicated*
*twice, but it would be interesting for future research to replicate our findings again, in a*
*more representative dataset.”*

*Concerning the responses to past reviewers, which I was asked to evaluate, my impression is that the*
*authors place too much trust in their computational model. It is unclear whether it represents a*
*sufficiently accurate and unbiased account of the empirical data to draw such strong inferences. Even if*
*recovery is demonstrated using simulations, it is always wise not to place too much trust in any*
*computational model as the data-generating process could be entirely different. In fact, given the*
*research highlighted in my review, this is to be expected. To address this, I recommend that the authors*
*present a separate descriptive analysis of switching behavior and individual differences to investigate*
*the hypothesis expressed in the first comment of reviewer two and demonstrate the existence of*
*individual differences questioned in the second comment of reviewer two.*

We thank the reviewer for taking the time to also assess our responses to Reviewer #2 – we
hope our response below will dissipate any lingering concerns.

Regarding our use of a computational model to understand behavioural effects. We fully agree
with the reviewer that all models are approximations. Our approach does not exhaustively cover
the full range of possible exploration strategies (i.e., we do not claim that this is the only and
definitive account), nor does it provide the full detail of how computations unfold in participants'
brains (i.e. we are not describing the *neural* data-generating process). But this was not our goal.
Rather, we wanted to devise a plausible approximation of the cognitive mechanisms at play, and
to derive cognitively-relevant variables, as a way to qualify and quantify behaviour more
accurately. The model implements some strategies well established in the literature (reward
seeking, random and directed exploration, and repetition bias – e.g. Sutton & Barto 1998, Wilson
et al., 2014 & 2021, Wu et al. 2018, Dubois et al., 2021), with the addition of a specific early
sampling feature, added as a plausible approximation of the cognitive process underlying early
repetitive sampling. This behaviour had not been documented in related tasks before, so our
study was rooted in behavioural observation, and a discovery-replication approach.

Previous rounds of review have helped us converge on a cognitively relevant account (in terms
of evidence to accumulate before a switch rather than a mere count). This model fulfills its goal.
With a contained number of parameters, it provides a very accurate fit of participant's behaviour
when inspecting simulations of the best-fitting model and knock-out simulations
(Supplementary Figures 3, 4 and 7), it captures both average behaviour and inter-individual
differences (Figure 8, Supplementary Figure 11), and it is also well recoverable (Supplementary
Figures 5 and 6) and replicated well in two independent datasets (Figures 4 and 7). In other
words, we hope to convince the reviewer that while we do not trust our model to account for
each and every aspect of sampling, the approach is methodologically sound, and it is well
suited for its purpose. The model allows us to qualify and quantify participants' behaviour, and
it formulates a cognitively motivated account of the underlying mechanisms.

Now, we address more directly the reviewer's lingering concerns about our responses to the
previous reviewer, and their recommendation for additional descriptive analyses.

In their first comment, Reviewer #2 proposed alternative heuristics to explain the behaviour
difference between the DRAW (immediate rewards) and GUESS (delayed rewards no target)
conditions, based on the different goals of the conditions. We quote Reviewer #2 again here, as
a basis for our response:

Reviewer #2: *"The focus should be on comparing the two options in the DRAW and FIND*
*conditions (i.e. which option has more of the target colour), but in the GUESS condition, the*
*focus needs to be on identifying the nature of each option, which can be done in isolation and*
*does not require contrasting the options. [In the GUESS condition] "streaky" exploration may be*
*subject to less memory demands than more evenly distributed exploration. A heuristic for the*
*other conditions [DRAW and FIND] is to pitch the options against each other, for instance*
*sampling one and then the other and just determining which one had the most of the target*
*colour in each consecutive pairwise comparison"*

First, we would like to mention that our initial submission featured additional behavioural
analyses, but Reviewer #1 and #2 pointed out that presenting several indices demonstrating the
same behaviour was redundant and obscured the message of the paper. The current version of
the manuscript still features the important behavioural indices on Figure 2, Figure 6 and Figure
8a-d. Thus, to address Reviewer #3's recommendations for additional behavioural analyses of
switching behaviour without crowding the manuscript, we provide additional behavioural
analyses for the pooled dataset as Supplementary Information (Supplementary Figure 9), but we
refrain from discussing them at length in the main text.

We take the heuristic suggestion from Reviewer #2 to apply specifically to the initial choices of
a sequence. They argue that the behavioural difference is not due to the reward and the target
colour manipulations as much as it is due to *"the different goals between the conditions"*. Their
proposal is that participants produce repetitive streaks in the GUESS condition because they
learn the option colours *in isolation*, whereas in the DRAW condition they alternate because
they want to *compare* option values. Even in the early choices, in all conditions, participants
repeated their choices more than they switched (Figures 2.c, Supp.1.c, 6.b, Supp.10). So even in
the DRAW condition, they were not doing alternate sampling and pitting options against each
other, as Reviewer #2 suggests.

Still, they did alternate more in the early choices in the DRAW than in the GUESS condition
(Supplementary Figure 10.a) – this is the main effect we report. If this reflected a systematic
comparison strategy to learn option values in the early choices of the DRAW condition, we
would expect (a) that participants switched regardless of the outcome they observed, and (b)
that switching behaviour would change after the initial choices, once participants have gathered
information about option values. To assess this, we plotted the switch rate at choice t , as a
function of the rewarding value of the outcome observed at choice $t-1$ (binned into 8 levels –
Supplementary Figure 10.b). We observed that participants’ decisions to switch were a minority
(always < 0.5), and depended critically on how rewarding the preceding outcome was. We also
observed no evidence for a shift in switching behaviour between early and later choices. In
other words, in the DRAW condition we found no evidence that participants alternated in order
to initially compare options – rather, their decisions to switch appeared to depend primarily on
the rewards obtained (as predicted by our computational model - Supplementary Figure 10.b),
and this was sustained throughout the sequence.

In addition, there was no principled need to compare options in the DRAW condition: i.e. if they
were rewarded for blue and the first option they sampled was blue, they did not need to
investigate the colour of the second option. And indeed, when participants played a DRAW
game in which both options were associated with the target colour, they switched less between
options from early in the game (Supplementary Figure 10.c). On the other hand, there was a
strong incentive to explore both options in the GUESS condition: participants needed to obtain
as much information as possible about both options as soon as possible, before the end of the
game. Yet it is *only* in the GUESS condition that participants chose to learn in streaks of
repeated decisions. While we agree with Reviewer #2 that repetitive sampling is likely
cognitively easier, it is striking that participants only did it when they were incentivised to
minimise global uncertainty about the task structure. We discuss the tension between cognitive
costs and information gathering on p. 16:

*“Thus, it is striking that participants chose to prioritise local uncertainty minimisation*
*precisely in conditions where they were incentivised to minimise global uncertainty. This*
*likely reflects a trade-off between information search and cognitive costs: while directed*
*sampling of uncertain option was most efficient in reducing global uncertainty, it was*
*also cognitively costly to update beliefs about several options in parallel – e.g. ‘switch*
*costs’, which have been associated with mental effort^{24,25} and even subjective pain²⁶.*
*Initially prioritising local information by chunking samples into streaks of a single option*
*likely alleviated these costs. This strategy, however, was only used by tested participants*
*when rewards were delayed.”*

We mention the hypothesis from Reviewer #2 and discuss these points in the Supplementary
Discussion of the Supplementary Information document:

*“Another alternative explanation for the difference in sampling patterns between*
*conditions is to propose that participants relied on different heuristics because the*
*instructions gave them different goals. One could propose that participants repeated*
*choices in the delayed rewards (GUESS) condition because they were free to study each*
*option in isolation, whereas in the DRAW and FIND conditions they were pushed to*

compare options by sampling one after the other, juxtaposing pairs of outcomes. We did not find any evidence that participants tended to alternate between options in the DRAW and FIND sequences (Supplementary Figure 10). Rather, repetition biases were positive (meaning that participants did not tend to alternate between options in any condition), comparable and correlated across all three conditions (Figure 7.a. and Supplementary Figure 7.b.), and so was the sensitivity to uncertainty across the GUESS and FIND conditions (Figure 7.a.). Given such consistency in parameter estimates across conditions, it is very unlikely that participants resorted to wildly different heuristics in the different conditions. In addition, in the DRAW condition, even in the initial choices, participants' decisions to switch option appeared to depend mainly on the reward value of the preceding outcome (Supplementary Figure 10), rather than demonstrating systematic alternation."

We hope that our additional analyses and explanations will convince Reviewer #3 that the hypothesis from Reviewer #2 cannot not explain the behavioural effects we find and report.

Supplementary Figure 10. Alternation behaviour in the initial choices. Figures display the pooled datasets ($N = 85$). **a. Alternation behaviour.** Average switch rate at each choice position in the sequence (group mean \pm 95% within participant CI). **b. Alternation rate by previous outcome.** In the DRAW condition, alternation rate in participants' decisions as a function of the rewarding value of the preceding outcome (binned into 8 bins), in the first four, following four, and remaining twelve choices. Group means (dots) and 95% CI (confidence bars). **c. Choice to choice similarity.** Average similarity of each choice to the other choices in the same sequence. In the DRAW condition, for sequences where a both, a single, or none of the options were associated to the target colour. In the GUESS condition, for sequences where the options were associated to the same, or different colours.

The second comment of Reviewer #2 relates to the "Positive covariations between exploration patterns across individuals" section. Reviewer #2 was concerned that the inter-individual differences in parameter values we report were based on a biased estimation of those parameters, and reflected differences in model fitting rather than genuine differences in behaviour.

For this analysis, participants were split between an INI+ and an INI- group, based on which model best fit their behaviour in the GUESS condition. Although participants were split based on which model (INI+ or INI-) best fit their behaviour, we compared parameters using the most parameterized version of the model (INI+). This approach allowed for a fair comparison between parameters, and for participants whose best-fit was the INI- model, threshold parameters in the INI+ model were estimated close to 0, effectively rendering it equivalent to an INI- model while accounting for the additional parameter. Based on this split, we reported group differences both in behavioural indices (e.g. higher performance in both conditions for the INI+ group) and in parameter estimates (higher estimates for the later phase parameters in the INI+ group: sensitivity to the target, to uncertainty, and repetition bias).

Reviewer #2's concern was that the later phase parameters would mechanically differ between groups due to not being optimised on the same choices (i.e. estimated on the full sequence in the INI- participants vs. only after the initial phase was over in the INI+ participants). In other words, Reviewer #2 suspected that the inter-individual differences in parameter estimates were a trivial by-product of the model fitting, rather than genuine differences in sampling strategies between participants.

We agreed with Reviewer #2 that this was a legitimate concern, and we have demonstrated through recovery that later phase parameters were equally well estimated regardless of the presence or absence of an initial sampling phase (Supplementary Figure 6). It was not the case that later phase parameters were mechanically biased by the presence or absence of an initial phase. Differences in parameter estimates between the INI+ and INI- groups do not trivially surface from the model fitting pipeline, but rather, the difference in parameters suggests underlying differences in sampling strategies. More specifically, that seemingly contradictory sampling strategies co-occur in participants (i.e. suboptimal initial repetitive sampling and optimal directed exploration), and that statistically suboptimal strategies can be associated with higher performance in participants (higher evidence for the initial sampling phase in the delayed rewards condition was associated to higher performance in both conditions). We hope that this clarification will convince Reviewer #3 that the model-based analyses are actually well suited for measuring the inter-individual differences questioned by Reviewer #2. We would like to reiterate that we acknowledge that, like all models, our model is ultimately an approximation of cognitive strategies. Yet the strict methodology we applied makes it valuable in revealing genuine differences in behaviour.

Reviewer #3 has also recommended that we perform a separate descriptive analysis to convince them of the existence of individual differences between INI+ and INI- participants. Because we split participants based on which of the INI+ or INI- model best fit their behaviour, some analyses of repetition rate are circular. For example, the analysis of repetition rate by choice

position, repetition rate by familiarity with the option, or choice imbalance are trivially expected
to differ between groups (Figure 8.a and Supplementary Figure 9.b). We mention this in the
corresponding subsection of the Results on p. 13:

“As expected from this splitting criterion [emphasis added], in the GUESS condition,
participants in the first group (INI+) showed a higher tendency to initially resample
options (Figure 8.a., b., d.)”

For this reason, we have focused our account around differences in choice direction and in
performance (Figure 8.b-c.). Both of these are behavioural indices, and they are not trivially
expected to differ from our splitting criterion. It is particularly striking that participants who
displayed stronger evidence for *suboptimal* initial repetitive sampling in the GUESS condition
(the INI+ group) also achieved higher performance in both tasks – a purely model-free measure
(Figure 8.c.). Additionally, the INI+ participants displayed more condition-relevant choices in
both conditions (Figure 8.b.). Such co-occurrence is surprising because initial repetitive
sampling is suboptimal yet associated with task-optimal choices, and with higher performance.
It remains to be investigated whether performance is directly improved by initial repetitive
sampling, or whether it is merely dependent on otherwise higher task-optimal behaviour in the
same participants (higher reward-drive and higher sensitivity to uncertainty) – but this was
beyond the scope of this initial study.

We hope that our additional explanation of the recovery analyses and of the rationale behind
our focus on performance and choice direction will convince Reviewer #3 of the existence of the
individual differences questioned in the second comment of Reviewer #2.

Supplementary Figure 9. Additional behavioural indices of the inter-individual differences in sampling patterns. The three datasets were pooled and participants were split in two groups based on whether a model with (*ini+*, $N = 67$) or without (*ini-*, $N = 18$) an initial sampling phase better accounted for their exploration patterns in GUESS sequences. **a. Overall fraction of repeat decisions.** Fraction of repeat decisions in each condition, averaged per participant for all the participants (left), and for the *ini+* and *ini-* participants separately (right) – median \pm inter-quartile range. **b. Choice imbalance.** Difference between the number of choices for the most and the least chosen option at each choice throughout the games. Solid lines display the average for participants whose best-fitting model in the GUESS condition included an initial sampling phase (*ini+*, $N = 67$), dotted lines show the average of participants whose best-fitting model did not include an initial sampling phase (*ini-*, $N = 18$) – group means \pm 95% CI.

Finally, I would appreciate it if the authors could clarify aspects of their model. If I understand correctly, the authors model whether a sampling decision is repeated as a function of the decision variable DV, independent of which option was sampled last. Consequently, I would expect that the probability of a repeat should be based on the absolute differences in the amount and direction of evidence for the two options rather than their signed differences (i.e., $\hat{x}_{1,t} - \hat{x}_{2,t}$ and $|\hat{x}_{1,t}| - |\hat{x}_{2,t}|$), but the definition of DV suggests the opposite. The result is that only differences in one direction (e.g., $\hat{x}_{1,t} > \hat{x}_{2,t}$) but not $\hat{x}_{1,t} < \hat{x}_{2,t}$) can trigger a switch to the other option. I don't understand how this makes sense.

We are happy to clarify our model formulation, since it triggered confusion which we would very much like to dissipate in the final version of the manuscript. Outside of the initial sampling phase, the decision variable (DV) used by the model to choose between option 1 and option 2 – using a standard softmax policy – consists of three terms: 1/ the difference in the log-odds that each option is associated with the target colour, controlled by a target sensitivity parameter β_{tar} , 2/ the difference in the absolute log-odds that each option is associated with a colour (i.e., the certainty of the association between each option and a colour), controlled by an uncertainty sensitivity parameter β_{unc} , and 3/ a bias toward sampling the same option as in the previous trial, controlled by a repetition bias parameter b_{rep} .

The first term controls reward sensitivity (in the DRAW and FIND conditions), whereas the second term reflects directed exploration (as in Wilson et al., 2014), where the model will preferentially sample (when $\beta_{unc} > 0$) the option whose value is most uncertain (as in e.g. Gershman 2018, Wilson 2014, Wilson 2021). The decision variable DV does make reference to which option was sampled on the previous trial, with the inclusion of the third term – the repetition bias. But to be perfectly explicit, our model computes the probability of choosing option 1 over option 2, not the probability of re-sampling the previous option. See below the pseudo-code adapted from <https://gitlab.com/cle-a/COLPUB>:

```
% accumulate information about each option
x(t,1) = x(t-1,1) + previous_outcome; % update option 1 (chosen)
x(t,2) = x(t-1,2); % do not update option 2 (not chosen)

% at each sample, compute the decision variable DV
dv = beta_tar * (x(t,1)-x(t,2)) * target_color; % first term
dv = dv + beta_unc * (abs(x(t,2))-abs(x(t,1))); % second term
dv = dv + beta_rep * previous_choice; % third term

% compute the choice probability of each option
choice_prob(1) = 1/(1+exp(-dv)); % p(choose option 1)
choice_prob(2) = 1-choice_prob(1); % p(choose option 2)
```

In this code, a higher β_{tar} will bias the agent/participant towards choosing the target-coloured
option (it is only defined when the target is given in the beginning of the sequence - in DRAW).
A higher β_{unc} will drive choices towards the most uncertain option (the option with least
accumulated evidence). Finally, a higher b_{rep} pushes choices towards re-sampling the same
option as in the previous trial. Note that in the online code, β_{unc} was positive when the model
was attracted toward the most certain option (which is the opposite of directed exploration),
such that negative values were observed in human participants. We have above (before the
original submission, but after putting the first version of the model code online) flipped β_{unc} to
be positive when the model is attracted toward the most uncertain option. We hope this clarifies
any lingering ambiguity about the implementation of the model.

To conclude, we thank Reviewer #3 for taking the time to assess our manuscript as well as our
responses to Reviewer #2. We are pleased to read that they are inclined to recommend
publication, and we hope that our detailed responses to each of their different suggestions have
addressed all remaining concerns. Please find references and copies of the figures that are
useful when reading this response in the remaining pages below.

References:

- Baker, D. H., Vilidaite, G., Lygo, F. A., Smith, A. K., Flack, T. R., Gouws, A. D., & Andrews, T. J. Power contours: Optimising sample size and precision in experimental psychology and human neuroscience. *Psychological methods*, 26(3), 295 (2021).
- Dubois, M., & Hauser, T. U. Value-free random exploration is linked to impulsivity. *Nature Communications*, 13(1), 4542 (2022).
- Smith, P. L., & Little, D. R. Small is beautiful: In defense of the small-N design. *Psychonomic bulletin & review*, 25, 2083-2101 (2018).
- Spektor, M. S., & Wulff, D. U. Predecisional information search adaptively reduces three types of uncertainty. *Proceedings of the National Academy of Sciences*, 121(47), e2311714121 (2024).
- Sutton, R. S., & Barto, A. G. Reinforcement learning: An introduction (Vol. 1, No. 1, pp. 9-11). Cambridge: MIT press (1998).
- Wilson, R. C., Geana, A., White, J. M., Ludvig, E. A., & Cohen, J. D. Humans use directed and random exploration to solve the explore–exploit dilemma. *Journal of experimental psychology: General*, 143(6), 2074 (2014).
- Wilson, R. C., Bonawitz, E., Costa, V. D., & Ebitz, R. B. Balancing exploration and exploitation with information and randomization. *Current opinion in behavioral sciences*, 38, 49-56 (2021).
- Wu, C. M., Schulz, E., Speekenbrink, M., Nelson, J. D., & Meder, B. Generalization guides human exploration in vast decision spaces. *Nature human behaviour*, 2(12), 915-924 (2018).

a overall fraction of repeat decisions

b choice imbalance

Supplementary Figure 9. Additional behavioural indices of the inter-individual differences in sampling patterns. The three datasets were pooled and participants were split in two groups based on whether a model with (*ini+*, $N = 67$) or without (*ini-*, $N = 18$) an initial sampling phase better accounted for their exploration patterns in GUESS sequences. **a. Overall fraction of repeat decisions.** Fraction of repeat decisions in each condition, averaged per participant for all the participants (left), and for the ini+ and ini- participants separately (right) – median \pm inter-quartile range. **b. Choice imbalance.** Difference between the number of choices for the most and the least chosen option at each choice throughout the games. Solid lines display the average for participants whose best-fitting model in the GUESS condition included an initial sampling phase (ini+, $N = 67$), dotted lines show the average of participants whose best-fitting model did not include an initial sampling phase (ini-, $N = 18$) – group means \pm 95% CI.

a alternation behaviour

b alternation rate by previous outcome

c choice to choice similarity

Supplementary Figure 10. Alternation behaviour in the initial choices. Figures display the pooled datasets ($N = 85$). **a. Alternation behaviour.** Average switch rate at each choice position in the sequence (group mean \pm 95% within participant CI). **b. Alternation rate by previous outcome.** In the DRAW condition, alternation rate in participants' decisions as a function of the rewarding value of the preceding outcome (binned into 8 bins), in the first four, following four, and remaining twelve choices. Group means (dots) and 95% CI (confidence bars). **c. Choice to choice similarity.** Average similarity of each choice to the other choices in the same sequence. In the DRAW condition, for sequences where a both, a single, or none of the options were associated to the target colour. In the GUESS condition, for sequences where the options were associated to the same, or different colours.

a variance explained in participants' behaviour

by PCs obtained from participants' behaviour

by PCs obtained from simulated behaviour

b first component coefficients and scores

c median split on the scores of the first component in participants' behaviour

split in participants' behaviour

split in simulated behaviour

Supplementary Figure 11. Principal component analysis (PCA) of participants' behaviour vs. simulated behaviour.

The three datasets were pooled for this analysis ($N = 85$). **a. Variance explained in participants' behaviour.** Percentage of variance explained by the first five principal components (PCs) of the PCA on the participants' behaviour (left), and on the simulated behaviour (right), in the DRAW and the GUESS conditions. All three datasets were pooled for this analysis, using only the DRAW and GUESS conditions. The total variance explained by the first five principal components is annotated, plain line represents the cumulative explained variance, and horizontal black lines on the simulation plots (right) reproduce the variance explained in the participants' PCA. **b. First component coefficients and scores.** Comparison of the coefficients and scores of the first component of the PCA done on participants' behaviour v. on simulated behaviour. Bar plots show the cosine similarity of the first component coefficients in each PCA. Scatter plots show the correlation of the individual scores (weights) in the first component of the participants v. the simulations PCAs. Dotted lines are the identity line, solid lines are regression lines. Pearson's ρ correlation coefficients and bootstrapped 95% confidence intervals. **c. Median split on the scores of the first component in participants' behaviour.** Choice similarity matrices plotted for the participants' behaviour, for participants whose first PC was above and below the median (left). And for simulated behaviour based on the best-fitting parameter values of participants whose first PC scores were above or below the median (right).

a distribution of parameter estimates

Supplementary Figure 12. Parameter reliability in the first v. second half of the task. The three datasets were pooled for this analysis ($N = 85$). **a. Distribution of parameter estimates.** For the DRAW (top row) and the GUESS condition (bottom row): distribution of participant's parameter values estimated in the first v. the second half of the task – group means \pm 95% CI plotted in colours, and group medians \pm inter-quartile ranges for the bounded parameters plotted in grey. Data points outside of the first and third quartile by 1.5 inter-quartile ranges were considered outliers and excluded to avoid insensitive testing between first and second halves (Tuckey's method).

Point by point response to the reviewers

Reviewer #4

Having reviewed R3's comments and the author's responses to them, I believe that the authors have done a great job addressing all the comments thoroughly and convincingly, substantially improving the conceptual clarity of the paper and adding helpful model-independent analyses.

We thank the reviewer for taking the time to evaluate our response to Reviewer 3, and we are pleased to read that they are satisfied with our revisions.